

# Permafrost response and feedback under temperature stabilization and overshoot scenarios with different global warming levels

Min Cui[1], Duoying Ji[1] and Yangxin Chen[1]

[1]College of Earth System Science, Faculty of Geographical Science, Beijing Normal University, Beijing, 100875, China

*Correspondence to*: Duoying Ji (duoyingji@bnu.edu.cn)

**Abstract.**

Permafrost regions in the northern high latitudes face significant degradation risks under global warming and threaten the achievement of global climate goals. This study explores nonlinear permafrost response and feedback under temperature stabilization (SWL) and overshoot (OS) scenarios with various global warming levels (GWLs). Under the 1.5 °C and 2 °C
SWL scenarios, permafrost area loss is 4.5 [4.4 to 4.7] million km$^2$ and 6.5 [6.4 to 6.8] million km$^2$ respectively. In the OS scenarios, permafrost area can recover effectively, with an additional loss of only 0.3~1.1 million km$^2$ compared to the 1.5 °C SWL scenario. However, permafrost carbon loss in the OS scenarios is irreversible, with 9~44 PgC less loss compared to the SWL scenarios. Both SWL and OS scenarios show that additional warming due to permafrost carbon feedback rises with higher GWLs, and the most substantial permafrost carbon feedback in OS scenarios is anticipated to take place during the
cooling phase. In the OS scenarios, the proportion of additional permafrost area loss due to permafrost carbon feedback increases with higher GWLs, reaching 6~12 % of total permafrost degradation. In contrast, under the SWL and SSP5-8.5 scenarios, additional permafrost area loss generally decreases as GWLs rise. The additional permafrost area loss due to permafrost carbon feedback is influenced by both the magnitude of additional warming and the sensitivity of permafrost area to global warming (SPAW). The maximal SPAW falling between 1.5 °C and 2 °C has significant implications for achieving
the global warming levels of the Paris Agreement.

## 1 Introduction

Permafrost soils in the northern high latitudes contain an estimated 1100-1700 Pg of carbon, primarily in the form of frozen organic matter, which is roughly twice the amount of carbon in the atmosphere (Hugelius et al., 2014; Schuur et al., 2015). As the climate warms, the thawing of permafrost and subsequent microbial decomposition would gradually or
abruptly release carbon dioxide ($CO_2$) and methane ($CH_4$) into the atmosphere, thereby amplifying the warming effect (Koven et al., 2011; Feng et al., 2020; Smith et al., 2022). The positive feedback mechanism, combined with the fact that warming rates in the Arctic exceed the global average (Fyfe et al., 2013; Liang et al., 2022; Rantanen et al., 2022), underscores the critical role of permafrost as a key tipping element in the climate system (Armstrong McKay et al., 2022). However, current Earth system models inadequately represent or omit the permafrost carbon processes, which has become



one of the largest sources of uncertainty in future climate projections (Schädel et al., 2024). Therefore, researching the release of permafrost carbon and its feedback is crucial for accurately assessing climate risks and formulating effective emission reduction strategies.

The Paris Agreement aims to limit global average temperature rise to well below 2 °C above pre-industrial levels, with efforts to keep it below 1.5 °C. Despite these goals, global warming has already exceeded 1 °C and is on track to surpass 3
°C by the end of the 21st century, primarily due to increased anthropogenic $CO_2$ emissions (Haustein et al., 2017; Hausfather and Peters, 2020). If current emission rates persist, the remaining carbon budgets compatible with the 1.5 °C or 2 °C targets will be critically tight and likely exhausted within the next few years (Rogelj et al., 2015; Goodwin et al., 2018; Masson-Delmotte et al., 2018; Forster et al., 2023; Smith et al., 2023). It is unlikely that the 1.5 °C target set by the Paris Agreement will be met (Raftery et al., 2017). However, it might still be achievable after a period of temperature overshoot, by
compensating for excessive past and near-term emissions with net-negative emissions at a later time – i.e., through on-site $CO_2$ capture at emission sources and carbon dioxide removal from the atmosphere (Gasser et al., 2015; Sanderson et al., 2016; Seneviratne et al., 2018; Drouet et al., 2021; Schwinger et al., 2022).

Several existing studies have assessed the climate response to overshoot pathways. Many components of the physical climate system have been identified as reversible, although typically with some hysteresis behavior (Boucher et al., 2012;
Wu et al., 2015; Tokarska and Zickfeld, 2015; Li et al., 2020; Cao et al., 2023). In this context, reversibility refers to a partial recovery of climate conditions in an overshoot scenario toward an Earth system state without overshoot. These studies demonstrate that global mean surface air temperature (GSAT), sea surface temperature and permafrost area can recover within decades to centuries in response to net negative emissions. Carbon release from permafrost has been shown to be irreversible on multi-decadal to millennial timescales (MacDougall et al., 2013; Schwinger et al., 2022). Additionally, a
temporary warming of the permafrost regions entails important legacy effects and lasting impacts on its physical state and carbon cycle under various overshoot levels (de Vrese and Brovkin, 2021; Schwinger et al., 2022). However, these studies have yet to assess permafrost carbon-climate feedback under overshoot scenarios.

Few studies have examined permafrost carbon response and feedback under long-term climate stabilization scenario, but such scenarios may be more realistic if ambitious emission mitigation strategies are not implemented or carbon dioxide
removal methods are not effective. Especially the most carbon dioxide removal methods have not yet been proven on a large scale (Anderson et al., 2023), and sustainability concerns further limit land-based carbon dioxide removal options (Deprez et al., 2024). Multi-model ensemble mean projections suggest that global temperature change following the cessation of anthropogenic greenhouse gas emissions will likely be close to zero in the decades, although individual models show a range of temperature evolution after emissions cease from continued warming for centuries to substantial cooling (MacDougall et
al., 2020; MacDougall, 2021; Jayakrishnan et al., 2024). MacDougall (2021) demonstrated that under a cumulative emission scenario of 1000 PgC, permafrost carbon-climate feedback could raise global temperature by approximately 0.06 °C within 50 years after emissions cease. However, systematic research is still lacking on how permafrost carbon response and feedback under temperature stabilization scenarios and how that differs from temperature overshoot scenarios reaching the



same peak warming. Additionally, most post-Paris Agreement studies have focused on understanding the impacts of 1.5 °C
and 2 °C global warming levels (GWLs), with little attention given to the higher GWLs (Rogelj et al., 2011; Comyn-Platt et
al., 2018; King et al., 2024). Assessment of permafrost carbon-climate feedback under various GWLs would provide a more
comprehensive understanding of the critical role of permafrost in the climate system.

This study aims to fill these gaps using an Earth system model of intermediate complexity to systematically assess the
permafrost response and feedback under temperature stabilization or overshoot scenarios achieving various GWLs. The
structure of the remainder of this paper is as follows: Section 2 introduces the model description, experimental design of
temperature stabilization and overshoot scenarios at different GWLs, as well as the methodology for perturbed parameter
ensemble experiments. Section 3 provides the results of these experiments. Finally, Section 4 offers conclusions and
discussion of our findings.

## 2 Methods

### 2.1 Model description

This study uses the University of Victoria Earth System Climate Model version 2.10 (UVic ESCM v2.10), an
intermediate-complexity Earth system model with a uniform horizontal resolution of 3.6° longitude by 1.8° latitude, to
simulate permafrost carbon response and feedback under temperature stabilization and overshoot scenarios. The atmospheric
component of UVic ESCM is a single layer moisture-energy balance model. The oceanic component incorporates a fully
three-dimensional general circulation model, with a vertical resolution of 19 levels, coupled to a thermodynamic and
dynamic sea-ice model. The terrestrial component uses the Top-down Representation of Interactive Foliage and Flora
Including Dynamics (TRIFFID) vegetation model, which represents vegetation dynamics including five plant functional
types. A more detailed description of UVic ESCM v2.10 can be found in Weaver et al. (2001) and Mengis et al. (2020).

In the UVic ESCM v2.10, permafrost carbon is represented as a separate carbon pool. It is generated when carbon is
transported across the permafrost table through a diffusion-based cryoturbation scheme and can only be decomposed into
$CO_2$ (MacDougall and Knutti, 2016). The permafrost carbon pool is characterized by four key parameters: (1) a decay rate
constant; (2) the available fraction of the pool, which represents the combined size of both the fast and slow carbon pools
subject to decay; (3) a passive pool transformation rate, which governs the rate at which passive permafrost carbon
transitions into the available fraction; and (4) a saturation factor used to calibrate the total size of the permafrost carbon pool,
which is linked to soil mineral porosity and accounts for the decreasing concentration of soil carbon with depth in permafrost
regions (Hugelius et al., 2014). These four key parameters determine the size and vulnerability to decay of the permafrost
carbon pool. However, the UVic ESCM v2.10 only simulates permafrost carbon in the top 3.35 m of soil, omitting soil
carbon stored in deep deposits of Yedoma regions. A more detailed description of the UVic ESCM's permafrost carbon
parameterization scheme and its simulated permafrost carbon characteristics are discussed in MacDougall and Knutti (2016).





As an Earth system model of intermediate complexity, the UVic ESCM offers relatively low computational costs and serves as an ideal instrument for performing experiments that are not yet feasible with state-of-the-art Earth system models (Weaver et al., 2001; MacIsaac et al., 2021). In recent years, the UVic ESCM has played a key role in assessments of carbon-climate feedbacks (Matthews and Caldeira, 2008; Matthews et al., 2009; Tokarska and Zickfeld, 2015; Zickfeld et al., 2016) and long-term climate change uncertainties (Ehlert et al., 2018; Mengis et al., 2020; MacDougall et al., 2021).

UVic ESCM v2.10 is able to reproduce historical temperature and carbon fluxes changes well (Mengis et al., 2020), as compared with the observational dataset (Haustein et al., 2017) and Global Carbon Project 2018 (Le Quéré et al., 2018). Moreover, it has been validated for simulating permafrost area, permafrost carbon stocks and the distribution of anthropogenic carbon emissions across the atmosphere, ocean and land (Mengis et al., 2020).

## 2.2 Experimental design

To quantify the impact of permafrost carbon response under different climate conditions, we designed a series of idealized climate scenarios, including four stabilized warming level (SWL) and three overshoot (OS) trajectories, spanning a long period from 1850 to 2300. In these scenarios, the GSAT increases to a maximum of 1.5 °C, 2 °C, 3 °C, and 4 °C above pre-industrial levels (1850-1900). Following this initial warming phase, the scenarios diverge into two groups following distinct temperature trajectories. In the SWL scenarios, temperature stabilizes at the respective GWLs and remain steady

until 2300, referred to as SWL-1.5, SWL-2, SWL-3, and SWL-4. In contrast, the OS scenarios allow temperatures to temporarily reach these warming peaks before entering a cooling phase symmetrical to the warming phase, gradually reducing the GSAT to 1.5 °C above pre-industrial levels, where it remains steady until 2300. These overshoot pathways are referred to as OS-2, OS-3, and OS-4, respectively.

A simple proportional control scheme (Zickfeld et al., 2009) is used to derive the $CO_2$ emissions pathway

corresponding to each SWL and OS scenarios, adjusting emissions to align the GSAT with the prescribed trajectory. The proportional control equation is given by:

$$E_{i+1} = k_{PE}\left(T_i - T_{i,goal}\right),$$
(1)

Where $E_{i+1}$ represents the carbon emission at year $i + 1$, $T_i$ represents the simulated annual mean GSAT at year $i$, and $T_{i,goal}$ represents the prescribed GSAT trajectory at year $i$ for each scenario. The time-invariant coefficient $k_{PE}$ is set to 1515 PgC

$K^{-1}$, ensuring that the control scheme responds neither too quickly nor too slowly to the diagnosed temperature derivation. Base on the proportional control scheme, a set of emission trajectories were derived from initial experiments of SWL and OS scenarios using the UVic ESCM v2.10 without the permafrost carbon module. Specifically, all these initial experiments of SWL and OS scenarios firstly follow a unified emissions trajectory, which is based on historical $CO_2$ emissions (Friedlingstein et al., 2022) until 2021, and thereafter follow the emissions trajectory of the Shared Socioeconomic Pathway

5–8.5 (SSP5-8.5) (O'Neill et al., 2016) until the highest temperatures of each scenario are approached. As the highest temperatures of each scenario are approached, the proportional control gradually steps in to mediate $CO_2$ emissions and



realize the warming trajectory of each scenario. The $CO_2$ emission trajectories derived from the initial experiments were used to drive formal experiments of SWL and OS scenarios (Fig. 1a).

To isolate the contribution of permafrost carbon feedback, two parallel sets of formal experiments were conducted driven by the derived $CO_2$ emission trajectories. One set included the permafrost carbon module, with experiments designated as SWL-1.5, OS-2, SWL-2, OS-3, SWL-3, OS-4, and SWL-4. The other set disabled the permafrost carbon module, referred to as SWL-1.5-np, OS-2-np, SWL-2-np, OS-3-np, SWL-3-np, OS-4-np and SWL-4-np. Comparing these two sets provides a robust framework for assessing the permafrost carbon response and feedback in SWL and OS scenarios. Figure 1b demonstrates that the derived emissions can effectively achieve the designed temperature trajectories for the set of
experiments with permafrost carbon module disabled. For comparison, two parallel experiments with or without permafrost carbon module were also conducted for the high-emissions SSP5-8.5 scenario.

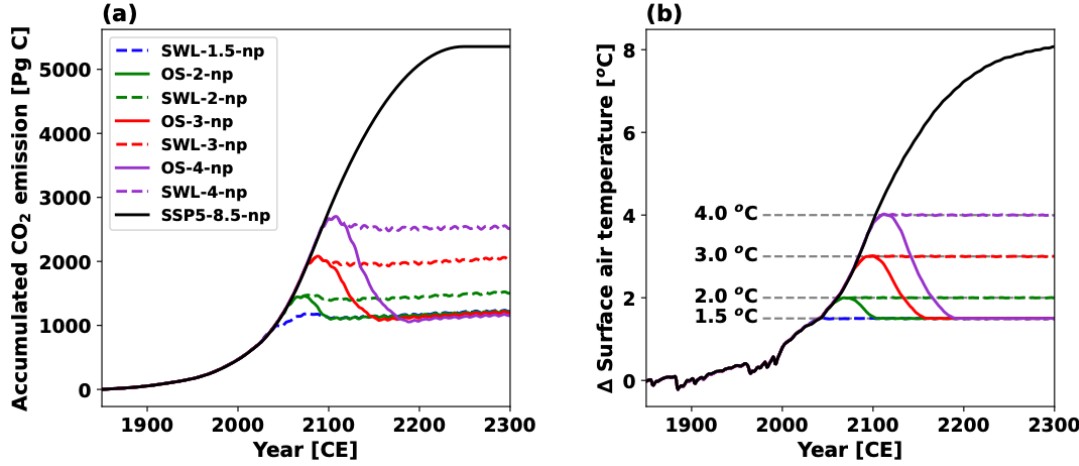

**Figure 1. Temperature stabilization and overshoot scenarios designed through a simple proportional control scheme (Zickfeld et al., 2009) on $CO_2$ emission and UVic ESCM v2.10 with permafrost carbon module disabled. (a) Accumulated $CO_2$ emission and (b)**
**GSAT changes relative to the pre-industrial levels (1850-1900), in the historical and SSP5-8.5 experiments (black), the OS (solid lines) and SWL (dashed lines) experiments at 1.5 °C (blue), 2.0 °C (green), 3.0 °C (red) and 4 °C (purple) GWLs.**

To evaluate the uncertainty in permafrost carbon response under SWL and OS scenarios, we perturbed the four key permafrost carbon parameters following the methodologies of MacDougall and Knutti (2016) and MacDougall (2021). The permafrost carbon decay constant was derived from the mean residence time (MRT) of the slow soil carbon pool at 5 °C in
permafrost soils and adjusted to reflect decay at 25 °C using the method proposed by Kirschbaum (2006). The probability density function (PDF) of MRT was taken as a normal distribution with a mean of 7.45 years and a standard deviation of 2.67 years from Schädel et al. (2014). The available fraction of permafrost carbon was derived from the size of the fast, slow and passive soil organic carbon pools separately for organic, shallow mineral, and deep mineral soils measured by Schädel et al. (2014). The PDF of available fraction of permafrost carbon was weighted gamma distributions, with each distribution
respectively describing the PDF of available fraction of permafrost carbon in organic, shallow mineral, and deep mineral soils. The passive carbon pool transformation rate was estimated from the $^{14}C$ age of the passive carbon pool from





midlatitude soils, yielding an estimated value of $0.25 \times 10^{-10}$ to $4 \times 10^{-10}$ s$^{-1}$ (Trumbore, 2000). The PDF of the passive carbon pool transformation rate was taken as uniform in base-two log space (MacDougall and Knutti, 2016). For the initial quantity of permafrost region soil carbon, its PDF was taken as a normal distribution with a mean of 1035 PgC and a standard deviation of 75 PgC, informed by Hugelius et al. (2014). A series of 5,000-year sensitivity runs were performed under preindustrial steady conditions with varying saturation factors to determine their relationship with the quantity of the permafrost region soil carbon pool. This relationship was then utilized to tune the permafrost region soil carbon pool, ensuring alignment with observational data (Hugelius et al., 2014).

MacDougall and Knutti (2016) and MacDougall (2021) additionally perturbed two physical climate parameters controlling climate sensitivity and Arctic amplification, but they are not perturbed in this study due to their limited influence on GSAT in SWL scenarios. Next, Latin hypercube sampling was utilized to generate 250 parameter sets (constituting 250 model variants), with each parameter sampled from 25 equal-probability intervals, forming a "cube" of 25 parameter sets, repeated 10 times. For each parameter set, the UVic ESCM v2.10 was first run through a 10,000-year spin-up phase under pre-industrial conditions to achieve a quasi-equilibrium state. For these spin-up runs, we set the $CO_2$ concentration to 284.7 ppm and the solar constant to 1360.747 W m$^{-2}$. After completing the spin-up phase, the 250 model variants simulated soil carbon stock in the top 3.35 m of soil in permafrost regions was 1033 [918 to 1146] PgC, with 482 [381 to 583] PgC being perennially frozen soil carbon, which is consistent with observational data (Hugelius et al., 2014). Following the spin-up, emission-driven transient experiments were conducted for SWL, OS and SSP5-8.5 scenarios. Results of experiments are presented as the median of all model variants, with uncertainty estimated between the 5th to the 95th percentiles.

## 3 Results

### 3.1 Permafrost Response

The Northern Hemisphere high-latitude permafrost area, defined as regions where the soil layer remains perennially frozen for at least two consecutive years, is strongly correlated with GSAT changes (Fig. 2a, b). As global warming increases from 1.5 °C to 4 °C, the permafrost area declines from 13.9 million km$^2$ to 8.3 million km$^2$. In the SWL scenarios, when global warming is stabilized at the Paris Agreement targets of 1.5 °C or 2 °C, permafrost degradation can be effectively suppressed compared to the SSP5-8.5 scenario. By 2300, permafrost area decreases by 4.5 [4.4 to 4.7] million km$^2$ and 6.5 [6.4 to 6.8] million km$^2$ relative to the pre-industrial period under SWL-1.5 and SWL-2 scenarios, respectively, accounting for 39 [38 to 40] % and 56 [54 to 58] % of the reduction observed under SSP5-8.5 scenario. The incremental permafrost degradation under the SWL-3 compared to SWL-2 is significantly larger than that under SWL-4 compared to SWL-3. This is mainly because the remaining permafrost available for degradation becomes progressively limited under higher GWLs, and the simulated permafrost area under higher SWL scenarios has not yet reached a steady state in our experiments. Additionally, the permafrost area under the SWL-3 and SWL-4 scenarios exceeds that under the SSP5-8.5 scenario by only 1.9 [1.8 to 2.0] million km$^2$ and 0.9 [0.8 to 1.0] million km$^2$ by 2300, respectively. This is primarily because





the permafrost area in the SSP5-8.5 scenario is smaller and, as a transient experiment, is further from equilibrium compared

to SWL-3 and SWL-4. However, during the cooling phase of the OS scenarios, as the GSAT returns to 1.5 °C above pre-industrial levels, the permafrost area gradually recovers. By 2300, it converges to similar levels of 11.3~12.3 million km$^2$ under the OS-2, OS-3 and OS-4 scenarios, with an additional loss of only 0.3~1.1 million km$^2$ compared to the SWL-1.5 scenario (Fig. 3a). This indicates that permafrost area is reversible and largely follows the GSAT trajectory, recovering as temperature reduction, consistent with previous studies (MacDougall, 2013; Lee et al., 2021; Schwinger et al., 2022). It is

worth mentioning that permafrost area loss under OS scenarios typically shows a hysteresis effect (Boucher et al., 2012; MacDougall, 2013; Eliseev et al., 2014), peaking about 10~30 years after GSAT reaches its maximum in our experiments.

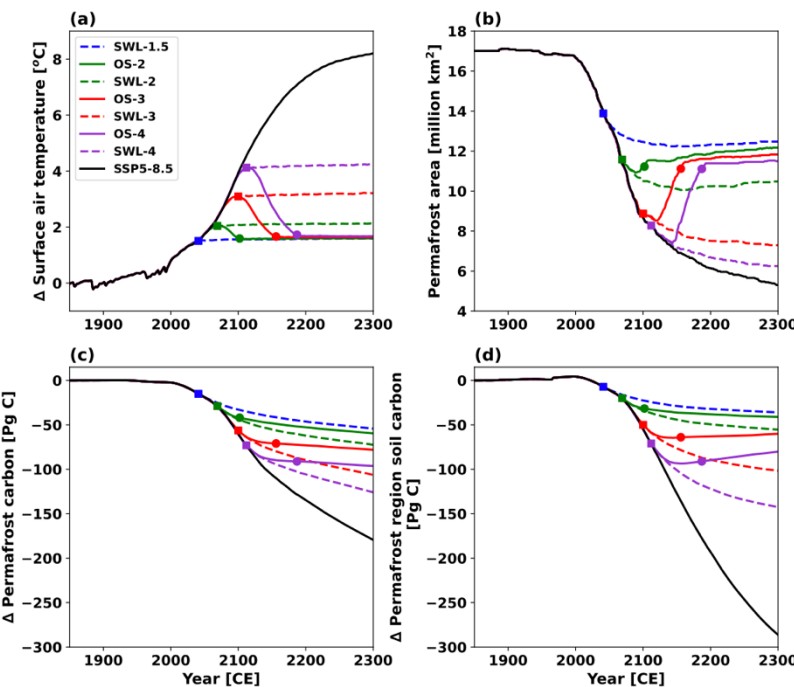

**Figure 2. Timeseries of (a) GSAT changes, (b) permafrost area, (c) permafrost carbon changes and (d) permafrost region soil carbon changes under OS (solid lines) and SWL (dashed lines) scenarios at 1.5°C (blue), 2.0°C (green), 3.0°C (red), and 4°C**

**(purple) GWLs, as well as the SSP5-8.5 scenario. Square markers indicate the time points when the temperature overshoot reaches its peak or stabilized warming begins, while circle markers indicate when the overshoot returns to 1.5°C. All changes are relative to the pre-industrial period (1850-1900). All results represent the median values from 250 ensemble experiments.**

Under both temperature stabilization and overshoot scenarios, permafrost carbon declines monotonically over time (Fig. 2c), driven by the imbalance between weaker permafrost carbon inputs and relative faster decomposition. The primary

source of permafrost carbon input is a very slow physical process of downward diffusion through the permafrost table due to cryoturbation mixing effect (MacDougall and Knutti, 2016). As a result, the rates of permafrost carbon decomposition significantly exceed its input (Fig. 4a, c). Under the SWL-1.5, SWL-2, SWL-3, and SWL-4 scenarios, permafrost carbon losses are projected to be 54 [32 to 79], 72 [42 to 104], 106 [64 to 151], and 126 [74 to 180] PgC by 2300, respectively, while under the OS-2, OS-3, and OS-4 scenarios, the losses are 60 [35 to 86], 78 [50 to 111], and 96 [63 to 135] PgC. This



indicates that despite GSAT returning to 1.5 °C in OS scenarios, considerable permafrost carbon losses still occur, accounting for 82 [80 to 85] %, 73 [71 to 78] %, and 76 [72 to 85] % of the losses in the corresponding SWL scenarios achieving same GWLs. Under the SSP5-8.5 scenario, permafrost carbon inputs show a decreasing trend in the last 140 years (Fig. 4a), which is likely due to less soil carbon transported downward into the permafrost carbon pool as the permafrost area decreases. Additionally, the extra permafrost carbon losses under the SWL-2, SWL-3, and SWL-4 scenarios relative to the

SWL-1.5 scenario continue to increase over time, whereas in the OS scenarios, these additional losses decrease during the stabilization phase (Fig. 3b). This decrease occurs because, after global warming cools to 1.5 °C, the permafrost carbon decomposition rate under the OS scenarios closely aligns with that of the SWL-1.5 scenario, while the permafrost carbon inputs under the OS scenarios surpass that of the SWL-1.5 scenario (Fig. 4a, c).

Permafrost region soil carbon shows a strong tendency to recovery after temperature overshoot. Meanwhile, permafrost

region soil carbon continues to decrease under temperature stabilization scenarios, but the rate of decrease is gradually slowing down. Although permafrost region soil carbon in the OS scenarios does not recover to that under the SWL-1.5 scenario, its release is significantly mitigated compared to SWL scenarios at same GWLs (Fig. 2d). By 2300, permafrost region soil carbon losses under OS-2, OS-3, and OS-4 scenarios are projected to be 41 [15 to 72], 60 [29 to 98], and 80 [43 to 123] PgC, respectively, with reductions of 14 [8 to 21], 41 [25 to 57], and 62 [41 to 82] PgC compared to the SWL-2,

SWL-3, and SWL-4 scenarios, respectively. Permafrost region soil carbon inputs primarily originate from robust biophysical processes related to vegetation litterfall, with their intensity influenced by warming levels and $CO_2$ fertilization effects, while permafrost region soil carbon decomposition is closely tied to GSAT. In the OS scenarios, the peak for permafrost region soil carbon inputs occurs slightly later than the GSAT peak, whereas the peak for permafrost region soil carbon decomposition happens marginally earlier than the GSAT peak (Fig. 4b, d). During the stabilization phase in OS-2, OS-3,

and OS-4 scenarios, additional permafrost region soil carbon losses compared to the SWL-1.5 scenario decrease (Fig. 3c). Notably, the permafrost region soil potentially turns into a carbon sink for atmospheric $CO_2$ during the stabilization phase in OS-3 and OS-4 (Fig. 2d), due to the soil carbon inputs surpass the reduced decomposition activity as a result of the depletion of permafrost region soil carbon stocks and reduced warming levels. However, the permafrost region soil serves as a net carbon source for atmospheric $CO_2$ for all OS and SWL scenarios simulated in this study.

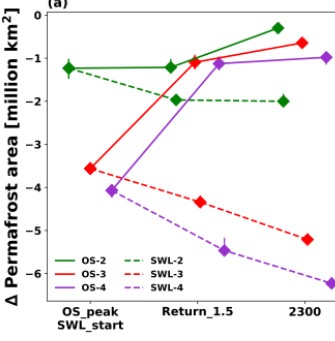
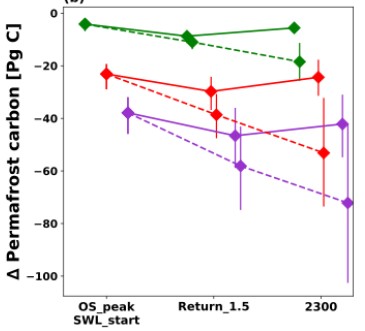
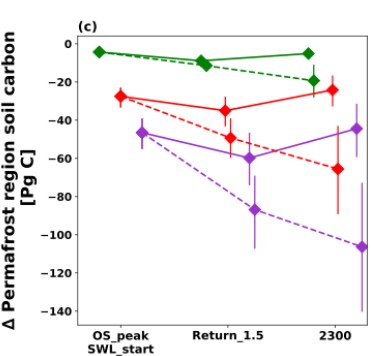


**Figure 3.** Changes in (a) permafrost area, (b) permafrost carbon and (c) permafrost region soil carbon in the OS and SWL scenarios at 2.0 °C (green), 3.0 °C (red) and 4 °C (purple) GWLs relative to SWL-1.5 scenario. The horizontal axis represents three distinct time points: the peak of overshoot (OS_peak) or the commencement of stabilized warming (SWL_start), the return of overshoot to 1.5°C (Return_1.5), and the year 2300. The uncertainty bars represent the 5th to 95th percentiles of 250 ensemble
experiments.

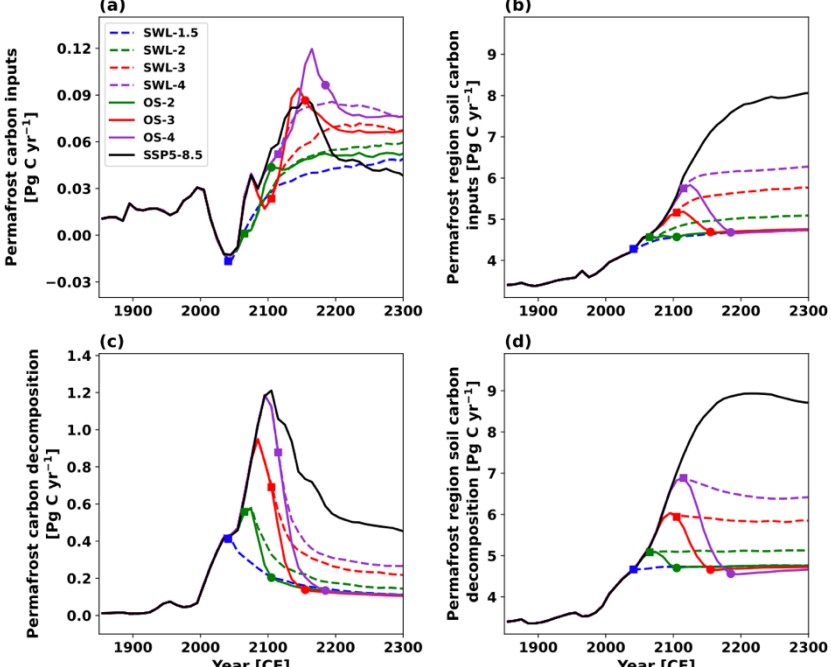

**Figure 4.** Timeseries of changes in (a) permafrost carbon inputs, (b) permafrost region soil carbon inputs, (c) permafrost carbon decomposition and (d) permafrost region soil carbon decomposition, under the OS (solid lines) and SWL (dashed lines) scenarios at 1.5 °C (blue), 2.0 °C (green), 3.0 °C (red) and 4 °C (purple) GWLs, along with the SSP5-8.5 scenario (black). Square markers
indicate the time points when the temperature overshoot reaches its peak or stabilized warming begins, while circle markers indicate when the overshoot returns to 1.5°C. All results represent the median values from 250 ensemble experiments.

### 3.2 Radiative Impacts of Permafrost Carbon Release

The permafrost carbon release would increase global mean radiative forcing and surface temperature. The time evolution of additional surface warming closely resembles the additional radiative forcing (Fig. 5a, b) due to approximately
linear relationship between radiative forcing and temperature change based on the energy balance of the climate system (Forster et al., 1997; Myhre et al., 2014). Both in the temperature stabilization and overshoot scenarios, the magnitude of additional radiative forcing and warming increases with higher GWLs. By the year 2300, the additional warming under the overshoot scenarios steadily rises from 0.10 [0.06 to 0.15] °C in OS-2 to 0.14 [0.09 to 0.20] °C in OS-3 and 0.18 [0.11 to 0.25] °C in OS-4. Similarly, in the temperature stabilization scenarios, the additional warming increases from 0.10 [0.06 to
0.14] °C in SWL-1.5 to 0.13 [0.07 to 0.19] °C in SWL-2, 0.21 [0.12 to 0.31] °C in SWL-3, and 0.24 [0.15 to 0.35] °C in SWL-4. This is because higher GWLs lead to more significant reductions in permafrost carbon and permafrost region soil carbon (Fig. 2c, d), and further intensifying global warming. Furthermore, in SWL scenarios, the additional warming





continues to increase over time due to delayed degradation and positive permafrost carbon-climate feedback. In OS scenarios, the additional warming tends to stabilize once the temperature returns to 1.5 °C above pre-industrial levels (Fig.

5b). By 2300, the additional warming under SWL-2, SWL-3, and SWL-4 exceeds that of OS-2, OS-3, and OS-4 by 0.03 [0.01 to 0.04] °C, 0.07 [0.02 to 0.11] °C and 0.07 [0.03 to 0.10] °C, respectively, amplifying the additional warming in the overshoot scenarios by 22 % to 56 %.

However, the additional warming during the cooling phase is most substantial in OS scenarios, and it is also significantly greater than that in SWL scenarios over the same period. This is primarily due to the sustained reduction in

atmospheric $CO_2$ concentration during the cooling phase, which amplifies the radiative forcing caused by permafrost carbon release. Specifically, because of the logarithmic relationship between $CO_2$ concentration and radiative forcing (Etminan et al., 2016), the decline of background $CO_2$ concentration to low levels causes the additional increases in $CO_2$ concentration due to permafrost carbon release to produce more significant changes in radiative forcing. Similarly, the global mean warming under the SSP5-8.5 scenario is 2.43 [2.42 to 2.44] °C higher than under the SWL-1.5 scenario by 2100, but the

additional warming difference due to permafrost carbon release is minimal at 0.01 [0 to 0.03] °C. Despite global mean warming reaching 8.20 [8.13 to 8.26] °C by 2300 in the SSP5-8.5 scenario (Fig. 2a), the additional warming is limited to 0.14 [0.08 to 0.20] °C due to the significantly higher background $CO_2$ concentration, the additional warming is only comparable to that of SWL-2 and OS-3 scenarios.

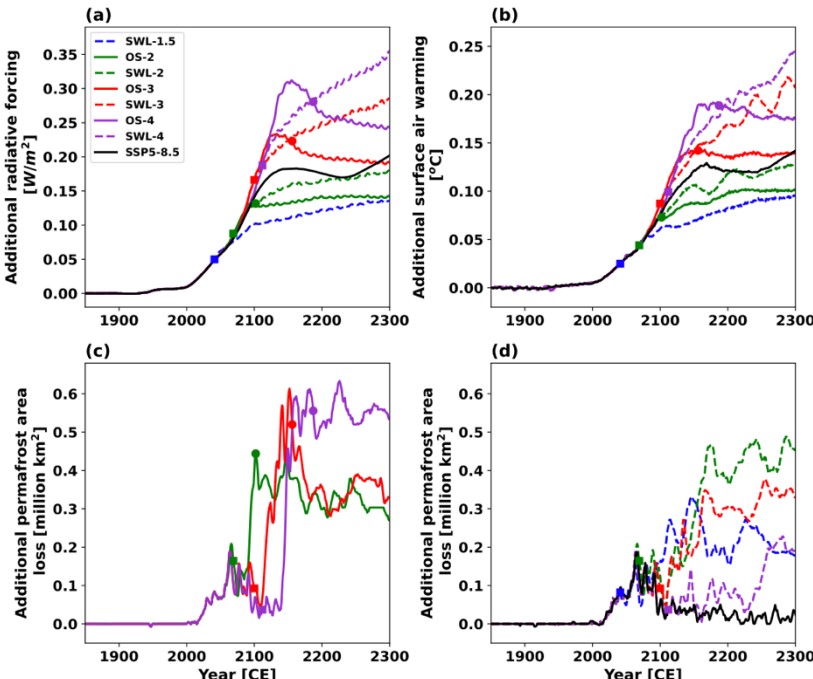

**Figure 5. Additional changes in (a) radiative forcing, (b) global mean surface air warming and (c, d) permafrost area due to permafrost carbon-climate feedback in the OS (solid lines) and SWL (dashed lines) scenarios at 1.5 °C (blue), 2.0 °C (green), 3.0 °C (red) and 4 °C (purple) GWLs, along with the SSP5-8.5 scenario (black). Square markers indicate the time points when the temperature overshoot reaches its peak or stabilized warming begins, while circle markers indicate when the overshoot returns to**





**1.5 °C. All results represent the median values from 250 ensemble experiments. In panel (a), the additional radiative forcing is**
**calculated using the simplified expressions (Etminan et al., 2016) based on simulated CO₂ concentrations. In panels (c) and (d), the**
**additional permafrost area loss is smoothed using a 5-year rolling average to eliminate shorter interannual variability.**

The permafrost carbon-climate feedback also causes additional permafrost area loss under OS, SWL and SSP5-8.5
scenarios. During the cooling phase of OS-2, OS-3, and OS-4, there is a significantly additional permafrost area loss of 0.3
[0.1 to 0.4], 0.4 [0.2 to 0.6], and 0.5 [0.3 to 0.7] million $km^2$ respectively (Fig. 5c), contributing 6 [2 to 8] %, 8 [4 to 12] %
and 9 [5 to 12] % of total permafrost degradation by 2300. Although global warming has been identified as the primary
driver of permafrost decline (McGuire et al., 2016; Liu et al., 2021), the temporal evolution of additional permafrost area
loss does not align with changes in the additional warming (Fig. 5b), particularly in the SWL and SSP5-8.5 scenarios (Fig.
5d). To better understand this puzzle, we conducted sensitivity analysis based on the SSP5-8.5 scenario. The results show the
sensitivity of permafrost area to global warming (SPAW) is not constant, the maximal sensitivity of 3.2 million $km^2$/°C
occurs around 1.5~2 °C GWLs (Fig. 6a). The magnitude of SPAW decreases as global surface warming from 2 °C to 4 °C,
suggesting the response of permafrost area to rising temperature weakens and a diminishing feedback effect. Interestingly,
by multiplying transient SPAW by the additional warming, the temporal evolution of the additional permafrost area loss can
be well reconstructed across various scenarios (Fig. 6b-i). The root mean square error (RMSE) between original and
reconstructed additional permafrost area loss, ranging from 0.03 to 0.07 million $km^2$, indicates high reconstruction accuracy.
This further confirms that the additional permafrost area loss induced by permafrost carbon-climate feedback can be
explained by the combined effects of the additional warming and the decrease in SPAW.

Permafrost carbon-climate feedback induces greater additional warming at higher GWLs, but the lower SPAW
diminishes its impact on permafrost area loss. Under the SSP5-8.5 scenario, the additional permafrost area loss peaks around
the 2060s, reaching approximately 0.19 million $km^2$, coinciding with a strong sensitivity of 3.0 million $km^2$/°C. After this
peak, despite continued rise in additional warming, the permafrost area loss declines notably due to the gradual weakening of
SPAW. Similarly, in the SWL scenario, SPAW decreases markedly at higher GWLs, even as additional warming increases,
preventing a positive correlation between permafrost area loss and GWLs (Fig. 5d). For example, the additional warming in
the SWL-4 scenario reaches 0.24 [0.15 to 0.35] °C by 2300, nearly twice that of the SWL-2 scenario (0.13 [0.07 to 0.19]
°C). However, due to its SPAW being only 1.1 million $km^2$/°C, less than half of 3.1 million $km^2$/°C under SWL-2, the
additional permafrost area loss under SWL-4 scenario is only 0.2 [0.1 to 0.3] million $km^2$, significantly lower than 0.5 [0.2 to
0.7] million $km^2$ under SWL-2. This suggests the maximal SPAW between 1.5~2 °C GWLs has significant implications for
the Paris Agreement's targets of limiting global warming at the same levels.





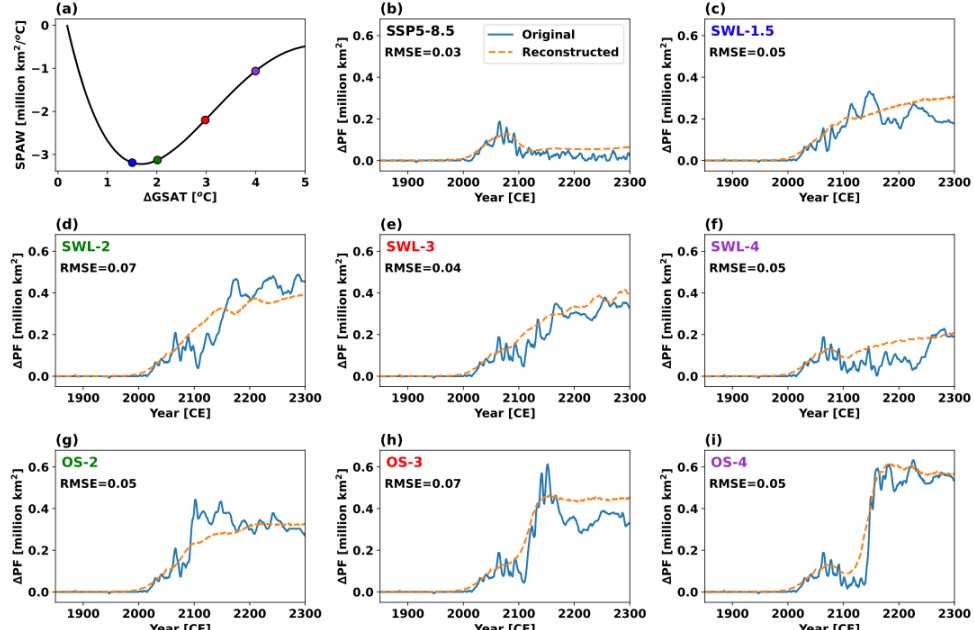

**Figure 6. (a) Relationship between SPAW and GSAT change derived from the SSP5-8.5 scenario. Black curve represents SPAW calculated as the slope of a polynomial regression between permafrost area and GSAT change under the SSP5-8.5 scenario. Colored circle markers represent transient SPAW at 1.5°C (blue), 2°C (green), 3°C (red), and 4°C (purple) GWLs. (b-i) Time series of original (solid line) and reconstructed (dashed line) additional permafrost area loss (ΔPF) due to permafrost carbon-climate feedback for each scenario. The constructed results are derived by multiplying the transient SPAW with the additional warming (Fig. 5b) under each scenario. Smaller RMSE values indicate higher reconstruction accuracy.**

To evaluate the relative importance of perturbed model parameters on permafrost region soil carbon release under different temperature pathways, we calculated the correlation between these parameters and permafrost region soil carbon losses by 2300, across the SSP5-8.5, OS4, and SWL4 scenarios. In these scenarios, the influence of model parameters on the uncertainty of permafrost carbon losses is relatively consistent, with the strongest correlations observed for the permafrost passive carbon pool transformation rate (R=0.81~0.85), followed by the initial quantity of permafrost region soil carbon (R=0.55~0.61). This finding aligns with Ji et al. (2024), which highlighted the critical role of these two parameters in the uncertainty of permafrost region soil carbon losses under temperature overshoot and 1.5 °C warming stabilization scenarios.

## 4 Conclusions and Discussion

This study utilizes the intermediate-complexity Earth system model UVic ESCM v2.10 and perturbed parameter ensemble modelling approach to study the permafrost response and feedback under temperature stabilization and overshoot scenarios across different GWLs. UVic ESCM can realistically reproduce historical permafrost area and permafrost carbon stocks. In addition to demonstrating the changes of permafrost area, permafrost carbon and permafrost region soil carbon under various warming trajectories, the findings also quantify the additional radiative forcing, global mean surface air



warming and permafrost area degradation induced by permafrost carbon release, providing insights into the implications of
permafrost carbon-climate feedback for long-term climate change and mitigation strategies.

Permafrost area reduction shows a strong correlation with global warming under both SWL and OS scenarios, and the maximal sensitivity of permafrost area to global warming (SPAW) falls within the 1.5~2 °C GWLs. In SWL scenarios, lower GWLs effectively mitigate permafrost area reduction compared to the SSP5-8.5 scenario, whereas higher GWLs lead to substantial reductions in permafrost area due to cumulative warming effects. In OS scenarios, permafrost area largely
recovers as global warming returns to 1.5 °C levels, though this recovery is delayed by hysteresis effects, with degradation persisting for decades after temperature peaks. The maximal SPAW derived from UVic ESCM simulation of SSP5-8.5 scenario is 3.2 million $km^2/°C$, which is much higher than 1.6 million $km^2/°C$ derived through an equilibrium permafrost model (Liu et al., 2021), but our result is close to an observation-constrained equilibrium projection of SPAW, approximately 3.5 million $km^2/°C$ (Nitzbon et al., 2024). Nitzbon et al. (2024) also noted that permafrost area decreases
quasi-linearly with increasing GSAT, especially below 4 °C GWL. However, we found the transient SPAW is not a constant based on the SSP5-8.5 scenario and is a function of global warming with an upward-opening parabolic shape. The maximal SPAW occurs at approximately 1.5~2 °C GWL, suggesting the fastest permafrost degradation is anticipated to take place within Paris Agreement's warming levels. This is in line with Comyn-Platt et al. (2018), who found the feedback processes due to permafrost thaw respond more quickly at temperatures below 1.5 °C, and their differences between the 1.5 and 2 °C
targets are rather small. Our findings have significant implications for the development of mitigation and adaptation strategies addressing permafrost-thaw impacts consistent with keeping global warming at the Paris Agreement's levels.

Permafrost carbon declines under both SWL and OS scenarios, driven by the dynamic balance between soil carbon inputs and decomposition. Despite significant carbon losses persisting during the cooling and stabilization phases of OS scenarios, highlighting the essentially irreversible nature of this process. OS scenarios partially mitigate permafrost carbon
losses compared to SWL scenarios at the same GWLs. Furthermore, permafrost region soil carbon exhibits a certain degree of recovery under OS scenarios. In fact, soil in these regions even transitions into a carbon sink during the stabilization phase of OS scenarios with high GWLs, supported by reduced decomposition rates and sustained inputs from vegetation litterfall. However, the higher the overshoot levels, the less recovery there is. These findings underscore the critical role of temporary temperature overshooting levels in affecting long-term permafrost carbon release and recovery potential.

Our study highlights substantial permafrost carbon-climate feedback during the cooling phase of OS scenario. Permafrost carbon release significantly increases global radiative forcing and amplify global warming, permafrost carbon-climate feedback can be more profound in temperature stabilization and overshoot scenarios than in high-emissions scenarios. In SWL scenarios, additional radiative forcing and warming persistently increase over time due to delayed degradation and positive permafrost carbon-climate feedback. In contrast, in OS scenarios, additional warming almost
stabilizes once global warming drops to 1.5 °C levels. During the cooling phase of OS scenarios, lower background $CO_2$ concentrations amplify the warming effect of permafrost carbon release. In contrast, under the high-emissions SSP5-8.5 scenario, additional warming is limited due to higher background $CO_2$ levels reducing the additional radiative forcing from



permafrost carbon release. The additional permafrost area losses due to permafrost carbon-climate feedback range from 5~11 % of the total permafrost area losses in the scenarios explored in this study. The additional permafrost area losses can be well

explained by the SPAW and the additional warming. The complex interactions between global warming and permafrost degradation emphasize the importance of accounting for these nonlinear effects in climate projections, particularly at 1.5~2 °C global warming levels.

This study aims to serve as a meaningful supplement to the limited existing research on permafrost response and feedback under temperature stabilization and overshoot scenarios. For example, CMIP6 includes only one overshoot

scenario (SSP5-3.4-OS), which is insufficient for comprehensively analyzing the potential contribution of permafrost feedback under different GWLs (Melnikova et al., 2021). The upcoming CMIP7 is planned to encompass a broader range of overshoot scenarios, from very low to medium and high overshoot scenarios. This expansion aims to delve into the potential for climate restoration, the feasibility of achieving Paris Agreement targets, and the risks of irreversibility and hysteresis in the slower components of the Earth system from beyond 2125 to 2300 (WRCP, 2024). Our study investigated the permafrost

response and feedback under overshoot scenarios with varying levels of warming, which can be used for comparative analysis of the upcoming CMIP7 multi-model experimental results. Additionally, research on permafrost changes under different climate stabilization scenarios is also limited. Some of existing studies are based on equilibrium permafrost models (e.g. Liu et al., 2021), which do not consider transient effects and cannot predict the timing of permafrost loss based on imposed warming scenarios (Smith et al., 2022). Our study utilizes a process-based model and provides insights into the

nonlinear relationship between permafrost degradation and global warming, such as varying SPAW under different global warming levels.

This study demonstrates a method to quantify the permafrost carbon-climate feedback under specified global warming levels or warming trajectories with climate-carbon cycle fully coupled Earth system model driven by pre-designed $CO_2$ emission pathway. As permafrost carbon-climate feedback is highly sensitive to permafrost frozen states and climate

conditions, controlling temperature or warming trajectory is beneficial to isolate individual contributions from climate condition and permafrost carbon processes, especially for multi-model intercomparison study. Due to tight link between terrestrial carbon feedback and physical climate feedback, Goodwin (2019) found around half the uncertainty in derived terrestrial carbon feedback originates from uncertainty in the physical climate feedback. Therefore, controlling global warming levels or warming trajectory might be helpful to accurately assess a specific aspect of terrestrial carbon feedback by

isolating the uncertainty from the physical climate feedback. We encourage the exploration of potential application for this method in quantifying other aspects of carbon-climate feedbacks. Of course, the findings of this study are based on a single Earth system model, and it motivates further studies aimed at comprehensively understanding of permafrost response and feedback and their impacts on Arctic communities, infrastructure, and climate policies.



**Data availability**

The source code of the UVic ESCM version 2.10 can be downloaded from https://terra.seos.uvic.ca/model/2.10/. All experimental simulation data utilized in this study are publicly accessible and can be obtained from Cui. (2024).

**Author contributions**

DJ designed the research. MC performed the model simulations, analyzed and interpreted the model data, and wrote the initial manuscript. DJ, MC and YC revised the manuscript.

**Competing interests**


The authors declare that they have no conflict of interest.

**Acknowledgements**

The authors thank the Super Computing Center of Beijing Normal University for providing computing resources.

**Financial support**

This research has been supported by the National Natural Science Foundation of China (No. 41875126).

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
