# Peer review of "Permafrost response and feedback under temperature stabilization and overshoot scenarios with different global warming levels"

_EGUsphere, 2024_

## Author Comment (AC1)

We express our sincere gratitude to your insightful and constructive comments. Please find below our point-by-point replies. All the comments are presented in black text and the corresponding replies are highlighted in blue.

The authors use a fully coupled climate model to evaluate the response of permafrost under temperature stabilization and overshoot scenarios. The methods appear rigorous, and the manuscript is well-written. However, the manuscript could be improved by discussing the implications of some of the feedbacks being modeled. Substantial revisions to some sections of the text and figures could further improve the overall clarity of the manuscript and link it more directly to existing literature.

Major comments:

Line 92 and throughout: Yedoma represents a significant, deep proportion of the permafrost carbon stock in some regions and was formed over extremely long timescales. Given the focus on differences between overshoot and stabilization scenarios, a greater discussion of this limitation could be included in the methods and conclusions.

We appreciate your insightful comment. The UVic ESCM v2.10 simulates permafrost carbon only in the top six layers (to a depth of 3.35 m), and therefore omits soil carbon stored in the deep deposits of Yedoma regions. As a result, we cannot directly estimate the impacts of temperature overshoot on deep Yedoma carbon, or compare these changes relative to stabilization scenarios. To address this limitation, we analyzed the average and maximum active layer thickness (ALT) in Yedoma regions between the overshoot and stabilization scenarios simulated in this study. Using differences in ALT as a proxy to infer the potential impacts on deep Yedoma carbon (Figure R1). To clarify this point, we have added the following paragraph to Section "4 Conclusions and Discussion":

"Yedoma deposits represent a significant deep carbon reservoir and are widespread across Siberia, Alaska, and the Yukon region of Canada, having primarily formed during the late Pleistocene, especially in the late glacial period. These deep, perennially frozen sediments are particularly ice-rich, and the freeze-locked organic matter in such deposits can be re-mobilized on short time-scales, representing one of the most vulnerable permafrost carbon pools under future warming scenarios (Schuur et al., 2015; Strauss et al., 2017). According to Zimov et al. (2006), these perennially frozen Yedoma sediments cover more than 1 million $km^2$, with an average depth of approximately 25 m. Recent estimates place the organic carbon stock in Yedoma deposits at 213 ± 24 PgC, constituting a significant portion of the total permafrost carbon pool (Strauss et al., 2017). However, the UVic ESCM v2.10 utilized in this study simulates permafrost carbon only within the top 3.35 m of soil, limiting our ability to directly assess the impacts of temperature overshoot on deep Yedoma carbon. Considering their ice-rich nature and potential susceptibility to rapid-thaw processes, we analyzed the average and maximum active layer thickness (ALT) in Yedoma regions (Strauss et al., 2021, 2022) under the simulated scenarios to approximate potential impacts. We find that the average ALT in Yedoma regions remains below 1 m in all stabilization and overshoot scenarios, while the maximum ALT rarely exceeds 3.35 m in overshoot scenarios but does exceed this depth in some stabilization scenarios. However, in all scenarios, the maximum ALT does not

exceed 6 m, which is relatively shallow compared to the average depth (~25 m) of Yedoma deposits. Therefore, the impact of overshoot scenarios on deep Yedoma carbon is likely minor relative to stabilization scenarios."

[Figure]

[Figure]

Figure R1. Timeseries of annual (a) average and (b) maximum active layer thickness (ALT) in Yedoma regions under overshoot (colored solid lines) and stabilization (colored dashed lines) scenarios at 1.5 °C (blue), 2.0 °C (green), 3.0 °C (red), and 4 °C (purple) GWLs, as well as the SSP5-8.5 scenario (black solid line). Results represent the ensemble median of 250 simulations. The Yedoma region mask used in this analysis is based on the Ice-Rich Yedoma Permafrost Database Version 2 (IRYP v2) (Strauss et al., 2021, 2022; https://doi.org/10.1594/PANGAEA.940078).

Line 90 – 103: More background on the UVICC ESM permafrost carbon model, validation, and perturbed parameter approach would be particularly useful to readers.

Thank you for your valuable suggestion. To improve clarity and provide a more comprehensive background, we have made the following revisions to this section:

"The UVic ESCM model represents the terrestrial subsurface with 14 layers, extending to a total depth of 250.3 m to correctly capture the transient response of permafrost on centennial timescales. The top eight layers (10.0 m) are involved in the hydraulic cycle, while the deeper layers are modeled as impermeable bedrock (Avis et al., 2011). The carbon cycle is active in the top six layers (3.35 m), where organic carbon from litterfall, simulated by the TRIFFID vegetation model, is allocated to soil layers with temperatures above 1 °C according to an exponentially decreasing function with depth. If all soil layers are below 1 °C, the organic carbon is added to the top soil layer. The soil respiration is calculated for each layer individually as a function of temperature and moisture, but the respiration ceases when the soil layer temperature falls below 0 °C (Meissner et al., 2003; Mengis et al., 2020). In regions where permafrost exists—defined as areas where soil temperature remains below 0°C for at least two consecutive years—the model applies a revised diffusion-based cryoturbation scheme to redistribute soil carbon within the soil column. The permafrost carbon is generated when soil carbon is transported across the permafrost table. Compared to the original diffusion-based cryoturbation scheme proposed by Koven et al. (2009), the revised cryoturbation scheme calculates carbon diffusion using an effective carbon

concentration that incorporates the volumetric porosity of the soil layer, rather than the actual carbon concentration, thereby resolving the disequilibrium problem of the permafrost carbon pool during model spin-up (MacDougall and Knutti, 2016). Notably, the UVic ESCM v2.10 does not include a methane production module, meaning that soil carbon emissions to the atmosphere are limited to $CO_2$. A more detailed description of the UVic ESCM's permafrost carbon parameterization scheme and its simulated permafrost carbon characteristics can be found in MacDougall and Knutti (2016)."

Model validation has been incorporated into the first paragraph in Section "3 Results". The added content is:

"The UVic ESCM v2.10 reliably simulates historical temperature changes, permafrost area, and the partitioning of anthropogenic carbon emissions among the atmosphere, ocean and land. During the period 2011–2020, the model estimated a GSAT increase of 1.14 [1.13 to 1.15] °C relative to preindustrial levels, which is closely aligned with the observed rise of 1.09 [0.91 to 1.23] °C (Gulev et al., 2021). For the period 1960–1990, the model simulated the northern hemisphere permafrost area at 16.8 [16.7 to 16.9] million km$^2$, which falls within the reconstructed range of 12.0 to 18.2 million km$^2$ (Chadburn et al., 2017) and the observation derived extent of 12.21 to 16.98 million km$^2$ (Zhang et al., 2000). Additionally, the modeled soil carbon stock in the top 3.35 m of permafrost regions for this same period was 1034 [919 to 1151] PgC, with 483 [382 to 587] PgC classified as perennially frozen carbon, accounting for 47% [42% to 51%] of the total permafrost soil carbon stock, in agreement with Hugelius et al. (2014). From 2010 to 2019, the model estimated that anthropogenic carbon emissions were distributed as follows: 5.5 [5.4 to 5.6] PgC yr$^{-1}$ to the atmosphere, 3.0 [2.98 to 3.03] PgC yr$^{-1}$ to the ocean, and 2.5 [2.4 to 2.6] PgC yr$^{-1}$ to terrestrial ecosystems. These estimates are broadly consistent with the global anthropogenic $CO_2$ budget assessment by the Global Carbon Project (GCP) with figures of 5.1±0.02 PgC yr$^{-1}$ for the atmosphere, 2.5±0.6 PgC yr$^{-1}$ for the ocean, and 3.4±0.9 PgC yr$^{-1}$ for terrestrial ecosystems (Friedlingstein et al., 2020)."

The perturbed parameter approach is described at the end of Section "2.2 Experimental design". We have refined the description to improve coherence. Additionally, we have added a figure to illustrate the probability distribution functions of the four perturbed key permafrost carbon parameters (Figure 2). The revised text and added figure are as follows:

"The Latin hypercube sampling method (McKay et al., 1979) was used to explore the effects of parameter uncertainty on projections of permafrost carbon change. In this study, the probability distribution function of each key permafrost carbon parameter was divided into 25 intervals of equal probability. One value was randomly selected from each interval for a given parameter, and then randomly matched with values of the other three key parameters selected in the same manner to generate parameter sets. This sampling procedure was repeated 10 times, resulting in 250 unique parameter sets (i.e., 250 model variants). For each parameter set, the UVic ESCM v2.10 was first run through a 10,000-year spin-up phase under pre-industrial conditions to achieve a quasi-equilibrium state. For these spin-up runs, the atmospheric $CO_2$ concentration was fixed at 284.7 ppm and the solar constant was set to 1360.747 W m$^{-2}$. Following the spin-up, emission-driven transient experiments were conducted under the stabilization, overshoot and

SSP5-8.5 scenarios. The results are presented as the median across all model variants, with uncertainty quantified as the range between the 5th to the 95th percentiles."

[Figure]

Figure 2. Probability distribution functions of the four key permafrost carbon parameters perturbed in the UVic ESCM v2.10 to represent uncertainty in permafrost carbon response. Panel (d) employs a logarithmic scale on the horizontal axis to better illustrate the distribution of the corresponding parameter. This figure is reproduced from MacDougall (2021).

Line 129 – 136: More clarity is needed about this aspect of the method and the interaction with any permafrost feedback loops. It appears these experiments were done to create drivers for the overshoot scenarios (i.e. the proportional control scheme is not active when the final model runs for analysis are done). It then appears based on this text and the text in section 3.2 that any permafrost carbon fluxes would be tacked onto the emissions and removals needed to accomplish these scenarios. Significant edits are needed here for clarity. The additional warming in Figure 5 also appears well-suited for additional discussion.

Thank you for your detailed comments. Actually, the proportional control scheme was used only in the initial set of simulations, which were conducted to generate $CO_2$ emission trajectories that follow the intended warming pathways of each scenario (Fig. 1a). These emission trajectories were then used to drive the formal experiments for both stabilization and overshoot scenarios, without any further application of the proportional control scheme during the final model integrations.

To isolate the contribution of permafrost carbon feedback, we conducted two parallel sets of formal experiments for each scenario, both driven by the same $CO_2$ emission trajectories. The only difference between these sets lies in whether the permafrost carbon module was activated. As you noted, any permafrost carbon fluxes would be tacked onto the emissions and removals needed to accomplish these scenarios. To avoid confusion between scenario names and experiment types, we have introduced distinct suffixes to denote the inclusion or exclusion of permafrost carbon feedback. Specifically, the suffix "-pc" refers to experiments with the permafrost carbon module enabled, while "-npc" refers to those with the module disabled.

To improve clarity, we have revised the relevant section as follows:

"To isolate the contribution of permafrost carbon feedback, two parallel sets of formal experiments were conducted for each scenario, both driven by the same $CO_2$ emission trajectories. One set included the permafrost carbon module, with experiments designated as SWL-1.5-pc, OS-2-pc, SWL-2-pc, OS-3-pc, SWL-3-pc, OS-4-pc, and SWL-4-pc. The other set disabled the permafrost carbon module and is referred to as SWL-1.5-npc, OS-2-npc, SWL-2-npc, OS-3-npc, SWL-3-npc, OS-4-npc, and SWL-4-npc. Notably, the emission trajectories used in these experiments were derived from initial simulations in which the proportional control scheme was applied to achieve the desired temperature pathways. Once these emission trajectories were established, they were used to drive the formal experiments without further application of the proportional control scheme. Since the emissions trajectories were generated using the UVic ESCM v2.10 without the permafrost carbon module, applying them in simulations with the permafrost module enabled results in any permafrost carbon fluxes being effectively added on top of the pre-derived emissions, thereby causing additional warming. In other words, to achieve the intended climate targets under the same emission pathways, removals equivalent to the permafrost carbon emissions would be required. Therefore, the comparison between these two experiment sets provides a robust framework for quantifying the permafrost carbon feedback under stabilization and overshoot scenarios."

We have also revised the first paragraph of Section 3.2 on radiative impacts of permafrost carbon release to better clarify the expression of additional warming. The revised text is as follows:

"The permafrost carbon release would increase global mean radiative forcing and surface temperature. By comparing two parallel sets of experiments with and without the permafrost carbon module, we were able to quantify the additional radiative forcing and warming caused by permafrost carbon release."

In Section "4 Conclusions and Discussion", we have added a paragraph on the additional warming caused by the permafrost carbon release and its implications on $CO_2$ emission budgets, as follows:

"Different permafrost carbon release and associated additional warming under overshoot scenarios confirm the path-dependent fate of permafrost region carbon (Kleinen and Brovkin, 2018) and the path-dependent reductions in $CO_2$ emission budgets (MacDougall et al., 2015; Gasser et al., 2018). As the permafrost carbon was accumulated very slowly during the last millions of years, its release would be tacked onto the anthropogenic $CO_2$ emissions, and the resulting additional warming

poses a challenge to achieving global climate goals by substantially reducing the remaining carbon budget compatible with the Paris Agreement (MacDougall et al., 2015; Natali et al., 2021). In the overshoot scenarios simulated in this study, permafrost carbon release by 2300 ranges from 60 [35 to 86] PgC to 96 [63 to 135] PgC. The associated additional warming caused by the release ranges from 0.10 [0.06 to 0.15] °C to 0.18 [0.11 to 0.25] °C. This permafrost carbon feedback contributes a substantial addition on top of 1.5 °C warming target under overshoot scenarios, and the magnitude of this additional warming rises with the amplitude of overshoot. To accomplish the 1.5 °C target under the OS-2, OS-3, and OS-4 scenarios, anthropogenic carbon emissions would be reduced by amounts equivalent to the permafrost carbon release. The proportion of carbon removal required to offset permafrost emissions is estimated at 4.9 [2.9 to 7.1] %, 6.5 [4.1 to 9.2] %, and 8.3 [5.4 to 11.5] % by 2300, respectively. Our findings are consistent with previous research utilizing the Monte Carlo ensemble method to evaluate the response of permafrost carbon and its influence on $CO_2$ emission budgets under overshoot scenarios targeting a 1.5 °C warming limit (Gasser et al., 2018). Specifically, for overshoot amplitudes of 0.5 °C (peak warming of 2 °C) and 1 °C (peak warming of 2.5 °C), the reductions in anthropogenic $CO_2$ emissions due to permafrost are estimated to be 130 (with a range of 30–300) $PgCO_2$ and 210 (with a range of 50–430) $PgCO_2$, respectively, to meet the long-term 1.5 °C target (Gasser et al., 2018). These results are comparable to our estimates of 60 [35 to 86] PgC under OS-2 and 78 [50 to 111] PgC under OS-3. The differences between the two studies may be partly attributed to differences in the warming trajectories to achieve the same 1.5 °C target. Our study further confirms that if negative $CO_2$ emissions were to be used to reverse the anthropogenic climate change, the delayed permafrost carbon release would reduce its effectiveness (MacDougall, 2013; Tokarska and Zickfeld, 2015).

Line 159 – 169: I appreciate the perturbed parameter approach that's been taken here. It's presented well as uncertainty bounds in the text but includes some cues about the quantity of runs used and uncertainty bounds in more key figures would highlight it. Moreover, I recommend some discussion of any overlapping trajectories given the range of parameter uncertainty. Otherwise, these aspects of the manuscript may not be as apparent to the reader.

Thank you for your valuable suggestion. We have included the uncertainty bounds in the figures showing changes in GSAT, permafrost area, permafrost carbon loss and permafrost region soil carbon loss (Figure 2 in the original manuscript, Figure 3 in the revised manuscript); changes in permafrost carbon inputs and decomposition, as well as permafrost region soil carbon inputs and decomposition (Figure 4 in the original manuscript, Figure 5 in the revised manuscript); and additional changes in radiative forcing, GSAT warming and permafrost area due to permafrost carbon-climate feedback (Figure 5 in the original manuscript, Figure 7 in the revised manuscript) under overshoot and stabilization scenarios.

To facilitate the discussion of overlapping trajectories of the permafrost carbon, we have expanded the paragraph on the assessment of the relative importance of perturbed model parameters for permafrost region soil carbon release. In addition, we have placed this paragraph in Section "3.1 Permafrost Response" to enhance content coherence. The expanded paragraph reads as follows:

"The uncertainty in permafrost region soil carbon release is nearly the same as that of permafrost carbon release (Fig. 3c, d). For example, the 5th to 95th percentile ranges of permafrost region soil

carbon release under the OS-2 and OS-4 scenarios are 58 PgC and 81 PgC respectively, compared to 52 PgC and 72 PgC for permafrost carbon release. This indicates that the uncertainty in permafrost region soil carbon release is largely driven by the uncertainty in permafrost carbon release. Therefore, we evaluate the relative importance of perturbed permafrost carbon parameters on permafrost region soil carbon release under different temperature pathways through calculating their correlations across all ensemble simulations. In the SSP5-8.5, OS-4, and SWL-4 scenarios, the influence of model parameters on the uncertainty of permafrost carbon losses by 2300 is relatively consistent, with the strongest correlations observed for the permafrost passive carbon pool transformation rate (R=0.81~0.85), followed by the initial quantity of permafrost region soil carbon (R=0.55~0.61). This finding aligns with Ji et al. (2024), which highlights the critical role of these two parameters in the uncertainty of permafrost region soil carbon loss under temperature overshoot and 1.5 °C warming stabilization scenarios."

We have added a discussion of overlapping trajectories in the relevant quantities and explained the overlaps with the aid of the relative importance of perturbed parameters. The added discussion of the overlapping trajectories reads as follows:

"The uncertainty represented by perturbed model parameters for each scenario can be interpreted as model uncertainty. We note that model uncertainty in permafrost carbon release gradually increases with the peak warming level and the duration of overshoot for each scenario (Fig. 3c). However, the uncertainty ranges in permafrost carbon release for overshoot and stabilization scenarios with adjacent warming levels, such as OS-2, SWL-1.5 and SWL-2, substantially overlap. This is especially evident in low-level warming scenarios, where the uncertainty in projected permafrost carbon release is mainly driven by model uncertainty due to parameter perturbations, rather than scenario-related uncertainty. Given the significant roles of the permafrost passive carbon pool transformation rate and the initial quantity of permafrost region soil carbon in determining the uncertainty of permafrost region soil carbon release, it is expected that these two parameters contribute significantly to the overlapping uncertainty ranges of permafrost carbon and permafrost region soil carbon losses across different warming levels. Due to the interaction with soil carbon inputs, the overlapping uncertainty in permafrost region soil carbon release tends to differ from that of permafrost carbon release. For example, the uncertainty ranges in permafrost carbon release under OS-4 and SSP5-8.5 scenarios show considerable overlap (Fig. 3c), but the same does not apply to permafrost region soil carbon release (Fig. 3d), which results from significant differences in soil carbon inputs under distinct $CO_2$ fertilization backgrounds. The large overlapping uncertainty in projecting permafrost carbon release under low-level warming scenarios, as shown in this study and in previous research (MacDougall, 2015; MacDougall and Knutti, 2016; Gasser et al., 2018), constitutes a significant challenge in accurately estimating the remaining carbon budgets."

[Figure]

Figure 3. Timeseries of annual mean (a) GSAT changes, (b) permafrost area, (c) permafrost carbon loss and (d) permafrost region soil carbon loss under overshoot (solid lines) and stabilization (dashed lines) scenarios at 1.5 °C (blue), 2.0 °C (green), 3.0 °C (red), and 4 °C (purple) GWLs, as well as the SSP5-8.5 scenario. Square markers indicate the time points when the temperature overshoot reaches its peak or stabilized warming begins, while circle markers indicate when the temperatures return to 1.5 °C in the overshoot scenarios. All changes are relative to the pre-industrial period (1850-1900). Results represent the ensemble median of 250 simulations. Dots on the right panels represent values in the year 2300, with uncertainty ranges estimated as the 5th to 95th percentiles.

[Figure]

Figure 5. Timeseries of (a) permafrost carbon inputs, (b) permafrost region soil carbon inputs, (c) permafrost carbon decomposition and (d) permafrost region soil carbon decomposition, under the overshoot (solid lines) and stabilization (dashed lines) scenarios at 1.5 °C (blue), 2.0 °C (green), 3.0 °C (red) and 4 °C (purple) GWLs, along with the SSP5-8.5 scenario (black). Square markers indicate the time points when the temperature overshoot reaches its peak or stabilized warming begins, while circle markers indicate when the temperatures return to 1.5 °C in the overshoot scenarios. Results represent the ensemble median of 250 simulations. Dots on the right panels represent values in the year 2300, with uncertainty ranges estimated as the 5th to 95th percentiles.

[Figure]

Figure 7. Additional changes in (a) radiative forcing, (b) global mean surface air warming and (c, d) permafrost area due to permafrost carbon-climate feedback in the the overshoot (solid lines) and stabilization (dashed lines) scenarios at 1.5 °C (blue), 2.0 °C (green), 3.0 °C (red) and 4 °C (purple) GWLs, along with the SSP5-8.5 scenario (black). Square markers indicate the time points when the temperature overshoot reaches its peak or stabilized warming begins, while circle markers indicate when the temperatures return to 1.5 °C in the overshoot scenarios. Results represent the ensemble median of 250 simulations. Dots on the right panels represent values in the year 2300, with uncertainty ranges estimated as the 5th to 95th percentiles. In panel (a), the additional radiative forcing is calculated using the simplified expressions (Etminan et al., 2016) based on simulated $CO_2$ concentrations. In panels (c) and (d), the additional permafrost area loss is smoothed using a 5-year rolling average to eliminate shorter interannual variability.

Line 200: Greater elaboration on this result could be valuable. Assessing this impact at the year 2300 seems reasonable, however, given enough time will all the overshoot scenarios eventually converge with SWL-1.5?

We sincerely appreciate your valuable suggestion. To further investigate whether all overshoot scenarios eventually converge with SWL-1.5, we have extended the SWL-1.5 and overshoot

simulations to the year 2400 and compared the permafrost carbon inputs, decomposition and surface climate between them post-overshoot (Figure R2). A new paragraph has been added to Section "4 Conclusions and Discussion" to elaborate on this point and help readers better understand the model's long-term behavior. The added discussion reads as follows:

"The permafrost carbon under overshoot scenarios shows a certain degree of recovery relative to the stabilization scenario of SWL-1.5 (Fig. 4b). It is therefore of interest to assess whether the permafrost carbon under overshoot scenarios will eventually converge with that under SWL-1.5. To better explore this question, we extended the SWL-1.5 and overshoot simulations to year 2400 (data not shown). The permafrost carbon inputs are larger under overshoot scenarios than SWL-1.5, while the permafrost carbon decomposition shows much smaller differences between the two. This tends to indicate that the smaller permafrost carbon stocks under overshoot scenarios in 2300 would eventually catch up to the levels under SWL-1.5. To test this hypothesis, we estimate the time required to reach convergence by the ratio between the difference in permafrost carbon stocks and the difference in net permafrost carbon inputs (i.e., annual permafrost carbon inputs minus decomposition) for the overshoot scenarios relative to the SWL-1.5 scenario. Based on simulation outputs for the year 2300, the estimated median times for OS-2, OS-3 and OS-4 to catch up permafrost carbon with SWL-1.5 are 1076, 1008 and 1433 years, respectively. When based on outputs from the year 2400, the corresponding estimates increase to 1377, 1199 and 1568 years. This means that convergence would take even longer if estimated from later simulation years, mainly due to gradually weakened permafrost carbon inputs. The relatively larger permafrost carbon inputs under overshoot scenarios result not only from warmer climates during the stabilization period, but also from increased litterfall during the overshoot phase. This extra litterfall gradually moves through the active layer and is transported to the permafrost zone. Over time, however, the effect of this extra litterfall gradually diminishes, leading to a reduction in permafrost carbon inputs. Consequently, it may take extremely long timescales for the overshoot scenarios to fully converge with SWL-1.5 in terms of permafrost carbon stocks. In addition, the additional permafrost carbon release during the temperature overshoot period leads to greater additional warming under overshoot scenarios than the SWL-1.5 scenario, and this enhanced warming persists until the year 2400 without significant decline. Greater soil carbon loss under overshoot conditions also substantially alters the hydrological and thermal properties of soil, affecting the processes that govern carbon cycling. These persistent changes in surface climate and soil properties might stabilize the carbon, water, and energy cycles at alternative equilibria after overshoot through the interactions between physical and biophysical processes (de Vrese and Brovkin, 2021). Therefore, the overshoot scenarios might ultimately fail to converge to SWL-1.5 scenario in terms of permafrost carbon stocks."

[Figure]

Figure R2. Timeseries of changes in (a) permafrost carbon inputs, (b) permafrost carbon decomposition, (c) GSAT and (d) soil temperature in permafrost regions under overshoot scenarios (OS-2-pc, OS-3-pc, OS-4-pc) from the year 2200 to 2400. All variables are shown as anomalies relative to the SWL-1.5 scenario. Panel (d) shows the regional average of soil temperature in permafrost regions, averaged over the top 3.35 m of soil.

Line 220 and throughout: There is a substantial body of literature related to the response of arctic vegetation to climate change and the processes represented therein. Providing the reader with additional information on the vegetation model within UVICC ESM, the processes represented, limitations therein, including some information about the response of vegetation productivity and framing these results in that context would enhance their presentation. Additional background on the permafrost model would add additional clarity as to why permafrost carbon inputs do not appear to follow the same trajectory as soil carbon.

Thank you for your suggestion. We have expanded the background information on the vegetation model within UVic ESCM as follows:

"The terrestrial component uses the Top-down Representation of Interactive Foliage and Flora Including Dynamics (TRIFFID) vegetation model to describe the states of five plant functional types (PFT): broadleaf tree, needleleaf tree, C3 grass, C4 grass, and shrub (Cox, 2001; Meissner et al., 2003). A coupled photosynthesis-stomatal conductance model is used to calculate carbon uptake via photosynthesis, which is subsequently allocated to vegetation growth and respiration. The resulting net carbon fluxes drive changes in vegetation characteristics, including areal coverage, leaf area index, and canopy height for each PFT. The UVic ESCM v2.10 utilized in this

study does not account for nutrient limitations in the terrestrial carbon cycle, leading to an overestimation of global gross primary productivity and an enhanced capacity of land to take up atmospheric carbon (De Sisto et al., 2023). However, the model reasonably represents the dominant PFTs of C3 grass, shrub and needleleaf tree at northern high latitudes, although it underestimates vegetation carbon density over this area (Mengis et al., 2020)."

We have also added the response of vegetation productivity in Section "3 Results" to better understanding permafrost region soil carbon inputs as following:

"The permafrost region soil carbon inputs generally track the trajectory of litter flux across the same area, with an approximate delay of 10-20 years (now shown). To attribute the contribution of permafrost region soil carbon inputs, we examined how dominant vegetation types (needleleaf tree, C3 grass and shrub) over the permafrost region adapt to temperature and atmospheric $CO_2$ concentrations in both overshoot and stabilization scenarios (Figure 6). Needleleaf trees expand slowly and continuously in the permafrost region in both overshoot and stabilization scenarios, while the areal coverage of shrubs closely follows the trajectory of GSAT. The combined areal coverage of trees and shrubs is projected to cover about 62% upon 1.5 °C warming relative to pre-industrial levels around 2040s, slightly higher than the 24-52% range projected for 2050 using a statistical approach that links climate conditions to vegetation types under two distinct emission trajectories (Pearson et al., 2013). During the warming and cooling phases of overshoot scenarios, the expansion and reduction of shrubs correspond with the degradation and expansion of C3 grasses, respectively. Among the three dominant PFTs, only shrubs show a nearly reversible response in areal coverage, net primary productivity (NPP) and vegetation carbon with respect to GSAT under overshoot scenarios. In contrast, the continuous reduction of C3 grasses and the expansion of needleleaf trees suggest a degree of irreversibility in the structure and vegetation carbon density of northern high latitude terrestrial ecosystems under overshoot scenarios. This contrasts with a previous study that employed prescribed vegetation distributions, which reported only minor differences in vegetation carbon after the overshoots compared to the reference simulation with no overshoot (Schwinger et al., 2022). In our study, the shifts in vegetation composition and changes in living biomass, especially those associated with woody vegetation, are key drivers of permafrost region soil carbon inputs."

[Figure]

Figure 6. Timeseries of annual mean areal fraction (left column), net primary productivity (middle column) and vegetation carbon (right column) under overshoot (solid lines) and stabilization (dashed lines) scenarios at 1.5 °C (blue), 2.0 °C (green), 3.0 °C (red), and 4 °C (purple) GWLs, as well as the SSP5-8.5 scenario (black). Each row represents one of the three dominant plant functional type (PFT): (a-c) needleleaf tree, (d-f) C3 grass and (g-i) shrub. Square markers indicate the time points when the temperature overshoot reaches its peak or stabilized warming begins, while circle markers indicate when the temperatures return to 1.5 °C in the overshoot scenarios. Results represent the ensemble median of 250 simulations, and the uncertainty is not displayed due to the small range.

To explain why permafrost carbon inputs do not follow the same trajectory as soil carbon, especially under overshoot scenarios, we emphasized that in the model, litterfall is allocated to soil layers with temperatures above 1 °C according to an exponentially decreasing function of depth. When all soil layers are below 1 °C, organic carbon from the litterfall is added to the top soil layer. Meanwhile, permafrost carbon and non-permafrost soil carbon are both represented as depth-resolved carbon pools within the top six soil layers. The movement of permafrost carbon due to cryoturbation mixing is parameterized as being proportional to the gradient of total soil carbon with depth. Soil carbon that diffuses downward through the permafrost table is converted to permafrost carbon. During the cooling phase of overshoot scenarios, increased litterfall and a rising permafrost table lead to elevated carbon concentrations in surface soil layers, resulting in a surge in permafrost carbon inputs. Conversely, under the SSP5-8.5 scenario, permafrost carbon inputs only show a minor peak. This is due to the continuous reduction in permafrost area and the deepening of the permafrost table, both of which reduce carbon concentrations in the upper soil layers and weaken vertical diffusion, despite the increasing litter flux under a strong $CO_2$ fertilization background. We note that the approach adopted in the model may not accurately reflect natural processes of vertical carbon movement, which are influenced by soil porosity heterogeneity, freeze-thaw cycles, and ice expansion upon freezing.

Minor comments:

Abstract: for clarity suggest reducing the use of acronyms in the abstract and possibly parts of the text.

Thank you for your suggestion. In response to your suggestion, we have reduced the use of acronyms in the abstract and replaced "SWL" and "OS" with their full terms, "stabilization" and "overshoot." We will continue to review acronym usage throughout the manuscript to ensure clarity and accessibility.

Line 25: gradual and abrupt seem to refer to the rate of carbon loss suggest revision to distinguish this from the processes of gradual and abrupt thaw.

Thank you for your suggestion. We have revised the wording to ensure that gradual and abrupt explicitly refer to the thawing processes. Line 25 has been changed to "gradual or abrupt permafrost thaw, along with subsequent microbial decomposition, would release carbon dioxide ($CO_2$) and methane ($CH_4$) into the atmosphere...".

Line 50: suggest adding further discission of the mechanism behind the presence or absence of hysteresis behavior in different processes as this is useful background.

Thank you for this insightful suggestion. We agree that a more detailed discussion of the mechanism behind the presence or absence of hysteresis behavior in different processes would strengthen the background of our study. To address this, we have expanded the discussion in this section by highlighting key factors influencing hysteresis in permafrost carbon dynamics, including thermophysical inertia, microbial decomposition lag, and hydrological feedbacks.

The revised text now reads: "Additionally, a temporary warming of the permafrost regions entails important legacy effects and lasting impacts on their physical state and carbon cycle under various overshoot levels. The presence or absence of hysteresis effect in these processes is influenced by multiple factors, including the thermal inertia of permafrost soils, potential shifts in vegetation composition, and varying levels of compensation between irreversible permafrost carbon losses and gains in vegetation and non-permafrost soil carbon reservoirs (MacDougall, 2013; Schwinger et al., 2022). Furthermore, the soil carbon loss under overshoot scenarios significantly affects the hydrological and thermal properties of soils, which in turn modulate the processes involved. The interactions between physical and biophysical processes can potentially stabilize the carbon, water, and energy cycles at distinct post-overshoot equilibria (de Vrese and Brovkin, 2021)."

Line 229: Suggest clarifying the timescale being discussed in this summary information. It reads very similarly to the sentence immediately prior.

Thank you for the suggestion. We have revised this section to improve clarity and to specify the timescale. The revised manuscript is as follows:

"In all overshoot and stabilization scenarios simulated in this study, the permafrost region soil serves as a net carbon source for atmospheric $CO_2$ by 2300. However, during the stabilization phase of OS-3 and OS-4 (Fig. 2d), the permafrost region soil turns into a carbon sink, as soil carbon inputs surpass the reduced decomposition activity due to the depletion of soil carbon stocks and reduced warming levels."

**References**

Avis, C. A., Weaver, A. J., and Meissner, K. J.: Reduction in areal extent of high-latitude wetlands in response to permafrost thaw, Nature Geoscience, 4, 444–448, https://doi.org/10.1038/ngeo1160, 2011.

Chadburn, S. E., Burke, E. J., Cox, P. M., Friedlingstein, P., Hugelius, G., and Westermann, S.: An observation-based constraint on permafrost loss as a function of global warming, Nature Climate Change, 7, 340–344, https://doi.org/10.1038/nclimate3262, 2017.

Cox, P.: Description of the TRIFFID dynamic global vegetation model. Hadley Centre Technical., 24, 1–16, 2001.

De Sisto, M. L., MacDougall, A. H., Mengis, N., and Antoniello, S.: Modelling the terrestrial nitrogen and phosphorus cycle in the UVic ESCM, Geoscientific Model Development, 16, 4113–4136, https://doi.org/10.5194/gmd-16-4113-2023, 2023.

De Vrese, P. and Brovkin, V.: Timescales of the permafrost carbon cycle and legacy effects of temperature overshoot scenarios, Nature Communications, 12, 2688, https://doi.org/10.1038/s41467-021-23010-5, 2021.

Etminan, M., Myhre, G., Highwood, E. J., and Shine, K. P.: Radiative forcing of carbon dioxide, methane, and nitrous oxide: A significant revision of the methane radiative forcing, Geophysical Research Letters, 43, https://doi.org/10.1002/2016GL071930, 2016.

Friedlingstein, P., O'Sullivan, M., Jones, M. W., Andrew, R. M., Hauck, J., Olsen, A., Peters, G. P., Peters, W., Pongratz, J., Sitch, S., Le Quéré, C., Canadell, J. G., Ciais, P., Jackson, R. B., Alin, S., Aragão, L. E. O. C., Arneth, A., Arora, V., Bates, N. R., Becker, M., Benoit-Cattin, A., Bittig, H. C., Bopp, L., Bultan, S., Chandra, N., Chevallier, F., Chini, L. P., Evans, W., Florentie, L., Forster, P. M., Gasser, T., Gehlen, M., Gilfillan, D., Gkritzalis, T., Gregor, L., Gruber, N., Harris, I., Hartung, K., Haverd, V., Houghton, R. A., Ilyina, T., Jain, A. K., Joetzjer, E., Kadono, K., Kato, E., Kitidis, V., Korsbakken, J. I., Landschützer, P., Lefèvre, N., Lenton, A., Lienert, S., Liu, Z., Lombardozzi, D., Marland, G., Metzl, N., Munro, D. R., Nabel, J. E. M. S., Nakaoka, S.-I., Niwa, Y., O'Brien, K., Ono, T., Palmer, P. I., Pierrot, D., Poulter, B., Resplandy, L., Robertson, E., Rödenbeck, C., Schwinger, J., Séférian, R., Skjelvan, I., Smith, A. J. P., Sutton, A. J., Tanhua, T., Tans, P. P., Tian, H., Tilbrook, B., van der Werf, G., Vuichard, N., Walker, A. P., Wanninkhof, R., Watson, A. J., Willis, D., Wiltshire, A. J., Yuan, W., Yue, X., and Zaehle, S.: Global Carbon Budget 2020, Earth System Science Data, 12, 3269–3340, https://doi.org/10.5194/essd-12-3269-2020, 2020.

Gasser, T., Kechiar, M., Ciais, P., Burke, E. J., Kleinen, T., Zhu, D., Huang, Y., Ekici, A., and Obersteiner, M.: Path-dependent reductions in CO2 emission budgets caused by permafrost carbon release, Nature Geoscience, 11, 830–835, https://doi.org/10.1038/s41561-018-0227-0, 2018.

Gulev, S. K., Thorne, P. W., Ahn, J., Dentener, F. J., Domingues, C. M., Gerland, S., Gong, D., Kaufman, D. S., Nnamchi, H. C., Quaas, J., Rivera, J. A., Sathyendranath, S., Smith, S. L., Trewin, B., von Schuckmann, K., Vose, R. S., Allan, R., Collins, B., Turner, A., and Hawkins, E.: Changing state of the climate system, edited by: Masson-Delmotte, V., Zhai, P., Pirani, A., Connors, S. L., Péan, C., Berger, S., Caud, N., Chen, Y., Goldfarb, L., Gomis, M. I., Huang, M., Leitzell, K., Lonnoy, E., Matthews, J. B. R., Maycock, T. K., Waterfield, T., Yelekçi, O., Yu, R., and Zhou, B., Cambridge University Press, Cambridge, UK, 287–422, 2021.

Hugelius, G., Strauss, J., Zubrzycki, S., Harden, J. W., Schuur, E. a. G., Ping, C.-L., Schirrmeister, L., Grosse, G., Michaelson, G. J., Koven, C. D., O'Donnell, J. A., Elberling, B., Mishra, U., Camill, P., Yu, Z., Palmtag, J., and Kuhry, P.: Estimated stocks of circumpolar permafrost carbon with quantified uncertainty ranges and identified data gaps, Biogeosciences, 11, 6573–6593, https://doi.org/10.5194/bg-11-6573-2014, 2014.

Kleinen, T. and Brovkin, V.: Pathway-dependent fate of permafrost region carbon, Environmental Research Letters, 13, 094001, https://doi.org/10.1088/1748-9326/aad824, 2018.

Koven, C., Friedlingstein, P., Ciais, P., Khvorostyanov, D., Krinner, G., and Tarnocai, C.: On the formation of high-latitude soil carbon stocks: Effects of cryoturbation and insulation by organic matter in a land surface model, Geophysical Research Letters, 36, https://doi.org/10.1029/2009GL040150, 2009.

MacDougall, A. H.: Reversing climate warming by artificial atmospheric carbon-dioxide removal: Can a Holocene-like climate be restored? Geophysical Research Letters, 40, 5480–5485, https://doi.org/10.1002/2013GL057467, 2013.

MacDougall, A. H., Zickfeld, K., Knutti, R., and Matthews, H. D.: Sensitivity of carbon budgets to permafrost carbon feedbacks and non-CO2 forcings, Environmental Research Letters, 10, 125003, https://doi.org/10.1088/1748-9326/10/12/125003, 2015.

MacDougall, A. H. and Knutti, R.: Projecting the release of carbon from permafrost soils using a perturbed parameter ensemble modelling approach, Biogeosciences, 13, 2123–2136, https://doi.org/10.5194/bg-13-2123-2016, 2016.

MacDougall, A. H.: Estimated effect of the permafrost carbon feedback on the zero emissions commitment to climate change, Biogeosciences, 18, 4937–4952, https://doi.org/10.5194/bg-18-4937-2021, 2021.

McKay, M. D., Beckman, R. J., and Conover, W. J.: A Comparison of Three Methods for Selecting Values of Input Variables in the Analysis of Output from a Computer Code, Technometrics, 21, 239–245, https://doi.org/10.2307/1268522, 1979.

Meissner, K. J., Weaver, A. J., Matthews, H. D., and Cox, P. M.: The role of land surface dynamics in glacial inception: a study with the UVic Earth System Model, Climate Dynamics, 21, 515–537, https://doi.org/10.1007/s00382-003-0352-2, 2003.

Mengis, N., Keller, D. P., MacDougall, A. H., Eby, M., Wright, N., Meissner, K. J., Oschlies, A., Schmittner, A., MacIsaac, A. J., Matthews, H. D., and Zickfeld, K.: Evaluation of the University of Victoria Earth System Climate Model version 2.10 (UVic ESCM 2.10), Geoscientific Model Development, 13, 4183–4204, https://doi.org/10.5194/gmd-13-4183-2020, 2020.

Natali, S. M., Holdren, J. P., Rogers, B. M., Treharne, R., Duffy, P. B., Pomerance, R., and MacDonald, E.: Permafrost carbon feedbacks threaten global climate goals, Proceedings of the National Academy of Sciences, 118, e2100163118, https://doi.org/10.1073/pnas.2100163118, 2021.

Pearson, R. G., Phillips, S. J., Loranty, M. M., Beck, P. S. A., Damoulas, T., Knight, S. J., and Goetz, S. J.: Shifts in Arctic vegetation and associated feedbacks under climate change, Nature Climate Change, 3, 673–677, https://doi.org/10.1038/nclimate1858, 2013.

Schuur, E. a. G., McGuire, A. D., Schädel, C., Grosse, G., Harden, J. W., Hayes, D. J., Hugelius, G., Koven, C. D., Kuhry, P., Lawrence, D. M., Natali, S. M., Olefeldt, D., Romanovsky, V. E., Schaefer, K., Turetsky, M. R., Treat, C. C., and Vonk, J. E.: Climate change and the permafrost carbon feedback, Nature, 520, 171–179, https://doi.org/10.1038/nature14338, 2015.

Schwinger, J., Asaadi, A., Steinert, N. J., and Lee, H.: Emit now, mitigate later? Earth system reversibility under overshoots of different magnitudes and durations, Earth System Dynamics, 13, 1641–1665, https://doi.org/10.5194/esd-13-1641-2022, 2022.

Strauss, J., Schirrmeister, L., Grosse, G., Fortier, D., Hugelius, G., Knoblauch, C., Romanovsky, V., Schädel, C., Schneider von Deimling, T., Schuur, E. A. G., Shmelev, D., Ulrich, M., and Veremeeva, A.: Deep Yedoma permafrost: A synthesis of depositional characteristics and carbon vulnerability, Earth-Science Reviews, 172, 75–86, https://doi.org/10.1016/j.earscirev.2017.07.007, 2017.

Strauss, J., Laboor, S., Schirrmeister, L., Fedorov, A. N., Fortier, D., Froese, D., Fuchs, M., Günther, F., Grigoriev, M., Harden, J., Hugelius, G., Jongejans, L. L., Kanevskiy, M., Kholodov, A., Kunitsky, V., Kraev, G., Lozhkin, A., Rivkina, E., Shur, Y., Siegert, C., Spektor, V., Streletskaya, I., Ulrich, M., Vartanyan, S., Veremeeva, A., Anthony, K. W., Wetterich, S., Zimov, N., and Grosse, G.: Circum-Arctic Map of the Yedoma Permafrost Domain, Frontiers in Earth Science, 9, 758360, https://doi.org/10.3389/feart.2021.758360, 2021.

Strauss, J., Laboor, S., Schirrmeister, L., Fedorov, A. N., Fortier, D., Froese, D. G., Fuchs, M., Günther, F., Grigoriev, M. N., Harden, J. W., Hugelius, G., Jongejans, L. L., Kanevskiy, M. Z., Kholodov, A. L., Kunitsky, V., Kraev, G., Lozhkin, A. V., Rivkina, E., Shur, Y., Siegert, C., Spektor, V., Streletskaya, I., Ulrich, M., Vartanyan, S. L., Veremeeva, A., Walter Anthony, K. M., Wetterich,

S., Zimov, N. S., and Grosse, G.: Database of Ice-Rich Yedoma Permafrost Version 2 (IRYP v2), PANGAEA, https://doi.org/10.1594/PANGAEA.940078, 2022.

Tokarska, K. B. and Zickfeld, K.: The effectiveness of net negative carbon dioxide emissions in reversing anthropogenic climate change, Environmental Research Letters, 10, 094013, https://doi.org/10.1088/1748-9326/10/9/094013, 2015.

Zhang, T., Heginbottom ,J. A., Barry ,R. G., and Brown, J.: Further statistics on the distribution of permafrost and ground ice in the Northern Hemisphere 1, Polar Geography, 24, 126‒131, https://doi.org/10.1080/10889370009377692, 2000.

Zimov, S. A., Schuur, E. A. G., and Chapin, F. S.: Permafrost and the Global Carbon Budget, Science, 312, 1612‒1613, https://doi.org/10.1126/science.1128908, 2006.

---

## Author Comment (AC2)

We express our sincere gratitude to your insightful and constructive comments. Please find below our point-by-point replies. All the comments are presented in black text and the corresponding replies are highlighted in blue.

Overall:

This study "aims to fill these gaps using an Earth system model of intermediate complexity to systematically assess the permafrost response and feedback under temperature stabilization or overshoot scenarios achieving various GWLs. " I think it is an interesting paper and recommend publication, but I also think it could make some clearer points, as I discuss below.

Based on that stated aim, I expected to see one or more figures with total permafrost carbon losses plotted as a function of the GWL for both the stabilization and overshoot cases. I.e., is the permafrost carbon loss linear? Are there thresholds or tipping points? Figure 6a shows the areal loss as a function of global warming level, but why are carbon variables not quantified in this way? Does the permafrost carbon feedback strength (in units of Pg C / degree Celsius warming) show a similar nonlinearity as the SPAW shown in f.g 6a with maximum losses per unit warming in the 1.5-2 degree C range? Figure 3b seems to show that the highest sensitivity is in the ~3 degree warming range, but it is difficult to see quantitatively. Likewise it would be interesting to se the radiative forcing as well. So I'd recommend an additional figure with panels along the lines of 6a that allows the reader to trace how the (non-)linearity of each of these permafrost metrics as a function of global warming levels for the stabilization and overshoot cases changes between permafrost area, permafrost carbon, and permafrost radiative forcing.

Thank you for your insightful suggestion. We fully agree that using global warming levels (GWLs) as the horizontal axis to present key permafrost metrics helps reveal their linear or nonlinear behavior. In response, we have added a new figure (Figure 8) to the revised manuscript, which includes three panels showing (a) permafrost area change, (b) permafrost carbon loss, and (c) permafrost radiative forcing, all plotted as a function of GWLs. Colored dashed lines represent stabilization scenarios, colored solid lines represent overshoot scenarios, and the black solid line denotes the SSP5-8.5 scenario.

[Figure]

Figure 8. Relationship between global warming levels and three permafrost metrics: (a) permafrost area change, (b) permafrost carbon loss, and (c) permafrost radiative forcing in the overshoot (colored solid lines) and stabilization (colored dashed lines) scenarios at 1.5 °C (blue), 2.0 °C (green), 3.0 °C (red) and 4 °C (purple) GWLs, along with the SSP5-8.5 scenario (black). Square

and circle markers indicate values in the year 2300 for the stabilization and overshoot scenarios, respectively. Grey solid lines show linear fits of permafrost metrics to GWLs in stabilization scenarios by 2300, while black dashed lines show corresponding fits for the SSP5-8.5 scenario. Note that in panel (a), both the stabilization scenarios and the corresponding SSP5-8.5 points included in the linear fit are limited to GWLs between 1.5 °C and 3.0 °C, whereas in panels (b) and (c), the fits include points with GWLs ranging from 1.5 °C to 4.0 °C. For stabilization scenarios, only the results from the year 2300 are used for fitting, while for the SSP5-8.5 scenario, all results within the specified GWL ranges are used for fitting. Shaded regions represent the 5th to 95th percentile ranges across 250 ensemble simulations.

Our results show permafrost area change shows a strongly nonlinear relationship with GWLs greater than 1 °C. However, there is a quasilinear relation between them for the GWLs between 1.5 °C and 3 °C. The decreasing sensitivity of permafrost area to GWLs above 3 °C is evident in both the stabilization scenarios and SSP5-8.5 scenario (Figure 8a). In contrast, permafrost carbon loss and associated radiative forcing exhibit a nearly linear response to increasing GWLs, especially above 1.5 °C, for both stabilization and SSP5-8.5 scenarios (Figures 8b,c). Meanwhile, the sensitivities of permafrost area to global warming (SPAW), permafrost carbon feedback, and associated radiative forcing under stabilization scenarios are all stronger than those under the SSP5-8.5 scenario. Specifically, based on the simulated permafrost area in the year 2300 under stabilization scenarios with GWLs between 1.5 °C and 3 °C, the SPAW is -3.19 [-3.01 to -3.36] million $km^2$ $°C^{-1}$. In comparison, a linear fit of permafrost area against GWLs over the same temperature range in the SSP5-8.5 scenario yields a SPAW of -2.85 [-2.77 to -2.89] million $km^2$ $°C^{-1}$。 Similarly, the sensitivity of permafrost carbon feedback, derived from a linear fit based on the total permafrost carbon loss in the 2300 year under stabilization scenarios, is -27.6 [-16.5 to -38.2] PgC $°C^{-1}$. In contrast, the corresponding value under the SSP5-8.5 scenario, estimated from a linear fit over the 1.5 °C to 4.0 °C warming range, is -19.3 [-15.7 to -24.1] PgC $°C^{-1}$. Applying the same approach, the associated radiative forcing per degree of warming is estimated to be 0.08 [0.05 to 0.12] W $m^{-2}$ $°C^{-1}$ for the stabilization scenarios and 0.04 [0.03 to 0.05] W $m^{-2}$ $°C^{-1}$ for the SSP5-8.5. These differences between the stabilization and SSP5-8.5 scenarios are mainly attributable to the differing response time scales represented by the two scenarios: SSP5-8.5 reflects a typical transient response, while the stabilization scenarios maintain stabilized temperatures over extended periods and thus approximate a quasi-equilibrium response of the climate-carbon system. Furthermore, the smaller sensitivity of permafrost radiative forcing under the SSP5-8.5 can be partially attributed to its higher background atmospheric $CO_2$ concentration compared to the stabilization scenarios. The same amount of $CO_2$ emissions would produce smaller additional radiative forcing under higher background atmospheric $CO_2$ concentration, due to the logarithmic relationship between $CO_2$ concentration and radiative forcing (Etminan et al., 2016). Our results are consistent with the conclusions of Nitzbon et al. (2024), who reported that the accumulated response of Arctic permafrost to climate warming remains approximately quasilinear. Moreover, the permafrost carbon feedback sensitivities derived from both the stabilization and SSP5-8.5 scenarios fall within the ranges reported in existing literature, namely -18 [-3 to -41] PgC $°C^{-1}$ from Nitzbon et al. (2024) and -21 [-4 to -48] PgC $°C^{-1}$ from Canadell et al. (2021).

Under overshoot scenarios, permafrost area responds nearly reversibly and presents an almost closed loop (Figure 8a). In contrast, permafrost carbon loss exhibits an open loop with respect to

GWLs, that is, permafrost carbon loss does not reverse as temperatures decline, indicating irreversible permafrost radiative forcing. Among the three metrics investigated here, only permafrost area exhibits strong reversibility under the overshoot scenarios. This also explains why, in Figure 6a (figure number according to the original manuscript), the permafrost area sensitivity derived from the SSP5-8.5 scenario can be used, when multiplied by additional warming, to reasonably reconstruct permafrost area loss in the stabilization and overshoot cases.

Further, given the possibility of perturbing parameters due to the relatively low cost of running UVic-ESCM, I had expected to see if any of those parameters introduced nonlinearities or substantially changed the magnitude of the results. But I just see median lines. So it is hard to know how important the uncertainty is. I suggest showing the uncertainty via translucent colored plumes in all figures.

We sincerely appreciate your valuable suggestion. We previously attempted to display uncertainty via translucent colored plumes. However, results from different scenarios overlapped significantly (see Figure below), making the visualization overly cluttered and difficult to interpret.

[Figure]

Nevertheless, we fully acknowledge the importance of showing the uncertainty to accurately convey the results. Therefore, we have represented the 5th to 95th percentiles of all ensemble simulations for the year 2300 in all figures to help readers better understand the uncertainty associated with those parameters. For example, the following figures include uncertainties:

[Figure]

Figure 3. Timeseries of annual mean (a) GSAT changes, (b) permafrost area, (c) permafrost carbon loss and (d) permafrost region soil carbon loss under overshoot (solid lines) and stabilization (dashed lines) scenarios at 1.5 °C (blue), 2.0 °C (green), 3.0 °C (red), and 4 °C (purple) GWLs, as well as the SSP5-8.5 scenario. Square markers indicate the time points when the temperature overshoot reaches its peak or stabilized warming begins, while circle markers indicate when the temperatures return to 1.5 °C in the overshoot scenarios. All changes are relative to the pre-industrial period (1850-1900). Results represent the ensemble median of 250 simulations. Dots on the right panels represent values in the year 2300, with uncertainty ranges estimated as the 5th to 95th percentiles.

[Figure]

Figure 5. Timeseries of (a) permafrost carbon inputs, (b) permafrost region soil carbon inputs, (c) permafrost carbon decomposition and (d) permafrost region soil carbon decomposition, under the overshoot (solid lines) and stabilization (dashed lines) scenarios at 1.5 °C (blue), 2.0 °C (green), 3.0 °C (red) and 4 °C (purple) GWLs, along with the SSP5-8.5 scenario (black). Square markers indicate the time points when the temperature overshoot reaches its peak or stabilized warming begins, while circle markers indicate when the temperatures return to 1.5 °C in the overshoot scenarios. Results represent the ensemble median of 250 simulations. Dots on the right panels represent values in the year 2300, with uncertainty ranges estimated as the 5th to 95th percentiles.

[Figure]

Figure 7. Additional changes in (a) radiative forcing, (b) global mean surface air warming and (c, d) permafrost area due to permafrost carbon-climate feedback in the the overshoot (solid lines) and stabilization (dashed lines) scenarios at 1.5 °C (blue), 2.0 °C (green), 3.0 °C (red) and 4 °C (purple) GWLs, along with the SSP5-8.5 scenario (black). Square markers indicate the time points when the temperature overshoot reaches its peak or stabilized warming begins, while circle markers indicate when the temperatures return to 1.5 °C in the overshoot scenarios. Results represent the ensemble median of 250 simulations. Dots on the right panels represent values in the year 2300, with uncertainty ranges estimated as the 5th to 95th percentiles. In panel (a), the additional radiative forcing is calculated using the simplified expressions (Etminan et al., 2016) based on simulated $CO_2$ concentrations. In panels (c) and (d), the additional permafrost area loss is smoothed using a 5-year rolling average to eliminate shorter interannual variability.

Comments

line 36: the 1.5 degree budget will be exhausted within the next few years, but not the 2 degree budget. Please clarify.

Thank you for your suggestion. We have revised the sentence for greater clarity:

"If current emission rates persist, the remaining carbon budgets compatible with the 1.5 °C target will be critically tight and likely exhausted within the next few years (Rogelj et al., 2015; Goodwin et al., 2018; Masson-Delmotte et al., 2018; Forster et al., 2023; Smith et al., 2023)."

Paragraph starting line 142: This is great that you were able to perturb these key parameters. But I don't see any uncertainty plumes in any of the figures, only the median values. I think it would be informative to the reader to see the partameter uncertainty plumes plotted on all figures.

Thank you for your suggestion. We have incorporated the 5th to 95th percentiles of all ensemble simulations for the year 2300 in figures to better represent the uncertainty associated with these key perturbed parameters.

fig. 2b: Why doesn't the permafrost area recover all the way under the overshoot scenarios? Are there regional changes to the northern high latitude climate that are responsible for the differing permafrost amounts at a given GWL? If so, what are the drivers of that regional change? It might help to add a panel with the regional temperature difference to see whether it behaves differently from the global mean.

Thank you for your valuable comments. The incomplete recovery of permafrost area under the overshoot scenarios in Fig. 2b (figure number according to the original manuscript) is influenced by multiple factors: First, Figure 2b shows results from simulations with the permafrost carbon module switched on, and the persistent additional warming due to permafrost carbon feedback causes additional permafrost degradation. During stabilization phases of overshoot scenarios, the northern high-latitude permafrost regions experience additional warming compared to the SWL-1.5 scenario, with regional temperature increases of 0.01~0.14 °C by 2300. Second, the thermal inertia of deep soil layers limits the rate of permafrost recovery. Even when global temperatures return to the 1.5 °C target, residual heat accumulated in deeper soil layers during temperature overshoot period continues to inhibit permafrost refreezing, preventing full restoration to its initial state. Third, the greater soil carbon loss under overshoot scenarios has a strong impact on the soil's hydrological and thermal properties (Avis, 2012; Lawrence et al., 2008), which in turn affects the recovery of permafrost area. Moreover, irreversible shifts in vegetation composition of high-latitude terrestrial ecosystems also contribute to the incomplete recovery of permafrost area under overshoot scenarios. For instance, as two dominant vegetation types, needleleaf trees continue to expand and C3 grasses decline, even after global temperatures return to the 1.5 °C level. These irreversible changes may stabilize the carbon, water, and energy cycles over the permafrost region at different equilibria after overshoot through the interactions between physical and biophysical processes (de Vrese and Brovkin, 2021), constraining the ability of permafrost to fully recover under the overshoot scenarios.

Line 321: This paper doesn't really establish anything about the realism of the model, since there are no model-data comparisons, so suggest reword or provide citations to the papers that have shown this.

Thank you for your valuable comment. We acknowledge the need for additional clarification regarding the realism of the UVic ESCM model. To address this, we have incorporated model

validation into the first paragraph of Section "3 Results". This addition provides a direct comparison between model simulations and observational data, demonstrating that the UVic ESCM model realistically reproduces historical permafrost area and permafrost carbon stocks. The added content is:

"The UVic ESCM v2.10 reliably simulates historical temperature changes, permafrost area, and the partitioning of anthropogenic carbon emissions among the atmosphere, ocean and land. During the period 2011–2020, the model estimated a GSAT increase of 1.14 [1.13 to 1.15] °C relative to preindustrial levels, which is closely aligned with the observed rise of 1.09 [0.91 to 1.23] °C (Gulev et al., 2021). For the period 1960–1990, the model simulated the northern hemisphere permafrost area at 16.8 [16.7 to 16.9] million $km^2$, which falls within the reconstructed range of 12.0 to 18.2 million $km^2$ (Chadburn et al., 2017) and the observation derived extent of 12.21 to 16.98 million $km^2$ (Zhang et al., 2000). Additionally, the modeled soil carbon stock in the top 3.35 m of permafrost regions for this same period was 1034 [919 to 1151] PgC, with 483 [382 to 587] PgC classified as perennially frozen carbon, accounting for 47% [42% to 51%] of the total permafrost soil carbon stock, in agreement with Hugelius et al. (2014). From 2010 to 2019, the model estimated that anthropogenic carbon emissions were distributed as follows: 5.5 [5.4 to 5.6] PgC $yr^{-1}$ to the atmosphere, 3.0 [2.98 to 3.03] PgC $yr^{-1}$ to the ocean, and 2.5 [2.4 to 2.6] PgC $yr^{-1}$ to terrestrial ecosystems. These estimates are broadly consistent with the global anthropogenic $CO_2$ budget assessment by the Global Carbon Project (GCP) with figures of 5.1±0.02 PgC $yr^{-1}$ for the atmosphere, 2.5±0.6 PgC $yr^{-1}$ for the ocean, and 3.4±0.9 PgC $yr^{-1}$ for terrestrial ecosystems (Friedlingstein et al., 2020)."

Accordingly, we have revised the statement in Line 321 to: "UVic ESCM has been validated against observational and reconstructed datasets, demonstrating its ability to reproduce historical permafrost area and permafrost carbon stocks."

Data availability: I downloaded some of the data files in Cui et al., 2024, but they aren't clearly described and don't include any further details than what is in the paper (e.g., spatial information). This strikes me as a fairly minimal data archival effort.

We appreciate your comments regarding the data archiving. We have added more detailed descriptions to the uploaded data files, including variable names, units, and associated spatial and temporal dimensions, in order to improve the clarity.

Due to the large volume of spatial model output, we have archived only the key variables necessary to support the main analyses presented in the paper (https://zenodo.org/records/15148252). Although this may not capture all details, we are happy to provide more comprehensive datasets upon request. In addition, a data description file (README.md) has been included alongside the archived model output to facilitate understanding.

**References**

Avis, C. A., Weaver, A. J., and Meissner, K. J.: Reduction in areal extent of high-latitude wetlands in response to permafrost thaw, Nature Geoscience, 4, 444–448, https://doi.org/10.1038/ngeo1160, 2011.

Canadell, J., Forster, P., Meyer, C., and the Chapter 5 authors: Global carbon and other biogeochemical cycles and feedbacks, in: Climate Change 2021: The Physical Science Basis. Contribution of Working Group I to the Sixth Assessment Report of the Intergovernmental Panel on Climate Change, edited by: Masson-Delmotte, V., Zhai, P., Pirani, A., et al., Cambridge University Press, Cambridge, United Kingdom and New York, NY, USA, Chapter 5, https://doi.org/10.1017/9781009157896.007, 2021.

Chadburn, S. E., Burke, E. J., Cox, P. M., Friedlingstein, P., Hugelius, G., and Westermann, S.: An observation-based constraint on permafrost loss as a function of global warming, Nature Climate Change, 7, 340–344, https://doi.org/10.1038/nclimate3262, 2017.

De Vrese, P. and Brovkin, V.: Timescales of the permafrost carbon cycle and legacy effects of temperature overshoot scenarios, Nature Communications, 12, 2688, https://doi.org/10.1038/s41467-021-23010-5, 2021.

Etminan, M., Myhre, G., Highwood, E. J., and Shine, K. P.: Radiative forcing of carbon dioxide, methane, and nitrous oxide: A significant revision of the methane radiative forcing, Geophysical Research Letters, 43, https://doi.org/10.1002/2016GL071930, 2016.

Forster, P. M., Smith, C. J., Walsh, T., Lamb, W. F., Lamboll, R., Hauser, M., Ribes, A., Rosen, D., Gillett, N., Palmer, M. D., Rogelj, J., von Schuckmann, K., Seneviratne, S. I., Trewin, B., Zhang, X., Allen, M., Andrew, R., Birt, A., Borger, A., Boyer, T., Broersma, J. A., Cheng, L., Dentener, F., Friedlingstein, P., Gutiérrez, J. M., Gütschow, J., Hall, B., Ishii, M., Jenkins, S., Lan, X., Lee, J.-Y., Morice, C., Kadow, C., Kennedy, J., Killick, R., Minx, J. C., Naik, V., Peters, G. P., Pirani, A., Pongratz, J., Schleussner, C.-F., Szopa, S., Thorne, P., Rohde, R., Rojas Corradi, M., Schumacher, D., Vose, R., Zickfeld, K., Masson-Delmotte, V., and Zhai, P.: Indicators of Global Climate Change 2022: annual update of large-scale indicators of the state of the climate system and human influence, Earth System Science Data, 15, 2295–2327, https://doi.org/10.5194/essd-15-2295-2023, 2023.

Friedlingstein, P., O'Sullivan, M., Jones, M. W., Andrew, R. M., Hauck, J., Olsen, A., Peters, G. P., Peters, W., Pongratz, J., Sitch, S., Le Quéré, C., Canadell, J. G., Ciais, P., Jackson, R. B., Alin, S., Aragão, L. E. O. C., Arneth, A., Arora, V., Bates, N. R., Becker, M., Benoit-Cattin, A., Bittig, H. C., Bopp, L., Bultan, S., Chandra, N., Chevallier, F., Chini, L. P., Evans, W., Florentie, L., Forster, P. M., Gasser, T., Gehlen, M., Gilfillan, D., Gkritzalis, T., Gregor, L., Gruber, N., Harris, I., Hartung, K., Haverd, V., Houghton, R. A., Ilyina, T., Jain, A. K., Joetzjer, E., Kadono, K., Kato, E., Kitidis, V., Korsbakken, J. I., Landschützer, P., Lefèvre, N., Lenton, A., Lienert, S., Liu, Z., Lombardozzi, D., Marland, G., Metzl, N., Munro, D. R., Nabel, J. E. M. S., Nakaoka, S.-I., Niwa, Y., O'Brien, K., Ono, T., Palmer, P. I., Pierrot, D., Poulter, B., Resplandy, L., Robertson, E., Rödenbeck, C., Schwinger, J., Séférian, R., Skjelvan, I., Smith, A. J. P., Sutton, A. J., Tanhua, T., Tans, P. P., Tian, H., Tilbrook, B., van der Werf, G., Vuichard, N., Walker, A. P., Wanninkhof, R., Watson, A. J.,

Willis, D., Wiltshire, A. J., Yuan, W., Yue, X., and Zaehle, S.: Global Carbon Budget 2020, Earth System Science Data, 12, 3269 – 3340, https://doi.org/10.5194/essd-12-3269-2020, 2020.

Goodwin, P., Katavouta, A., Roussenov, V. M., Foster, G. L., Rohling, E. J., and Williams, R. G.: Pathways to 1.5 ° C and 2 ° C warming based on observational and geological constraints, Nature Geoscience, 11, 102 – 107, https://doi.org/10.1038/s41561-017-0054-8, 2018.

Gulev, S. K., Thorne, P. W., Ahn, J., Dentener, F. J., Domingues, C. M., Gerland, S., Gong, D., Kaufman, D. S., Nnamchi, H. C., Quaas, J., Rivera, J. A., Sathyendranath, S., Smith, S. L., Trewin, B., von Schuckmann, K., Vose, R. S., Allan, R., Collins, B., Turner, A., and Hawkins, E.: Changing state of the climate system, edited by: Masson-Delmotte, V., Zhai, P., Pirani, A., Connors, S. L., Péan, C., Berger, S., Caud, N., Chen, Y., Goldfarb, L., Gomis, M. I., Huang, M., Leitzell, K., Lonnoy, E., Matthews, J. B. R., Maycock, T. K., Waterfield, T., Yelekçi, O., Yu, R., and Zhou, B., Cambridge University Press, Cambridge, UK, 287 – 422, 2021.

Hugelius, G., Strauss, J., Zubrzycki, S., Harden, J. W., Schuur, E. a. G., Ping, C.-L., Schirrmeister, L., Grosse, G., Michaelson, G. J., Koven, C. D., O'Donnell, J. A., Elberling, B., Mishra, U., Camill, P., Yu, Z., Palmtag, J., and Kuhry, P.: Estimated stocks of circumpolar permafrost carbon with quantified uncertainty ranges and identified data gaps, Biogeosciences, 11, 6573–6593, https://doi.org/10.5194/bg-11-6573-2014, 2014.

Lawrence, D. M., Slater, A. G., Tomas, R. A., Holland, M. M., and Deser, C.: Accelerated Arctic land warming and permafrost degradation during rapid sea ice loss, Geophysical Research Letters, 35, https://doi.org/10.1029/2008GL033985, 2008.

Masson-Delmotte, V., Zhai, P., Pörtner, H.-O., Roberts, D. C., Skea, J., Shukla, P. R., Pirani, A., Moufouma-Okia, W., Péan, C., Pidcock, R., Connors, S., Matthews, J. B. R., Yang, C., Zhou, X., and Steg, L.: Global warming of 1.5 ° C: Summary for policy makers, IPCC - The Intergovernmental Panel on Climate Change, https://doi.org/10.1017/9781009157940.001, 2018.

Nitzbon, J., Schneider Von Deimling, T., Aliyeva, M., Chadburn, S. E., Grosse, G., Laboor, S., Lee, H., Lohmann, G., Steinert, N. J., Stuenzi, S. M., Werner, M., Westermann, S., and Langer, M.: No respite from permafrost-thaw impacts in the absence of a global tipping point, Nature Climate Change, 14, 573 – 585, https://doi.org/10.1038/s41558-024-02011-4, 2024.

Rogelj, J., Meinshausen, M., Schaeffer, M., Knutti, R., and Riahi, K.: Impact of short-lived non-CO2 mitigation on carbon budgets for stabilizing global warming, Environmental Research Letters, 10, 075001, https://doi.org/10.1088/1748-9326/10/7/075001, 2015.

Smith, S., Geden, O., Nemet, G., Gidden, M., Lamb, W., Powis, C., Bellamy, R., Callaghan, M., Cowie, A., Cox, E., Fuss, S., Gasser, T., Grassi, G., Greene, J., Lueck, S., Mohan, A., Müller-Hansen, F., Peters, G., Pratama, Y., Repke, T., Riahi, K., Schenuit, F., Steinhauser, J., Strefler, J., Valenzuela, J., and Minx, J.: State of Carbon Dioxide Removal - 1st Edition, https://doi.org/10.17605/OSF.IO/W3B4Z, 2023.

Zhang, T., Heginbottom ,J. A., Barry ,R. G., and Brown, J.: Further statistics on the distribution of permafrost and ground ice in the Northern Hemisphere 1, Polar Geography, 24, 126－131, https://doi.org/10.1080/10889370009377692, 2000.

---

## Author Response (AR1)

We express our sincere gratitude to the Editor and the two referees for their insightful and constructive comments. Please find below our point-by-point replies. All the comments are presented in black text and the corresponding replies are highlighted in blue.

**Referee #1**

The authors use a fully coupled climate model to evaluate the response of permafrost under temperature stabilization and overshoot scenarios. The methods appear rigorous, and the manuscript is well-written. However, the manuscript could be improved by discussing the implications of some of the feedbacks being modeled. Substantial revisions to some sections of the text and figures could further improve the overall clarity of the manuscript and link it more directly to existing literature.

**Major comments:**

Line 92 and throughout: Yedoma represents a significant, deep proportion of the permafrost carbon stock in some regions and was formed over extremely long timescales. Given the focus on differences between overshoot and stabilization scenarios, a greater discussion of this limitation could be included in the methods and conclusions.

We appreciate your insightful comment. The UVic ESCM v2.10 simulates permafrost carbon only in the top six layers (to a depth of 3.35 m), and therefore omits soil carbon stored in the deep deposits of Yedoma regions. As a result, we cannot directly estimate the impacts of temperature overshoot on deep Yedoma carbon, or compare these changes relative to stabilization scenarios. To address this limitation, we analyzed the average and maximum active layer thickness (ALT) in Yedoma regions between the overshoot and stabilization scenarios simulated in this study. Using differences in ALT as a proxy to infer the potential impacts on deep Yedoma carbon (Figure S6). To clarify this point, we have added the following paragraph to Section "4 Conclusions and Discussion":

"This study does not simulate the changes of deep Yedoma carbon under the temperature stabilization and overshoot scenarios. Yedoma deposits represent a significant deep carbon reservoir and are widespread across Siberia, Alaska, and the Yukon region of Canada, having primarily formed during the late Pleistocene, especially in the late glacial period. These deep, perennially frozen sediments are particularly ice-rich, and the freeze-locked organic matter in such deposits can be re-mobilized on short time-scales, representing one of the most vulnerable permafrost carbon pools under future warming scenarios (Schuur et al., 2015; Strauss et al., 2017). According to Zimov et al. (2006), these perennially frozen Yedoma sediments cover more than 1 million km2, with an average depth of approximately 25 m. Recent estimates place the organic carbon stock in Yedoma deposits at  $213 \pm 24$  PgC, constituting a significant portion of the total permafrost carbon pool (Strauss et al., 2017). However, the UVic ESCM v2.10 utilized in this study simulates permafrost carbon only within the top 3.35 m of soil, limiting our ability to directly assess the impacts of temperature overshoot on deep Yedoma carbon. Considering their ice-rich nature and potential susceptibility to rapid-thaw processes, we analyzed the average and maximum active layer thickness (ALT) in Yedoma regions (Strauss et al., 2021, 2022) under the simulated scenarios to approximate potential impacts. We find that the average ALT in Yedoma regions remains below 1 m in all stabilization and overshoot scenarios, while the maximum ALT rarely exceeds 3.35 m in overshoot

scenarios but does exceed this depth in some stabilization scenarios. However, in all scenarios, the maximum ALT does not exceed 6 m, which is relatively shallow compared to the average depth (~25 m) of Yedoma deposits (Figure S6). Consequently, the impact on Yedoma is considered to be minimal in all scenarios, and the effect of overshoot scenarios on the deep Yedoma carbon is relatively minor compared to stabilization scenarios as well."

Figure S6. Timeseries of annual (a) average and (b) maximum active layer thickness (ALT) in Yedoma regions under overshoot (colored solid lines) and stabilization (colored dashed lines) scenarios at 1.5 °C (blue), 2.0 °C (green), 3.0 °C (red), and 4.0 °C (purple) GWLs, as well as the SSP5-8.5 scenario (black solid line). Results represent the ensemble median of 250 simulations based on the PFC simulations. The Yedoma region mask used in this analysis is based on the Ice-Rich Yedoma Permafrost Database Version 2 (IRYP v2) (Strauss et al., 2021, 2022; https://doi.org/10.1594/PANGAEA.940078).

Line 90 - 103: More background on the UVICC ESM permafrost carbon model, validation, and perturbed parameter approach would be particularly useful to readers.

Thank you for your valuable suggestion. To improve clarity and provide a more comprehensive background, we have made the following revisions to Section "2.1 Model description":

"The UVic ESCM v2.10 represents the terrestrial subsurface with 14 layers, extending to a total depth of 250.3 m to correctly capture the transient response of permafrost on centennial timescales. The top eight layers (10.0 m) are involved in the hydraulic cycle, while the deeper layers are modeled as impermeable bedrock (Avis et al., 2011). The carbon cycle is active in the top six layers (3.35 m), where organic carbon from litterfall, simulated by the TRIFFID vegetation model, is allocated to soil layers with temperatures above 1 °C according to an exponentially decreasing function with depth. If all soil layers are below 1 °C, the organic carbon is added to the top soil layer. The soil respiration is calculated for each layer individually as a function of temperature and moisture, but the respiration ceases when the soil layer temperature falls below 0 °C (Meissner et al., 2003; Mengis et al., 2020). In regions where permafrost exists—defined as areas where soil temperature remains below 0 °C for at least two consecutive years—the model applies a revised diffusion-based cryoturbation scheme to redistribute soil carbon within the soil column. Compared to the original diffusion-based cryoturbation scheme proposed by Koven et al. (2009), the revised cryoturbation scheme calculates carbon diffusion using an effective carbon concentration that

incorporates the volumetric porosity of the soil layer, rather than the actual carbon concentration, thereby resolving the disequilibrium problem of the permafrost carbon pool during model spin-up (MacDougall and Knutti, 2016). However, as the UVic ESCM v2.10 only simulates permafrost carbon in the top 3.35 m of soil, the current cryoturbation scheme cannot initiate the formation of Yedoma. As a result, soil carbon stored in deep deposits of Yedoma regions is omitted in our simulations."

The perturbed parameter approach is described at the end of Section "2.2 Experimental Design". We have refined the description to improve coherence. Additionally, we have added a figure to illustrate the probability distribution functions of the four perturbed key permafrost carbon parameters (Figure 2). The revised text and added figure are as follows:

"The Latin hypercube sampling method (McKay et al., 1979) was used to explore the effects of parameter uncertainty on projections of permafrost carbon change. In this study, the probability distribution function of each key permafrost carbon parameter was divided into 25 intervals of equal probability. One value was randomly selected from each interval for a given parameter, and then randomly matched with values of the other three key parameters selected in the same manner to generate parameter sets. This sampling procedure was repeated 10 times, resulting in 250 unique parameter sets (i.e., 250 model variants). For each parameter set, the UVic ESCM v2.10 was first run through a 10,000-year spin-up phase under pre-industrial conditions to achieve a quasi-equilibrium state. For these spin-up runs, the atmospheric CO2 concentration was fixed at 284.7 ppm and the solar constant was set to 1360.747 W m-2. Following the spin-up, emission-driven transient experiments were conducted under the stabilization, overshoot and SSP5-8.5 scenarios. The results are presented as the median across all model variants, with uncertainty quantified as the range between the 5th to the 95th percentiles."

Figure 2. Probability distribution functions of the four key permafrost carbon parameters perturbed in the UVic ESCM v2.10 to represent uncertainty in permafrost carbon response. Panel (d) employs a logarithmic scale on the horizontal axis to better illustrate the distribution of the corresponding parameter. This figure is reproduced from MacDougall (2021).

Model validation has been incorporated into the first paragraph in Section "3 Results". The added content is:

"The UVic ESCM v2.10 reliably simulates historical temperature changes, permafrost area, and the partitioning of anthropogenic carbon emissions among the atmosphere, ocean and land. Under preindustrial conditions, the simulated Northern Hemisphere permafrost area, defined as regions where the soil layer remains perennially frozen for at least two consecutive years, was 17.01 [17.00 to 17.04] million km², the simulated total soil carbon stock in the permafrost regions was 1031 [915 to 1149] PgC, of which 484 [383 to 590] PgC was classified as perennially frozen carbon and 547 [533 to 559] PgC was classified as usual soil carbon. During For the period 1960–1990, the model simulated Northern Hemisphere permafrost area at 16.8 [16.7 to 16.9] million km², which falls within the reconstructed range from 12.0 to 18.2 million km² (Chadburn et al., 2017) and the observation derived extent from 12.21 to 16.98 million km² (Zhang et al., 2000). Additionally, the simulated soil carbon stock in the top 3.35 m of permafrost regions for this same period was 1034 [919 to 1151] PgC, with 483 [382 to 587] PgC classified as perennially frozen carbon, accounting for 47% [42% to 51%] of the total permafrost soil carbon stock, in agreement with Hugelius et al. (2014). During the period 2011–2020, the model estimated a global mean temperature increase of

1.14 [1.13 to 1.15] °C relative to preindustrial levels, which is closely aligned with the observed rise of 1.09 [0.91 to 1.23] °C (Gulev et al., 2021). From 2010 to 2019, the model estimated that anthropogenic carbon emissions were distributed as follows: 5.5 [5.4 to 5.6] PgC yr-1 to the atmosphere, 3.0 [2.98 to 3.03] PgC yr-1 to the ocean, and 2.5 [2.4 to 2.6] PgC yr-1 to terrestrial ecosystems. These estimates are broadly consistent with the global anthropogenic CO2 budget assessment by the Global Carbon Project (GCP) with figures of 5.1±0.02 PgC yr-1 for the atmosphere, 2.5±0.6 PgC yr-1 for the ocean, and 3.4±0.9 PgC yr-1 for terrestrial ecosystems (Friedlingstein et al., 2020)."

Line 129 – 136: More clarity is needed about this aspect of the method and the interaction with any permafrost feedback loops. It appears these experiments were done to create drivers for the overshoot scenarios (i.e. the proportional control scheme is not active when the final model runs for analysis are done). It then appears based on this text and the text in section 3.2 that any permafrost carbon fluxes would be tacked onto the emissions and removals needed to accomplish these scenarios. Significant edits are needed here for clarity. The additional warming in Figure 5 also appears well-suited for additional discussion.

Thank you for your detailed comments. Actually, the proportional control scheme was used only in the initial set of simulations, which were conducted to generate CO2 emission trajectories that follow the intended warming pathways of each scenario (Fig. 1a). These emission trajectories were then used to drive the formal experiments for both stabilization and overshoot scenarios, without any further application of the proportional control scheme during the final model integrations.

To isolate the contribution of permafrost carbon feedback, we conducted two parallel sets of formal experiments for each scenario, both driven by the same CO2 emission trajectories. The only difference between these sets lies in whether the permafrost carbon module was activated or not. As you noted, any permafrost carbon fluxes would be tacked onto the emissions and removals needed to accomplish these scenarios. To clearly distinguish the role of permafrost carbon feedback, we refer to the simulations with the permafrost carbon module activated as PFC simulations, and those with the module deactivated as NPFC simulations. The comparison between the PFC and NPFC simulations allows us to robustly isolate the additional warming and radiative forcing induced by permafrost carbon emissions under both stabilization and overshoot scenarios.

To improve clarity, we have revised the relevant section as follows:

"To isolate the contribution of permafrost carbon feedback, two parallel sets of formal experiments were conducted for each scenario, both driven by the same CO2 emission trajectories diagnosed from the initial simulations. One set activated the permafrost carbon module and is referred to as PFC simulations, while the other set deactivated the permafrost carbon module and is referred to as NPFC simulations. Since the emissions trajectories were diagnosed from the initial simulations in which the proportional control scheme was applied to achieve the desired temperature pathways, applying them in the formal simulations with the permafrost carbon module deactivated can effectively achieve the designed temperature trajectories (Fig. 1b). However, applying the diagnosed emissions in simulations with the permafrost module activated results in any permafrost carbon fluxes being effectively added on top of the diagnosed emissions, thereby causing additional

warming. In other words, to achieve the intended climate targets under the same emission pathways, removals equivalent to the permafrost carbon emissions would be required. Therefore, the comparison between the PFC and NPFC simulation sets provides a robust framework to isolate and quantify the additional warming and radiative forcing effects due to permafrost carbon emissions under stabilization and overshoot scenarios. For comparison, two parallel simulations with permafrost carbon module activated or deactivated were also conducted for the high-emissions SSP5-8.5 scenario."

We have also revised the first paragraph of Section "3.2 Radiative Impacts of Permafrost Carbon Release" on radiative impacts of permafrost carbon release to better clarify the expression of additional warming. The revised text is as follows:

"The permafrost carbon release would increase global mean radiative forcing and surface temperature. By comparing two parallel sets of simulations with the permafrost carbon module activated (PFC) or deactivated (NPFC), we were able to quantify the additional radiative forcing and warming caused by permafrost carbon release."

In Section "4 Conclusions and Discussion", we have added a paragraph on the additional warming caused by the permafrost carbon release and its implications on CO2 emission budgets, as follows:

"Different permafrost carbon release and associated additional warming under overshoot scenarios confirm the path-dependent fate of permafrost region carbon (Kleinen and Brovkin, 2018) and the path-dependent reductions in CO2 emission budgets (MacDougall et al., 2015; Gasser et al., 2018). As the permafrost carbon was accumulated very slowly during the last millions of years, its release would be tacked onto the anthropogenic CO2 emissions, and the resulting additional warming poses a challenge to achieving global climate goals by substantially reducing the remaining carbon budget compatible with the Paris Agreement (MacDougall et al., 2015; Natali et al., 2021). In the overshoot scenarios simulated in this study, permafrost carbon release by 2300 ranges from 60 [35 to 87] PgC to 97 [63 to 135] PgC. The associated additional warming caused by the release ranges from 0.10 [0.06 to 0.15] °C to 0.18 [0.11 to 0.25] °C. This permafrost carbon feedback contributes a substantial addition on top of 1.5 °C warming target under overshoot scenarios, and the magnitude of this additional warming rises with the amplitude of overshoot. To accomplish the 1.5 °C target under the OS-2, OS-3, and OS-4 scenarios, anthropogenic carbon emissions would be reduced by amounts equivalent to the permafrost carbon release. The proportion of carbon removal required to offset permafrost emissions is estimated at 4.9 [2.9 to 7.1] %, 6.5 [4.1 to 9.2] %, and 8.3 [5.4 to 11.6] % by 2300, respectively. Our findings are consistent with previous research utilizing the Monte Carlo ensemble method to evaluate the response of permafrost carbon and its influence on CO2 emission budgets under overshoot scenarios targeting a 1.5 °C warming limit (Gasser et al., 2018). Specifically, for overshoot amplitudes of 0.5 °C (peak warming of 2 °C) and 1 °C (peak warming of 2.5 °C), the reductions in anthropogenic CO2 emissions due to permafrost are estimated to be 130 (with a range of 30 - 300) Pg CO2 and 210 (with a range of 50 - 430) Pg CO2, respectively, to meet the long-term 1.5 °C target (Gasser et al., 2018). These results are comparable to our estimates of 60 [35 to 87] PgC under OS-2 and 78 [50 to 111] PgC under OS-3. The differences between the two studies may be partly attributed to different warming trajectories to achieve the same 1.5 °C target. Our study further confirms that if negative CO2 emissions were to be used to reverse the

anthropogenic climate change, the delayed permafrost carbon release would reduce its effectiveness (MacDougall, 2013; Tokarska and Zickfeld, 2015)."

Line 159 – 169: I appreciate the perturbed parameter approach that's been taken here. It's presented well as uncertainty bounds in the text but includes some cues about the quantity of runs used and uncertainty bounds in more key figures would highlight it. Moreover, I recommend some discussion of any overlapping trajectories given the range of parameter uncertainty. Otherwise, these aspects of the manuscript may not be as apparent to the reader.

Thank you for your valuable suggestion. We have included the uncertainty bounds in the figures showing global warming, permafrost area loss, permafrost carbon loss and permafrost region soil carbon loss (Figure 2 in the original manuscript, Figure 3 in the revised manuscript); changes in permafrost carbon inputs and decomposition, as well as permafrost region soil carbon inputs and decomposition (Figure 4 in the original manuscript, Figure 5 in the revised manuscript); and additional changes in radiative forcing, global warming and permafrost area due to permafrost carbon-climate feedback (Figure 5 in the original manuscript, Figure 7 in the revised manuscript) under overshoot and stabilization scenarios.

To facilitate the discussion of overlapping trajectories of the permafrost carbon, we have expanded the paragraph on the assessment of the relative importance of perturbed model parameters for permafrost region soil carbon release. In addition, we have placed this paragraph in Section "3.1 Permafrost Response" to enhance content coherence. The expanded paragraph reads as follows:

"The uncertainty in permafrost region soil carbon release is nearly the same as that of permafrost carbon release (Fig. 3c, d; Fig. S1c, d). For example, the 5th to 95th percentile range of permafrost region soil carbon release under the OS-2 and OS-4 scenarios is 58 PgC and 81 PgC respectively, compared to 52 PgC and 72 PgC for permafrost carbon release. This indicates that the uncertainty in permafrost region soil carbon release is largely driven by the uncertainty in permafrost carbon release. Therefore, we evaluate the relative importance of perturbed permafrost carbon parameters on permafrost region soil carbon release under different temperature pathways through calculating their correlations across all ensemble simulations. In the SSP5-8.5, OS-4, and SWL-4 scenarios, the influence of model parameters on the uncertainty of permafrost carbon losses by 2300 is relatively consistent, with the strongest correlations observed for the permafrost passive carbon pool transformation rate (R=0.81~0.85), followed by the initial quantity of permafrost region soil carbon (R=0.55~0.61). This finding aligns with Ji et al. (2024), who highlights the critical role of these two parameters in the uncertainty of permafrost region soil carbon loss under temperature overshoot and 1.5 °C warming stabilization scenarios."

We have added a discussion of overlapping trajectories in the relevant quantities and explained the overlaps with the aid of the relative importance of perturbed parameters. The added discussion of the overlapping trajectories reads as follows:

"This study, like previous ones, uncovers considerable uncertainty in projections of permafrost carbon under global warming. The uncertainty represented by perturbed model parameters for each scenario can be interpreted as model uncertainty. We note that model uncertainty in permafrost

carbon release gradually increases with the peak warming level and the duration of overshoot for each scenario (Fig. 3c; Fig. S1c). However, the uncertainty ranges in permafrost carbon release for overshoot and stabilization scenarios with adjacent warming levels, such as OS-2, SWL-1.5 and SWL-2, substantially overlap. This is especially evident in low-level warming scenarios, where the uncertainty in projected permafrost carbon release is mainly driven by model uncertainty due to parameter perturbations, rather than scenario-related uncertainty. Given the significant roles of the permafrost passive carbon pool transformation rate and the initial quantity of permafrost region soil carbon in determining the uncertainty of permafrost region soil carbon release, it is expected that these two parameters contribute significantly to the overlapping uncertainty ranges of permafrost carbon and permafrost region soil carbon losses across different warming levels. Due to the interaction with soil carbon inputs, the overlapping uncertainty in permafrost region soil carbon release tends to differ from that of permafrost carbon release. For example, the uncertainty ranges in permafrost carbon release under OS-4 and SSP5-8.5 scenarios show considerable overlap, but the same does not apply to permafrost region soil carbon release, which results from significant differences in soil carbon inputs under distinct CO2 fertilization backgrounds. The large overlapping uncertainty in projecting permafrost carbon release under low-level warming scenarios, as shown in this study and in previous research (MacDougall, 2015; MacDougall and Knutti, 2016; Gasser et al., 2018), constitutes a significant challenge in accurately estimating the remaining carbon budgets consistent with temperature goals of the Paris Agreement."

Figure 3. Timeseries of annual mean (a) global warming, (b) permafrost area, (c) permafrost carbon loss and (d) permafrost region soil carbon loss under stabilization (dashed lines) and overshoot (solid lines) scenarios at 1.5 °C (blue), 2.0 °C (green), 3.0 °C (red), and 4.0 °C (purple) global warming levels, as well as the SSP5-8.5 scenario. Square markers indicate the time points when the temperature overshoot reaches its peak or stabilized warming begins, while circle markers indicate when the overshoot returns to 1.5 °C. All changes are relative to the pre-industrial period (1850-1900). Results represent the ensemble median of 250 simulations based on the PFC simulations.

Dots on the right panels represent values in the year 2300, with uncertainty ranges estimated as the 5th to 95th percentiles.

Figure 5. Timeseries of changes in (a) permafrost carbon inputs, (b) permafrost region soil carbon inputs, (c) permafrost carbon decomposition and (d) permafrost region soil carbon decomposition, under the stabilization (dashed lines) and overshoot (solid lines) scenarios at 1.5 °C (blue), 2.0 °C (green), 3.0 °C (red) and 4.0 °C (purple) global warming levels, along with the SSP5-8.5 scenario (black). Square markers indicate the time points when the temperature overshoot reaches its peak or stabilized warming begins, while circle markers indicate when the overshoot returns to 1.5 °C. Results represent the ensemble median of 250 simulations based on the PFC simulations. Dots on the right panels represent values in the year 2300, with uncertainty ranges estimated as the 5th to 95th percentiles.

Figure 7. Additional changes in (a) radiative forcing, (b) global warming and (c, d) permafrost area due to permafrost carbon feedback, calculated as the difference between the PFC and NPFC simulations. Shown are results for the stabilization (dashed lines) and overshoot (solid lines) scenarios at 1.5 °C (blue), 2.0 °C (green), 3.0 °C (red) and 4.0 °C (purple) global warming levels, along with the SSP5-8.5 scenario (black). Square markers indicate the time points when the temperature overshoot reaches its peak or stabilized warming begins, while circle markers indicate when the overshoot returns to 1.5 °C. Results represent the ensemble median of 250 simulations. Dots on the right panels represent values in the year 2300, with uncertainty ranges estimated as the 5th to 95th percentiles. In panel (a), the additional radiative forcing is calculated using the simplified expressions (Etminan et al., 2016) based on simulated CO2 concentrations. In panels (c) and (d), the additional permafrost area loss is smoothed using a 5-year rolling average to eliminate interannual variability.

Line 200: Greater elaboration on this result could be valuable. Assessing this impact at the year 2300 seems reasonable, however, given enough time will all the overshoot scenarios eventually converge with SWL-1.5?

We sincerely appreciate your valuable suggestion. To further investigate whether all overshoot scenarios eventually converge with SWL-1.5, we have extended the SWL-1.5 and overshoot simulations to the year 2400 and compared the permafrost carbon inputs, decomposition and surface climate between them post-overshoot (Figure R1). A new paragraph has been added to Section "4 Conclusions and Discussion" to elaborate on this point and help readers better understand the model's long-term behavior. The added paragraph reads as follows:

"Although permafrost carbon loss is essentially irreversible, overshoot scenarios exhibit a certain degree of recovery relative to the SWL-1.5 stabilization scenario (Fig. 4b; Fig. S2b). It is therefore curious to know whether permafrost carbon under overshoot scenarios will eventually converge

with that under SWL-1.5. Our results show that permafrost carbon inputs are consistently higher under overshoot scenarios than under SWL-1.5, while permafrost carbon decomposition differ only slightly between the two (Fig. 5a, c; Fig. S3a, c). This tends to suggest that the smaller permafrost carbon stocks under overshoot scenarios by 2300 would eventually catch up to the levels under SWL-1.5. To assess this potential convergence, we extended our simulations of both SWL-1.5 and overshoot scenarios to the year 2400 (data not shown). Then we estimated the convergence time by calculating the ratio between the difference in permafrost carbon stocks and the difference in net permafrost carbon inputs (i.e., annual permafrost carbon inputs minus decomposition) for the overshoot scenarios relative to the SWL-1.5 scenario. Based on simulation results for the year 2300, the median estimated convergence times for OS-2, OS-3 and OS-4 are 1076, 1008 and 1433 years, respectively. When using results from the year 2400, the corresponding estimates increase to 1377, 1199 and 1568 years. This means that convergence would take even longer if estimated from later simulation results, mainly due to gradually weakened permafrost carbon inputs. The relatively larger permafrost carbon inputs under overshoot scenarios result mainly from increased litterfall during the overshoot phase. The extra litterfall during the overshoot phase gradually moves through the active layer and is transported to the permafrost zone. Over time, however, the effect of this extra litterfall gradually diminishes, leading to a reduction in permafrost carbon inputs. Consequently, it may take extremely long timescales for the overshoot scenarios to fully converge with SWL-1.5 in terms of permafrost carbon stocks. In addition, due to incomplete recovery of permafrost area and persistent changes in surface climate and soil properties, the overshoot scenarios might ultimately fail to converge to SWL-1.5 scenario in terms of permafrost carbon stocks."

Figure R1. Timeseries of changes relative to the SWL-1.5 scenario in (a) permafrost carbon inputs, (b) permafrost carbon decomposition, (c) global warming and (d) soil temperature in permafrost regions under overshoot scenarios at 2.0 °C (green), 3.0 °C (red), and 4.0 °C (purple) global warming levels from the year 2200 to 2400. Results represent the ensemble median of 250 simulations based on the PFC simulations. Panel (d) shows the regional average of soil temperature in permafrost regions, averaged over the top 3.35 m of soil.

Line 220 and throughout: There is a substantial body of literature related to the response of arctic vegetation to climate change and the processes represented therein. Providing the reader with additional information on the vegetation model within UVICC ESM, the processes represented, limitations therein, including some information about the response of vegetation productivity and framing these results in that context would enhance their presentation. Additional background on the permafrost model would add additional clarity as to why permafrost carbon inputs do not appear to follow the same trajectory as soil carbon.

Thank you for your suggestion. We have expanded the background information on the vegetation model within UVic ESCM as follows in Section "2.1 Model Description":

"The terrestrial component of UVic ESCM v2.10 uses the Top-down Representation of Interactive Foliage and Flora Including Dynamics (TRIFFID) vegetation model to describe the states of five plant functional types (PFT): broadleaf tree, needleleaf tree, C3 grass, C4 grass, and shrub (Cox, 2001; Meissner et al., 2003). A coupled photosynthesis-stomatal conductance model is used to calculate carbon uptake via photosynthesis, which is subsequently allocated to vegetation growth and respiration. The resulting net carbon fluxes drive changes in vegetation characteristics, including areal coverage, leaf area index, and canopy height for each PFT. The UVic ESCM v2.10 utilized in this study does not account for nutrient limitations in the terrestrial carbon cycle, leading to an overestimation of global gross primary productivity and an enhanced capacity of land to take up atmospheric carbon (De Sisto et al., 2023). However, the model reasonably represents the dominant PFTs of C3 grass, shrub and needleleaf tree at northern high latitudes, although it underestimates vegetation carbon density over this area (Mengis et al., 2020)."

We have also added the response of vegetation productivity in Section "3.1 Permafrost Response" to better understanding permafrost region soil carbon inputs as following:

"The permafrost region soil carbon inputs generally track the trajectory of litter flux across the same area, with an approximate delay of 10-20 years (not shown). To attribute the contribution of permafrost region soil carbon inputs, we examined how dominant vegetation types (needleleaf tree, C3 grass and shrub) over the permafrost region adapt to temperature and atmospheric CO2 concentrations in both overshoot and stabilization scenarios (Fig. 6). Needleleaf trees expand slowly and continuously in the permafrost region in both overshoot and stabilization scenarios, whereas that of shrubs closely follows the trajectory of global mean temperature. The combined areal coverage of trees and shrubs is projected to cover about 62% upon 1.5 °C warming relative to preindustrial levels around 2040s, slightly higher than the 24~52% range projected for 2050 using a statistical approach that links climate conditions to vegetation types under two distinct emission trajectories (Pearson et al., 2013). During the warming and cooling phases of overshoot scenarios, the expansion and reduction of shrubs correspond with the degradation and expansion of C3 grasses, respectively. Among the three dominant PFTs, only shrubs show a nearly reversible response in areal coverage, net primary productivity (NPP) and vegetation carbon with respect to global mean temperature under overshoot scenarios. In contrast, the continuous reduction of C3 grasses and the expansion of needleleaf trees suggest a degree of irreversibility in the structure and vegetation carbon density of northern high latitude terrestrial ecosystems under overshoot scenarios. Our results are in line with an earlier study by Tokarska and Zickfeld (2015), but contrast with Schwinger

et al. (2022) who reported only minor differences in vegetation carbon after the overshoots compared to the reference simulation with no overshoot by prescribing vegetation distributions. In our study, the shifts in vegetation composition and changes in living biomass, especially those associated with woody vegetation, are key drivers of permafrost region soil carbon inputs."

Figure 6. Timeseries of annual mean areal fraction (left column), net primary productivity (middle column) and vegetation carbon (right column) under stabilization (dashed lines) and overshoot (solid lines) scenarios at 1.5 °C (blue), 2.0 °C (green), 3.0 °C (red), and 4.0 °C (purple) global warming levels, as well as the SSP5-8.5 scenario (black). Each row represents one of the three dominant plant functional type (PFT): (a-c) needleleaf tree, (d-f) C3 grass and (g-i) shrub. Square markers indicate the time points when the temperature overshoot reaches its peak or stabilized warming begins, while circle markers indicate when the overshoot returns to 1.5 °C. Results represent the ensemble median of 250 simulations based on the PFC simulations, and the shadings denote the 5th and 95th percentile uncertainty ranges.

To explain why permafrost carbon inputs do not follow the same trajectory as soil carbon, especially under overshoot scenarios, we added the following paragraph in Section "3.1 Permafrost Response":

"Our simulations show that permafrost carbon inputs do not follow the same trajectory as soil carbon, especially under overshoot scenarios. This is likely due to inaccurate parameterization adopted in the current model. As noted in the model description (Section 2.1), litterfall is allocated to soil layers with temperatures above 1 °C according to an exponentially decreasing function of depth. When all soil layers are below 1 °C, organic carbon from the litterfall is added to the top soil layer. Meanwhile, permafrost carbon and non-permafrost soil carbon are both represented as depth-resolved carbon pools within the top six soil layers. The movement of permafrost carbon due to cryoturbation mixing is parameterized as being proportional to the gradient of total soil carbon with depth. Soil carbon that diffuses downward through the permafrost table is converted to permafrost carbon. During the

cooling phase of overshoot scenarios, increased litterfall and a rising permafrost table lead to elevated carbon concentrations in surface soil layers, resulting in enhanced vertical diffusion and a surge in permafrost carbon inputs. Conversely, under the SSP5-8.5 scenario, permafrost carbon inputs exhibit only a minor peak around the 2150s, followed by a sharp decline (Fig. 5a; Fig. S3a). This is due to the continuous reduction in permafrost area and the deepening of the permafrost table, both of which reduce carbon concentrations in the upper soil layers and weaken vertical diffusion, despite the increasing litter flux under a strong CO2 fertilization background. We note that the approach adopted in the model may not accurately reflect natural processes of vertical carbon movement, which are influenced by soil porosity heterogeneity, freeze-thaw cycles, and ice expansion upon freezing."

**Minor comments:**

Abstract: for clarity suggest reducing the use of acronyms in the abstract and possibly parts of the text.

Thank you for your suggestion. In response to your suggestion, we have reduced the use of acronyms in the abstract and replaced "SWL" and "OS" with their full terms, "stabilization" and "overshoot." We will continue to review acronym usage throughout the manuscript to ensure clarity and accessibility.

Line 25: gradual and abrupt seem to refer to the rate of carbon loss suggest revision to distinguish this from the processes of gradual and abrupt thaw.

Thank you for your suggestion. We have revised the wording to ensure that gradual and abrupt explicitly refer to the thawing processes. The sentence has been changed to "gradual or abrupt permafrost thaw, along with subsequent microbial decomposition, would release carbon dioxide (CO2) and methane (CH4) into the atmosphere...".

Line 50: suggest adding further discission of the mechanism behind the presence or absence of hysteresis behavior in different processes as this is useful background.

Thank you for this insightful suggestion. We agree that a more detailed discussion of the mechanism behind the presence or absence of hysteresis behavior in different processes would strengthen the background of our study. To address this, we have expanded the discussion in this section by highlighting key factors influencing hysteresis in permafrost carbon dynamics, including thermophysical inertia, microbial decomposition lag, and hydrological feedbacks.

The revised text now reads: "The presence or absence of hysteresis effect in the permafrost processes is influenced by multiple factors, including the thermal inertia of permafrost soils, potential shifts in vegetation composition, and the extent to which irreversible permafrost carbon losses are offset by gains in vegetation and non-permafrost soil carbon reservoirs (MacDougall, 2013; Schwinger et al., 2022). Furthermore, the soil carbon loss under overshoot scenarios

significantly affects the hydrological and thermal properties of soils (Zhu et al., 2019), which in turn modulate the processes involved. The interactions between physical and biophysical processes can potentially stabilize the carbon, water, and energy cycles at distinct post-overshoot equilibria (de Vrese and Brovkin, 2021). Therefore, a temporary warming of the permafrost regions entails important legacy effects and lasting impacts on its physical state and carbon cycle."

Line 229: Suggest clarifying the timescale being discussed in this summary information. It reads very similarly to the sentence immediately prior.

Thank you for the suggestion. We have revised this section to improve clarity and to specify the timescale. The revised manuscript is as follows:

"Notably, in all stabilization and overshoot scenarios simulated in this study, the permafrost region soil serves as a net carbon source for atmospheric CO2 by 2300. However, during the stabilization phase of OS-3 and OS-4, the permafrost region soil turns into a carbon sink, as soil carbon inputs surpass the reduced decomposition activity due to the depletion of soil carbon stocks and reduced warming levels."

**References**

Avis, C. A., Weaver, A. J., and Meissner, K. J.: Reduction in areal extent of high-latitude wetlands in response to permafrost thaw, Nature Geoscience, 4, 444–448, https://doi.org/10.1038/ngeo1160, 2011.

Chadburn, S. E., Burke, E. J., Cox, P. M., Friedlingstein, P., Hugelius, G., and Westermann, S.: An observation-based constraint on permafrost loss as a function of global warming, Nature Climate Change, 7, 340 – 344, https://doi.org/10.1038/nclimate3262, 2017.

Cox, P.: Description of the TRIFFID dynamic global vegetation model. Hadley Centre Technical., 24, 1 - 16, 2001.

De Sisto, M. L., MacDougall, A. H., Mengis, N., and Antoniello, S.: Modelling the terrestrial nitrogen and phosphorus cycle in the UVic ESCM, Geoscientific Model Development, 16, 4113 – 4136, https://doi.org/10.5194/gmd-16-4113-2023, 2023.

De Vrese, P. and Brovkin, V.: Timescales of the permafrost carbon cycle and legacy effects of temperature overshoot scenarios, Nature Communications, 12, 2688, https://doi.org/10.1038/s41467-021-23010-5, 2021.

Etminan, M., Myhre, G., Highwood, E. J., and Shine, K. P.: Radiative forcing of carbon dioxide, methane, and nitrous oxide: A significant revision of the methane radiative forcing, Geophysical Research Letters, 43, https://doi.org/10.1002/2016GL071930, 2016.

Friedlingstein, P., O' Sullivan, M., Jones, M. W., Andrew, R. M., Hauck, J., Olsen, A., Peters, G. P., Peters, W., Pongratz, J., Sitch, S., Le Quéré, C., Canadell, J. G., Ciais, P., Jackson, R. B., Alin,

S., Aragão, L. E. O. C., Arneth, A., Arora, V., Bates, N. R., Becker, M., Benoit-Cattin, A., Bittig, H. C., Bopp, L., Bultan, S., Chandra, N., Chevallier, F., Chini, L. P., Evans, W., Florentie, L., Forster, P. M., Gasser, T., Gehlen, M., Gilfillan, D., Gkritzalis, T., Gregor, L., Gruber, N., Harris, I., Hartung, K., Haverd, V., Houghton, R. A., Ilyina, T., Jain, A. K., Joetzjer, E., Kadono, K., Kato, E., Kitidis, V., Korsbakken, J. I., Landschützer, P., Lefèvre, N., Lenton, A., Lienert, S., Liu, Z., Lombardozzi, D., Marland, G., Metzl, N., Munro, D. R., Nabel, J. E. M. S., Nakaoka, S.-I., Niwa, Y., O' Brien, K., Ono, T., Palmer, P. I., Pierrot, D., Poulter, B., Resplandy, L., Robertson, E., Rödenbeck, C., Schwinger, J., Séférian, R., Skjelvan, I., Smith, A. J. P., Sutton, A. J., Tanhua, T., Tans, P. P., Tian, H., Tilbrook, B., van der Werf, G., Vuichard, N., Walker, A. P., Wanninkhof, R., Watson, A. J., Willis, D., Wiltshire, A. J., Yuan, W., Yue, X., and Zaehle, S.: Global Carbon Budget 2020, Earth System Science Data, 12, 3269 – 3340, https://doi.org/10.5194/essd-12-3269-2020, 2020.

Gasser, T., Kechiar, M., Ciais, P., Burke, E. J., Kleinen, T., Zhu, D., Huang, Y., Ekici, A., and Obersteiner, M.: Path-dependent reductions in CO2 emission budgets caused by permafrost carbon release, Nature Geoscience, 11, 830 – 835, https://doi.org/10.1038/s41561-018-0227-0, 2018.

Gulev, S. K., Thorne, P. W., Ahn, J., Dentener, F. J., Domingues, C. M., Gerland, S., Gong, D., Kaufman, D. S., Nnamchi, H. C., Quaas, J., Rivera, J. A., Sathyendranath, S., Smith, S. L., Trewin, B., von Schuckmann, K., Vose, R. S., Allan, R., Collins, B., Turner, A., and Hawkins, E.: Changing state of the climate system, edited by: Masson-Delmotte, V., Zhai, P., Pirani, A., Connors, S. L., Pé an, C., Berger, S., Caud, N., Chen, Y., Goldfarb, L., Gomis, M. I., Huang, M., Leitzell, K., Lonnoy, E., Matthews, J. B. R., Maycock, T. K., Waterfield, T., Yelekçi, O., Yu, R., and Zhou, B., Cambridge University Press, Cambridge, UK, 287 – 422, 2021.

Hugelius, G., Strauss, J., Zubrzycki, S., Harden, J. W., Schuur, E. a. G., Ping, C.-L., Schirrmeister, L., Grosse, G., Michaelson, G. J., Koven, C. D., O'Donnell, J. A., Elberling, B., Mishra, U., Camill, P., Yu, Z., Palmtag, J., and Kuhry, P.: Estimated stocks of circumpolar permafrost carbon with quantified uncertainty ranges and identified data gaps, Biogeosciences, 11, 6573–6593, https://doi.org/10.5194/bg-11-6573-2014, 2014.

Kleinen, T. and Brovkin, V.: Pathway-dependent fate of permafrost region carbon, Environmental Research Letters, 13, 094001, https://doi.org/10.1088/1748-9326/aad824, 2018.

Koven, C., Friedlingstein, P., Ciais, P., Khvorostyanov, D., Krinner, G., and Tarnocai, C.: On the formation of high-latitude soil carbon stocks: Effects of cryoturbation and insulation by organic matter in a land surface model, Geophysical Research Letters, 36, https://doi.org/10.1029/2009GL040150, 2009.

MacDougall, A. H.: Reversing climate warming by artificial atmospheric carbon-dioxide removal: Can a Holocene-like climate be restored? Geophysical Research Letters, 40, 5480 - 5485, https://doi.org/10.1002/2013GL057467, 2013.

MacDougall, A. H., Zickfeld, K., Knutti, R., and Matthews, H. D.: Sensitivity of carbon budgets to permafrost carbon feedbacks and non-CO2 forcings, Environmental Research Letters, 10, 125003, https://doi.org/10.1088/1748-9326/10/12/125003, 2015.

MacDougall, A. H. and Knutti, R.: Projecting the release of carbon from permafrost soils using a perturbed parameter ensemble modelling approach, Biogeosciences, 13, 2123–2136, https://doi.org/10.5194/bg-13-2123-2016, 2016.

MacDougall, A. H.: Estimated effect of the permafrost carbon feedback on the zero emissions commitment to climate change, Biogeosciences, 18, 4937–4952, https://doi.org/10.5194/bg-18-4937-2021, 2021.

McKay, M. D., Beckman, R. J., and Conover, W. J.: A Comparison of Three Methods for Selecting Values of Input Variables in the Analysis of Output from a Computer Code, Technometrics, 21, 239 - 245, https://doi.org/10.2307/1268522, 1979.

Meissner, K. J., Weaver, A. J., Matthews, H. D., and Cox, P. M.: The role of land surface dynamics in glacial inception: a study with the UVic Earth System Model, Climate Dynamics, 21, 515–537, https://doi.org/10.1007/s00382-003-0352-2, 2003.

Mengis, N., Keller, D. P., MacDougall, A. H., Eby, M., Wright, N., Meissner, K. J., Oschlies, A., Schmittner, A., MacIsaac, A. J., Matthews, H. D., and Zickfeld, K.: Evaluation of the University of Victoria Earth System Climate Model version 2.10 (UVic ESCM 2.10), Geoscientific Model Development, 13, 4183–4204, https://doi.org/10.5194/gmd-13-4183-2020, 2020.

Natali, S. M., Holdren, J. P., Rogers, B. M., Treharne, R., Duffy, P. B., Pomerance, R., and MacDonald, E.: Permafrost carbon feedbacks threaten global climate goals, Proceedings of the National Academy of Sciences, 118, e2100163118, https://doi.org/10.1073/pnas.2100163118, 2021.

Pearson, R. G., Phillips, S. J., Loranty, M. M., Beck, P. S. A., Damoulas, T., Knight, S. J., and Goetz, S. J.: Shifts in Arctic vegetation and associated feedbacks under climate change, Nature Climate Change, 3, 673 – 677, https://doi.org/10.1038/nclimate1858, 2013.

Schuur, E. a. G., McGuire, A. D., Schädel, C., Grosse, G., Harden, J. W., Hayes, D. J., Hugelius, G., Koven, C. D., Kuhry, P., Lawrence, D. M., Natali, S. M., Olefeldt, D., Romanovsky, V. E., Schaefer, K., Turetsky, M. R., Treat, C. C., and Vonk, J. E.: Climate change and the permafrost carbon feedback, Nature, 520, 171 - 179, https://doi.org/10.1038/nature14338, 2015.

Schwinger, J., Asaadi, A., Steinert, N. J., and Lee, H.: Emit now, mitigate later? Earth system reversibility under overshoots of different magnitudes and durations, Earth System Dynamics, 13, 1641 – 1665, https://doi.org/10.5194/esd-13-1641-2022, 2022.

Strauss, J., Schirrmeister, L., Grosse, G., Fortier, D., Hugelius, G., Knoblauch, C., Romanovsky, V., Schädel, C., Schneider von Deimling, T., Schuur, E. A. G., Shmelev, D., Ulrich, M., and Veremeeva, A.: Deep Yedoma permafrost: A synthesis of depositional characteristics and carbon vulnerability, Earth-Science Reviews, 172, 75 – 86, https://doi.org/10.1016/j.earscirev.2017.07.007, 2017.

Strauss, J., Laboor, S., Schirrmeister, L., Fedorov, A. N., Fortier, D., Froese, D., Fuchs, M., Günther, F., Grigoriev, M., Harden, J., Hugelius, G., Jongejans, L. L., Kanevskiy, M., Kholodov, A., Kunitsky,

V., Kraev, G., Lozhkin, A., Rivkina, E., Shur, Y., Siegert, C., Spektor, V., Streletskaya, I., Ulrich, M., Vartanyan, S., Veremeeva, A., Anthony, K. W., Wetterich, S., Zimov, N., and Grosse, G.: Circum-Arctic Map of the Yedoma Permafrost Domain, Frontiers in Earth Science, 9, 758360, https://doi.org/10.3389/feart.2021.758360, 2021.

Strauss, J., Laboor, S., Schirrmeister, L., Fedorov, A. N., Fortier, D., Froese, D. G., Fuchs, M., Günther, F., Grigoriev, M. N., Harden, J. W., Hugelius, G., Jongejans, L. L., Kanevskiy, M. Z., Kholodov, A. L., Kunitsky, V., Kraev, G., Lozhkin, A. V., Rivkina, E., Shur, Y., Siegert, C., Spektor, V., Streletskaya, I., Ulrich, M., Vartanyan, S. L., Veremeeva, A., Walter Anthony, K. M., Wetterich, S., Zimov, N. S., and Grosse, G.: Database of Ice-Rich Yedoma Permafrost Version 2 (IRYP v2), PANGAEA, https://doi.org/10.1594/PANGAEA.940078, 2022.

Tokarska, K. B. and Zickfeld, K.: The effectiveness of net negative carbon dioxide emissions in reversing anthropogenic climate change, Environmental Research Letters, 10, 094013, https://doi.org/10.1088/1748-9326/10/9/094013, 2015.

Zhang, T., Heginbottom ,J. A., Barry ,R. G., and Brown, J.: Further statistics on the distribution of permafrost and ground ice in the Northern Hemisphere 1, Polar Geography, 24, 126 – 131, https://doi.org/10.1080/10889370009377692, 2000.

Zimov, S. A., Schuur, E. A. G., and Chapin, F. S.: Permafrost and the Global Carbon Budget, Science, 312, 1612 – 1613, https://doi.org/10.1126/science.1128908, 2006.

**Referee #2**

**Overall:**

This study "aims to fill these gaps using an Earth system model of intermediate complexity to systematically assess the permafrost response and feedback under temperature stabilization or overshoot scenarios achieving various GWLs." I think it is an interesting paper and recommend publication, but I also think it could make some clearer points, as I discuss below.

Based on that stated aim, I expected to see one or more figures with total permafrost carbon losses plotted as a function of the GWL for both the stabilization and overshoot cases. I.e., is the permafrost carbon loss linear? Are there thresholds or tipping points? Figure 6a shows the areal loss as a function of global warming level, but why are carbon variables not quantified in this way? Does the permafrost carbon feedback strength (in units of Pg C / degree Celsius warming) show a similar nonlinearity as the SPAW shown in f.g 6a with maximum losses per unit warming in the 1.5-2 degree C range? Figure 3b seems to show that the highest sensitivity is in the ~3 degree warming range, but it is difficult to see quantitatively. Likewise it would be interesting to se the radiative forcing as well. So I'd recommend an additional figure with panels along the lines of 6a that allows the reader to trace how the (non-)linearity of each of these permafrost metrics as a function of global warming levels for the stabilization and overshoot cases changes between permafrost area, permafrost carbon, and permafrost radiative forcing.

Thank you for your insightful suggestion. We fully agree that using global warming levels (GWLs) as the horizontal axis to present key permafrost metrics helps reveal their linear or nonlinear behavior. In response, we have added a new Section, "3.3 Linearity of Permafrost Response and Feedback", along with a new figure (Figure 10) to the revised manuscript. This figure includes three panels showing (a) permafrost area loss, (b) permafrost carbon loss, and (c) permafrost radiative forcing, all plotted as a function of GWLs. To further explore the potential for thresholds or tipping points, we have also added Figure 8 and corresponding text to examine the evolution of the permafrost feedback factor across scenarios.

Figure 10. Relationship between global warming levels and three permafrost metrics: (a) permafrost area loss, (b) permafrost carbon loss, and (c) permafrost radiative forcing in the stabilization (colored dashed lines) and overshoot (colored solid lines) scenarios at 1.5 °C (blue), 2.0 °C (green), 3.0 °C (red) and 4.0 °C (purple) global warming levels, along with the SSP5-8.5 scenario (black).

Square and circle markers indicate values in the year 2300 for the stabilization and overshoot scenarios, respectively. All results are based on the PFC simulations. Grey solid lines show linear fits of permafrost metrics to global warming levels in stabilization scenarios by 2300, while black dashed lines show corresponding fits for the SSP5-8.5 scenario. Note that in panel (a), both the stabilization scenarios and the corresponding SSP5-8.5 points included in the linear fit are limited to global warming levels between 1.5 °C and 3.0 °C, whereas in panels (b) and (c), the fits include points with global warming levels ranging from 1.5 °C to 4.0 °C. For stabilization scenarios, only the results from the year 2300 are used for fitting, while for the SSP5-8.5 scenario, all results within the specified global warming level ranges are used for fitting. Shaded regions represent the 5th to 95th percentile ranges across 250 ensemble simulations.

The new Section "3.3 Linearity of Permafrost Response and Feedback" is as following:

"After exploring the response and feedback of permafrost under temperature stabilization and overshoot scenarios at various global warming levels, it is natural to question the (non-)linearity of these response and feedback as functions of global warming levels. Our results show that the responses of permafrost area, permafrost carbon feedback and associated radiative forcing to a broad range of global warming are nearly linear (Fig. 10). The permafrost area change exhibits a strongly nonlinear relationship with global warming below 1.5 °C level, then a quasilinear relation between them in the global warming ranges from 1.5 °C to 3 °C. Above 3 °C global warming, the sensitivity of permafrost area to global warming decreases nonlinearly, and it is evident in both stabilization and SSP5-8.5 scenarios (Fig. 10a). In contrast, permafrost carbon loss and associated radiative forcing exhibit a nearly linear response to increasing global warming levels, especially above 1 °C, for both stabilization and SSP5-8.5 scenarios (Fig. 10b, c)."

"Meanwhile, the sensitivities of permafrost area, permafrost carbon loss, and associated radiative forcing to global warming under stabilization scenarios are all stronger than those under the SSP5-8.5 scenario. Specifically, based on the simulated permafrost area in the year 2300 under stabilization scenarios with global warming levels between 1.5 °C and 3 °C, the sensitivity of permafrost area to global warming is -3.19 [-3.01 to -3.36] million km2 °C-1. In comparison, a linear fit of permafrost area change against global warming levels over the same temperature range in the SSP5-8.5 scenario yields a sensitivity of -2.85 [-2.77 to -2.89] million km2 °C-1. Similarly, the permafrost carbon feedback per degree of global warming derived from a linear fit based on the total permafrost carbon loss in the year 2300 under stabilization scenarios, is -27.6 [-16.5 to -38.2] PgC °C-1. In contrast, the corresponding value under the SSP5-8.5 scenario, estimated from a linear fit over the 1.5 °C to 4.0 °C warming range, is -19.3 [-15.7 to -24.1] PgC °C-1. Applying the same approach, the associated radiative forcing per degree of global warming is estimated to be 0.08 [0.05 to 0.12] W m-2 °C-1 for the stabilization scenarios and 0.04 [0.03 to 0.05] W m-2 °C-1 for the SSP5-8.5. These differences between the stabilization and SSP5-8.5 scenarios are mainly attributable to the differing response time scales represented by the two scenarios: SSP5-8.5 reflects a typical transient response, while the stabilization scenarios maintain stabilized temperatures over extended periods and thus approximate a quasi-equilibrium response of the climate-carbon system. Furthermore, the smaller sensitivity of permafrost radiative forcing per degree of global warming under the SSP5-8.5 can be partially attributed to its higher background atmospheric CO2 concentration compared to the stabilization scenarios. The same amount of CO2 emissions would produce smaller additional radiative forcing under a higher background atmospheric CO2 concentration, due to the logarithmic relationship between CO2 concentration and radiative forcing (Etminan et al., 2016)."

"To a certain extent, our findings align with those of Nitzbon et al. (2024), who suggested that the accumulated response of Arctic permafrost to climate warming is approximately quasilinear. Nitzbon et al. (2024) reported a quasilinear decrease in the equilibrium permafrost extent to global warming, with a rate of approximately 3.5 million km² °C¹. This quasilinear relation holds for global warming ranges from 0 °C to 4 °C, derived from the empirical relationship between the local permafrost fraction and the annual mean global temperature. However, our results indicate the quasilinear relationship only holds for global warming levels between 1.5 °C and 3 °C. Furthermore, the permafrost carbon feedback and the associated radiative forcing per degree of warming, as derived from our simulations of both stabilization and SSP5-8.5 scenarios, are within the ranges of -18 [-3.1 to -41] PgC °C¹ and 0.09 [0.02 to 0.20] W m² °C¹, respectively, reported by Canadell et al. (2021). Our estimates also align with the estimated range of equilibrium sensitivity of permafrost carbon decline to global warming, which is -21 [-4 to -48] PgC °C¹. This may represent an upper limit for permafrost carbon feedback per degree of global warming, considering that the estimated reduction in permafrost carbon does not equate directly to carbon emissions released into the atmosphere, as noted by Nitzbon et al. (2024)."

"Under overshoot scenarios, permafrost area responds nearly reversibly and presents an almost closed loop (Fig. 10a). In contrast, permafrost carbon loss exhibits an open loop with respect to global warming levels. In other words, permafrost carbon loss does not reverse as temperatures decline, indicating irreversible permafrost carbon radiative forcing. Among the three metrics investigated here, only permafrost area exhibits strong reversibility under the overshoot scenarios. This also explains why, in Fig. 9a, the permafrost area sensitivity derived from the SSP5-8.5 scenario, when multiplied by additional warming, can reasonably reconstruct permafrost area loss in the stabilization and overshoot cases."

Figure 8 and corresponding text has been added to Section "3.2 Radiative Impacts of Permafrost Carbon Release" as follows:

Figure 8. Timeseries of permafrost feedback factor under stabilization (dashed lines) and overshoot (solid lines) scenarios at 1.5 °C (blue), 2.0 °C (green), 3.0 °C (red), and 4.0 °C (purple) global warming levels, as well as the SSP5-8.5 scenario. Square markers indicate the time points when the temperature overshoot reaches its peak or stabilized warming begins, while circle markers indicate when the overshoot returns to 1.5 °C. Results represent the ensemble median of 250 simulations. Dots on the right panels represent values in the year 2300, with uncertainty ranges estimated as the 5th to 95th percentiles. The permafrost feedback factor is calculated as the ratio of additional global warming caused by the permafrost carbon feedback (i.e., the difference between the PFC and NPFC simulations) to the global mean temperature change in the NPFC simulations.

"The additional warming caused by permafrost carbon release can be utilized to assess whether the permafrost carbon feedback could be classified as a global tipping point process. This means it is not only positive but also sufficiently strong to sustain itself. To qualify, an initial rise in global mean temperature would need to trigger permafrost carbon emissions that result in a further increase in global mean temperature surpassing the initial warming. As a result, the positive permafrost carbon feedback would induce sufficient additional thawing to initiate a self-sustaining feedback loop (Nitzbon et al., 2024). We employed the permafrost feedback factor, which is defined as the ratio of the additional warming to the initial warming simulated with the permafrost carbon module deactivated, to determine if the permafrost carbon feedback can be considered as a global tipping process. In all perturbed parameter ensemble simulations for the stabilization, overshoot and SSP5-8.5 scenarios, the maximum permafrost feedback factor is 0.21 °C °C-1. By 2300, the permafrost feedback factor for the OS4 and SSP5-8.5 scenarios are estimated at 0.12 [0.08 to 0.17] °C °C-1 and 0.02 [0.01-0.03] °C °C-1, respectively. The permafrost feedback parameter is the highest under the OS4 scenario, while it is the lowest under the SSP5-8.5 scenario (Fig. 8; Fig. S5). Interestingly, the feedback factors are quite similar across the stabilization scenarios, with values of 0.064 [0.037 to 0.096] °C °C-1, 0.064 [0.036 to 0.095] °C °C-1, 0.069 [0.040 to 0.103] °C °C-1 and 0.061 [0.038 to 0.089] °C °C-1 for the SWL-1.5, SWL-2, SWL-3 and SWL-4 scenarios by 2300, respectively. Although the feedback factor in the overshoot scenarios is substantially larger than the recent estimate of 0.035 (0.004-0.110) °C °C-1 based on the Sixth Assessment Report of the Intergovernmental Panel on Climate Change (Nitzbon et al., 2024), our findings indicate that the positive permafrost carbon feedback is unlikely to result in enough additional thawing and corresponding carbon emissions to initiate a self-perpetuating tipping process. Since this study only models the gradual thawing of permafrost through the deepening of the active layer, we cannot rule out the possibility of tipping points associated with the abrupt thawing of talik development, thermokarst and thermo-erosion processes."

Further, given the possibility of perturbing parameters due to the relatively low cost of running UVic-ESCM, I had expected to see if any of those parameters introduced nonlinearities or substantially changed the magnitude of the results. But I just see median lines. So it is hard to know how important the uncertainty is. I suggest showing the uncertainty via translucent colored plumes in all figures.

We sincerely appreciate your valuable suggestion. We fully acknowledge the importance of showing the uncertainty to accurately convey the results. In the revised manuscript, we have explicitly represented the 5th to 95th percentile of 250 ensemble simulations for the year 2300 in all relevant

figures to help readers better understand the uncertainty associated with parameter perturbations. For example, Figures 3, 5 and 7 now include uncertainty ranges using vertical bars. In addition, we have provided translucent colored plumes in the Supplementary Information (Figure S1–S5) to illustrate the full time-evolving uncertainty across all ensemble members.

The updated figures with uncertainty ranges are shown below:

Figure 3. Timeseries of annual mean (a) global warming, (b) permafrost area, (c) permafrost carbon loss and (d) permafrost region soil carbon loss under stabilization (dashed lines) and overshoot (solid lines) scenarios at 1.5 °C (blue), 2.0 °C (green), 3.0 °C (red), and 4.0 °C (purple) global warming levels, as well as the SSP5-8.5 scenario. Square markers indicate the time points when the temperature overshoot reaches its peak or stabilized warming begins, while circle markers indicate when the overshoot returns to 1.5 °C. All changes are relative to the pre-industrial period (1850-1900). Results represent the ensemble median of 250 simulations based on the PFC simulations. Dots on the right panels represent values in the year 2300, with uncertainty ranges estimated as the 5th to 95th percentiles.

Figure 5. Timeseries of changes in (a) permafrost carbon inputs, (b) permafrost region soil carbon inputs, (c) permafrost carbon decomposition and (d) permafrost region soil carbon decomposition, under the stabilization (dashed lines) and overshoot (solid lines) scenarios at 1.5 °C (blue), 2.0 °C (green), 3.0 °C (red) and 4.0 °C (purple) GWLs, along with the SSP5-8.5 scenario (black). Square markers indicate the time points when the temperature overshoot reaches its peak or stabilized warming begins, while circle markers indicate when the overshoot returns to 1.5 °C. Results represent the ensemble median of 250 simulations based on the PFC simulations. Dots on the right panels represent values in the year 2300, with uncertainty ranges estimated as the 5th to 95th percentiles.

Figure 7. Additional changes in (a) radiative forcing, (b) global warming and (c, d) permafrost area due to permafrost carbon feedback, calculated as the difference between the PFC and NPFC simulations. Shown are results for the stabilization (dashed lines) and overshoot (solid lines) scenarios at 1.5 °C (blue), 2.0 °C (green), 3.0 °C (red) and 4.0 °C (purple) global warming levels, along with the SSP5-8.5 scenario (black). Square markers indicate the time points when the temperature overshoot reaches its peak or stabilized warming begins, while circle markers indicate when the overshoot returns to 1.5 °C. Results represent the ensemble median of 250 simulations. Dots on the right panels represent values in the year 2300, with uncertainty ranges estimated as the 5th to 95th percentiles. In panel (a), the additional radiative forcing is calculated using the simplified expressions (Etminan et al., 2016) based on simulated CO2 concentrations. In panels (c) and (d), the additional permafrost area loss is smoothed using a 5-year rolling average to eliminate interannual variability.

**Comments**

line 36: the 1.5 degree budget will be exhausted within the next few years, but not the 2 degree budget. Please clarify.

Thank you for your suggestion. We have revised the sentence for greater clarity:

"If current emission rates persist, the remaining carbon budgets compatible with the 1.5 °C target will be critically tight and likely exhausted within the next few years (Rogelj et al., 2015; Goodwin et al., 2018; Masson-Delmotte et al., 2018; Forster et al., 2023; Smith et al., 2023)."

Paragraph starting line 142: This is great that you were able to perturb these key parameters. But I don't see any uncertainty plumes in any of the figures, only the median values. I think it would be informative to the reader to see the parameter uncertainty plumes plotted on all figures.

Thank you for your comment. As noted above, we have incorporated the 5th to 95th percentiles of 250 ensemble simulations for the year 2300 in all relevant figures. Additionally, time-evolving uncertainty associated with the perturbed parameters is illustrated using translucent colored plumes in the Supplementary Information (Figs. S1–S5).

fig. 2b: Why doesn't the permafrost area recover all the way under the overshoot scenarios? Are there regional changes to the northern high latitude climate that are responsible for the differing permafrost amounts at a given GWL? If so, what are the drivers of that regional change? It might help to add a panel with the regional temperature difference to see whether it behaves differently from the global mean.

Thank you for your valuable comments. In response, we plotted a new figure (Figure R2) to quantify additional warming in permafrost regions and to examine the regional amplification relative to the global mean. We added the following paragraph in Section "4 Conclusion and Discussion" to explain why the permafrost area does not recover all the way under the overshoot scenarios:

"Our results show incomplete recovery of permafrost area under the overshoot scenarios, which is influenced by multiple factors: First, the additional permafrost carbon release leads to greater additional warming under the overshoot scenarios than the SWL-1.5 scenario, causing additional permafrost degradation. By 2300, the northern high-latitude permafrost regions are 0.01~0.13 °C warmer compared to the SWL-1.5 scenario. Second, the thermal inertia of deep soil layers limits the rate of permafrost recovery. Even after global mean temperatures return to the 1.5 °C target, residual heat accumulated in deeper soil layers during temperature overshoot period continues to inhibit permafrost refreezing, preventing full restoration to its pre-overshoot state. Third, greater soil carbon loss under overshoot scenarios substantially alters the hydrological and thermal properties of soil, affecting the processes that govern carbon cycling (Zhu et al., 2019; Avis, 2012; Lawrence and Slate, 2008), which in turn affects the recovery of permafrost area. Moreover, irreversible shifts in vegetation composition of high-latitude terrestrial ecosystems also contribute to the incomplete recovery of permafrost area under overshoot scenarios. For instance, among the two dominant vegetation types, needleleaf trees continue to expand while C3 grasses decline, even after global temperatures return to the 1.5 °C warming level. These irreversible changes may stabilize the carbon, water, and energy cycles over the permafrost region at different equilibria after overshoot, through the interactions between physical and biophysical processes (de Vrese and Brovkin, 2021), thereby constraining the ability of permafrost to fully recover under the overshoot scenarios."

Figure R2. Timeseries of changes relative to the SWL-1.5 scenario for the overshoot scenarios at 2.0 °C (green), 3.0 °C (red), and 4.0 °C (purple) global warming levels. Panel (a) shows the additional warming in permafrost regions, calculated as the difference between the PFC and NPFC simulations. Panel (b) shows the regional amplification of warming, defined as the difference between additional warming in permafrost regions and the corresponding additional global warming.

Line 321: This paper doesn't really establish anything about the realism of the model, since there are no model-data comparisons, so suggest reword or provide citations to the papers that have shown this.

Thank you for your valuable comment. We acknowledge the need for additional clarification regarding the realism of the UVic ESCM model. To address this, we have incorporated model validation into the first paragraph of Section "3 Results". This addition provides a direct comparison

between model simulations and observational data, demonstrating that the UVic ESCM model realistically reproduces historical permafrost area and permafrost carbon stocks. The added paragraph reads:

"The UVic ESCM v2.10 reliably simulates historical temperature changes, permafrost area, and the partitioning of anthropogenic carbon emissions among the atmosphere, ocean and land. Under preindustrial conditions, the simulated Northern Hemisphere permafrost area, defined as regions where the soil layer remains perennially frozen for at least two consecutive years, was 17.01 [17.00 to 17.04] million km2, the simulated total soil carbon stock in the permafrost regions was 1031 [915 to 1149] PgC, of which 484 [383 to 590] PgC was classified as perennially frozen carbon and 547 [533 to 559] PgC was classified as usual soil carbon. For the period 1960–1990, the model simulated Northern Hemisphere permafrost area at 16.8 [16.7 to 16.9] million km2, which falls within the reconstructed range from 12.0 to 18.2 million km2 (Chadburn et al., 2017) and the observation derived extent from 12.21 to 16.98 million km2 (Zhang et al., 2000). Additionally, the simulated soil carbon stock in the top 3.35 m of permafrost regions for this same period was 1034 [919 to 1151] PgC, with 483 [382 to 587] PgC classified as perennially frozen carbon, accounting for 47% [42% to 51%] of the total permafrost soil carbon stock, in agreement with Hugelius et al. (2014). During the period 2011-2020, the model estimated a global mean temperature increase of 1.14 [1.13 to 1.15] °C relative to preindustrial levels, which is closely aligned with the observed rise of 1.09 [0.91 to 1.23] °C (Gulev et al., 2021). From 2010 to 2019, the model estimated that anthropogenic carbon emissions were distributed as follows: 5.5 [5.4 to 5.6] PgC yr-1 to the atmosphere, 3.0 [2.98 to 3.03] PgC yr-1 to the ocean, and 2.5 [2.4 to 2.6] PgC yr-1 to terrestrial ecosystems. These estimates are broadly consistent with the global anthropogenic CO2 budget assessment by the Global Carbon Project (GCP) with figures of 5.1±0.02 PgC yr-1 for the atmosphere, 2.5±0.6 PgC yr-1 for the ocean, and 3.4±0.9 PgC yr-1 for terrestrial ecosystems (Friedlingstein et al., 2020)."

Accordingly, we have revised the statement in Line 321 to: "The UVic ESCM has been validated against observational and reconstructed datasets, demonstrating its ability to reproduce historical permafrost area and permafrost carbon stocks."

Data availability: I downloaded some of the data files in Cui et al., 2024, but they aren't clearly described and don't include any further details than what is in the paper (e.g., spatial information). This strikes me as a fairly minimal data archival effort.

We appreciate your comments regarding the data archiving. We have added more detailed descriptions to the uploaded data files, including variable names, units, and associated spatial and temporal dimensions, in order to improve the clarity.

Due to the large volume of spatial model output, we have archived only the key variables necessary to support the main analyses presented in the paper (https://zenodo.org/records/15148252). Although this may not capture all details, we are happy to provide more comprehensive datasets upon request. In addition, a data description file (README.md) has been included alongside the archived model output to facilitate understanding.

**References**

Avis, C. A., Weaver, A. J., and Meissner, K. J.: Reduction in areal extent of high-latitude wetlands in response to permafrost thaw, Nature Geoscience, 4, 444–448, https://doi.org/10.1038/ngeo1160, 2011.

Canadell, J., Forster, P., Meyer, C., and the Chapter 5 authors: Global carbon and other biogeochemical cycles and feedbacks, in: Climate Change 2021: The Physical Science Basis. Contribution of Working Group I to the Sixth Assessment Report of the Intergovernmental Panel on Climate Change, edited by: Masson-Delmotte, V., Zhai, P., Pirani, A., et al., Cambridge University Press, Cambridge, United Kingdom and New York, NY, USA, Chapter 5, https://doi.org/10.1017/9781009157896.007, 2021.

Chadburn, S. E., Burke, E. J., Cox, P. M., Friedlingstein, P., Hugelius, G., and Westermann, S.: An observation-based constraint on permafrost loss as a function of global warming, Nature Climate Change, 7, 340 – 344, https://doi.org/10.1038/nclimate3262, 2017.

De Vrese, P. and Brovkin, V.: Timescales of the permafrost carbon cycle and legacy effects of temperature overshoot scenarios, Nature Communications, 12, 2688, https://doi.org/10.1038/s41467-021-23010-5, 2021.

Etminan, M., Myhre, G., Highwood, E. J., and Shine, K. P.: Radiative forcing of carbon dioxide, methane, and nitrous oxide: A significant revision of the methane radiative forcing, Geophysical Research Letters, 43, https://doi.org/10.1002/2016GL071930, 2016.

Forster, P. M., Smith, C. J., Walsh, T., Lamb, W. F., Lamboll, R., Hauser, M., Ribes, A., Rosen, D., Gillett, N., Palmer, M. D., Rogelj, J., von Schuckmann, K., Seneviratne, S. I., Trewin, B., Zhang, X., Allen, M., Andrew, R., Birt, A., Borger, A., Boyer, T., Broersma, J. A., Cheng, L., Dentener, F., Friedlingstein, P., Gutiérrez, J. M., Gütschow, J., Hall, B., Ishii, M., Jenkins, S., Lan, X., Lee, J.-Y., Morice, C., Kadow, C., Kennedy, J., Killick, R., Minx, J. C., Naik, V., Peters, G. P., Pirani, A., Pongratz, J., Schleussner, C.-F., Szopa, S., Thorne, P., Rohde, R., Rojas Corradi, M., Schumacher, D., Vose, R., Zickfeld, K., Masson-Delmotte, V., and Zhai, P.: Indicators of Global Climate Change 2022: annual update of large-scale indicators of the state of the climate system and human influence, Earth System Science Data, 15, 2295 – 2327, https://doi.org/10.5194/essd-15-2295-2023, 2023.

Friedlingstein, P., O' Sullivan, M., Jones, M. W., Andrew, R. M., Hauck, J., Olsen, A., Peters, G. P., Peters, W., Pongratz, J., Sitch, S., Le Quéré, C., Canadell, J. G., Ciais, P., Jackson, R. B., Alin, S., Aragão, L. E. O. C., Arneth, A., Arora, V., Bates, N. R., Becker, M., Benoit-Cattin, A., Bittig, H. C., Bopp, L., Bultan, S., Chandra, N., Chevallier, F., Chini, L. P., Evans, W., Florentie, L., Forster, P. M., Gasser, T., Gehlen, M., Gilfillan, D., Gkritzalis, T., Gregor, L., Gruber, N., Harris, I., Hartung, K., Haverd, V., Houghton, R. A., Ilyina, T., Jain, A. K., Joetzjer, E., Kadono, K., Kato, E., Kitidis, V., Korsbakken, J. I., Landschützer, P., Lefèvre, N., Lenton, A., Lienert, S., Liu, Z., Lombardozzi, D., Marland, G., Metzl, N., Munro, D. R., Nabel, J. E. M. S., Nakaoka, S.-I., Niwa, Y., O' Brien, K., Ono, T., Palmer, P. I., Pierrot, D., Poulter, B., Resplandy, L., Robertson, E., Rödenbeck, C., Schwinger, J., Séférian, R., Skjelvan, I., Smith, A. J. P., Sutton, A. J., Tanhua, T., Tans, P. P., Tian, H., Tilbrook, B., van der Werf, G., Vuichard, N., Walker, A. P., Wanninkhof, R., Watson, A. J., Willis,

D., Wiltshire, A. J., Yuan, W., Yue, X., and Zaehle, S.: Global Carbon Budget 2020, Earth System Science Data, 12, 3269 - 3340, https://doi.org/10.5194/essd-12-3269-2020, 2020.

Goodwin, P., Katavouta, A., Roussenov, V. M., Foster, G. L., Rohling, E. J., and Williams, R. G.: Pathways to 1.5 ° C and 2 ° C warming based on observational and geological constraints, Nature Geoscience, 11, 102 – 107, https://doi.org/10.1038/s41561-017-0054-8, 2018.

Gulev, S. K., Thorne, P. W., Ahn, J., Dentener, F. J., Domingues, C. M., Gerland, S., Gong, D., Kaufman, D. S., Nnamchi, H. C., Quaas, J., Rivera, J. A., Sathyendranath, S., Smith, S. L., Trewin, B., von Schuckmann, K., Vose, R. S., Allan, R., Collins, B., Turner, A., and Hawkins, E.: Changing state of the climate system, edited by: Masson-Delmotte, V., Zhai, P., Pirani, A., Connors, S. L., Pé an, C., Berger, S., Caud, N., Chen, Y., Goldfarb, L., Gomis, M. I., Huang, M., Leitzell, K., Lonnoy, E., Matthews, J. B. R., Maycock, T. K., Waterfield, T., Yelekçi, O., Yu, R., and Zhou, B., Cambridge University Press, Cambridge, UK, 287 – 422, 2021.

Hugelius, G., Strauss, J., Zubrzycki, S., Harden, J. W., Schuur, E. a. G., Ping, C.-L., Schirrmeister, L., Grosse, G., Michaelson, G. J., Koven, C. D., O'Donnell, J. A., Elberling, B., Mishra, U., Camill, P., Yu, Z., Palmtag, J., and Kuhry, P.: Estimated stocks of circumpolar permafrost carbon with quantified uncertainty ranges and identified data gaps, Biogeosciences, 11, 6573–6593, https://doi.org/10.5194/bg-11-6573-2014, 2014.

Lawrence, D. M. and Slater, A. G.: Incorporating organic soil into a global climate model, Clim Dyn, 30, 145 - 160, https://doi.org/10.1007/s00382-007-0278-1, 2008.

Masson-Delmotte, V., Zhai, P., Pörtner, H.-O., Roberts, D. C., Skea, J., Shukla, P. R., Pirani, A., Moufouma-Okia, W., Péan, C., Pidcock, R., Connors, S., Matthews, J. B. R., Yang, C., Zhou, X., and Steg, L.: Global warming of 1.5 ° C: Summary for policy makers, IPCC - The Intergovernmental Panel on Climate Change, https://doi.org/10.1017/9781009157940.001, 2018.

Nitzbon, J., Schneider Von Deimling, T., Aliyeva, M., Chadburn, S. E., Grosse, G., Laboor, S., Lee, H., Lohmann, G., Steinert, N. J., Stuenzi, S. M., Werner, M., Westermann, S., and Langer, M.: No respite from permafrost-thaw impacts in the absence of a global tipping point, Nature Climate Change, 14, 573 – 585, https://doi.org/10.1038/s41558-024-02011-4, 2024.

Rogelj, J., Meinshausen, M., Schaeffer, M., Knutti, R., and Riahi, K.: Impact of short-lived non-CO2 mitigation on carbon budgets for stabilizing global warming, Environmental Research Letters, 10, 075001, https://doi.org/10.1088/1748-9326/10/7/075001, 2015.

Smith, S., Geden, O., Nemet, G., Gidden, M., Lamb, W., Powis, C., Bellamy, R., Callaghan, M., Cowie, A., Cox, E., Fuss, S., Gasser, T., Grassi, G., Greene, J., Lueck, S., Mohan, A., Müller-Hansen, F., Peters, G., Pratama, Y., Repke, T., Riahi, K., Schenuit, F., Steinhauser, J., Strefler, J., Valenzuela, J., and Minx, J.: State of Carbon Dioxide Removal - 1st Edition, https://doi.org/10.17605/OSF.IO/W3B4Z, 2023.

Zhang, T., Heginbottom ,J. A., Barry ,R. G., and Brown, J.: Further statistics on the distribution of permafrost and ground ice in the Northern Hemisphere 1, Polar Geography, 24, 126 – 131, https://doi.org/10.1080/10889370009377692, 2000.

**Editor**

Dear authors,

Thank you for detailed response to reviewers' comments. I have now read through your comments. Please go ahead and incorporate reviewers' comments in revising your manuscript as you have indicated in your response. Based on my reading, I may not send the revised manuscript back to reviewers for their second opinion.

I also have given your manuscript a thorough read and have some comments of my own. As you revise your manuscript please address following minor comments as well.

1) Please clarify if permafrost table is the same as the active layer depth or not.

Yes. In the UVic ESCM v2.10 model, the depth of the permafrost table corresponds to the active layer depth, which is defined as the shallowest depth at which the soil remains frozen for at least two consecutive years. We have clarified this point in the revised manuscript.

2) Your translucent colour plots (in response to reviewer #2) can go in supplementary information. Despite the overlaps, I found them helpful. It's your decision.

Thank you for your suggestion. We have included the translucent colour plots in the supplementary materials.

3) Please consider introducing your stabilization and overshoot scenarios briefly in the abstract to provide some context before delving into results.

Thank you for your detailed comments. We have provided a brief introduction of the stabilization and overshoot scenarios in the revised abstract to offer essential context.

4) In the abstract, when you say 4.5 to 6.5 million km2 of permafrost is lost, please also considering mentioning the model simulated pre-industrial permafrost extent for context.

Thanks. The model simulated pre-industrial permafrost extent is 17.01 million km2, and this information has been incorporated into the revised abstract to provide context for the projected permafrost loss.

5) Mention units of SPAW in the abstract.

Thanks. We have removed the abbreviation "SPAW" from the abstract to reduce the use of acronyms in the abstract. Its unit (million  $km^2$  °C-1) is now explicitly stated in the main text where the concept is first introduced.

6) Lines 62 and 63, clarifying the difference between response and feedback will be helpful.

Thank you for your suggestion. To clarify the distinction between response and feedback, we have revised "...how permafrost carbon response and feedback under temperature stabilization scenarios..." to "...how permafrost carbon will be released and further amplify global warming under temperature stabilization scenarios...".

7) If the model soil depth went down to say 40 m, will cryo-turbation spin up Yedoma. Likely not. Can you please add a sentence to make this clear?

Thank you for your suggestion. While the UVic ESCM v2.10 model resolves soil depth down to 250.3 m, it simulates both usual and permafrost soil carbon only within the top six layers, extending to a depth of 3.35 m. Consequently, cryo-turbation processes cannot initiate the formation of Yedoma under the current model configuration. We have included a clarification to make this aspect explicit in the revised manuscript.

8) Please note the size of pre-industrial usual and frozen soil C pools from your spin up.

Thanks. Under pre-industrial conditions, the simulated total soil carbon stock in the permafrost regions was 1031 [915 to 1149] PgC, of which 484 [383 to 590] PgC was classified as perennially frozen carbon and 547 [533 to 559] PgC was classified as usual soil carbon.

9) Since your runs aim to achieve a certain temperature threshold, your simulations are emissions-driven. Correct? Can you please make this explicitly clear? If correct, I am confused how does permafrost C emissions play a role. Does it change diagnosed emissions (Figure 1a)? But you mention increased radiative forcing due to permafrost C emissions which implies that it's the temperature that's changing. So does this mean you run your simulations with permafrost C feedback (PCF) turned on with emissions from the simulations without PCF. Did I miss this? If not, please clarify this.

Thank you for your suggestion. Our transient simulations are all driven by CO2 emissions, whereas the spin-up simulations are conducted under fixed CO2 concentrations at pre-industrial levels. For temperature stabilization and overshoot scenarios, the prescribed CO2 emissions are derived from transient simulations with the permafrost carbon module deactivated, thereby excluding the influence of permafrost carbon emissions. Subsequently, these diagnosed CO2 emissions are utilized to drive two sets of simulations, one with the permafrost carbon module activated and another with it deactivated. This setup enables us to isolate and quantify the additional warming and radiative forcing effects due to permafrost carbon emissions. We have clarified this procedure more explicitly in the revised manuscript.

10) Unless I missed this, can you please clarify how is permafrost is defined to be able to calculated permafrost extent?

The UVic ESCM v2.10 defines the total spatial coverage of permafrost as the area in which soil remains perennially frozen for a minimum of two consecutive years.

11) Please clarify what determines the boundary between usual and frozen soil C.

Thank you for your suggestion. In the UVic ESCM v2.10 model, usual soil carbon and permafrost carbon are depicted as two distinct depth-resolved carbon pools within the upper six soil layers. Soil carbon that is transported downward and crosses the permafrost table is transformed into permafrost carbon. Conversely, permafrost carbon that is moved upward and crosses the permafrost table is converted back into usual soil carbon. We have clarified this point in the revised manuscript.

**12) Please introduce Figure 3 properly in the text and state its purpose.**

Thank you for you suggestion. In the revised manuscript, we have added a proper introduction to the figure (originally Figure 3, now updated to Figure 4) to state its purpose and have adopted the same visual style as Figure 3 to enhance clarity and consistency. Specifically, Figure 4 illustrates how losses of permafrost area, permafrost carbon, and permafrost region soil carbon evolve relative to the SWL-1.5 scenario, providing a direct assessment of the reversibility of permafrost responses under both stabilization and overshoot scenarios. The updated Figure 4 is shown below.

Figure 4. Similar to Figure 3, but showing timeseries of changes relative to the SWL-1.5 scenario in (a) permafrost area loss, (b) permafrost carbon loss, and (c) permafrost region soil carbon loss under stabilization and overshoot scenarios at 2.0 °C (green), 3.0 °C (red), and 4.0 °C (purple) global warming levels. Square markers indicate the time points when the temperature overshoot reaches its peak or stabilized warming begins, while circle markers indicate when the overshoot returns to 1.5 °C. Results represent the ensemble median of 250 simulations based on the PFC simulations. Dots on the right panels represent values in the year 2300, with uncertainty ranges estimated as the 5th to 95th percentiles.

13) In Figure 5, I found it hard to distinguish between filled squares and circles. Perhaps increase their size or put a black border around them.

Thank you for your suggestion. To improve visual clarity, we have increased the marker size and put a black border around them in the revised figure (originally Figure 5, now Figure 7). Similar improvements have also been applied to other relevant figures. The updated figure is shown below.

14) Line 351, "Permafrost C release significantly increase ...". Does "significantly" in this sentence means statistically significant? If not, try using some other word.

Thank you for your suggestion. To avoid confusion with statistical significance, we have replaced "significantly" with "evidently" in the revised sentence.

I look forward to reading a revised version of your manuscript.

Best regards, Vivek

---

## Editor Decision (ED1)

**Permafrost response and feedback under temperature stabilization and overshoot scenarios with different global warming levels**

and the carbon associated carbon

Min Cui1, Duoying Ji1 and Yangxin Chen1

1Faculty of Geographical Science, Beijing Normal University, Beijing, 100875, China

Correspondence to: Duoying Ji (duoyingji@bnu.edu.cn)

**Abstract.**

Permafrost regions in the northern high latitudes face significant degradation risks under global warming and threaten the achievement of global climate goals. This study explores the response and feedback of permafrost Linder temperature stabilization scenarios, where the global mean temperature stabilizes at various global warming levels, and overshoot scenarios, where the global mean temperature temporarily exceeds the 1.5 °C warming target before returning. Under the 1.5 °C and 2 °C stabilization scenarios, permafrost area is projected to decrease by 4.6 [4.5 to 4.7] million km2 and 6.6 [6.4 to 6.8] million km2 respectively, from a pre-industrial level of 17.0 million km2. Corresponding permafrost carbon losses are estimated at 54 [32 to 79] PgC and 72 [42 to 104] PgC, relative to a pre-industrial carbon stock of 484 [383 to 590] PgC. In overshoot scenarios, permafrost area shows effective recovery, with additional losses of only 0.6 [0.3 to 1.1] million km2 compared to the 1.5 °C stabilization scenario. In contrast, permafrost carbon loss remains largely irreversible, with additional loss of 24 [4 to 52] PgC compared to the 1.5 °C stabilization scenario, Both stabilization and overshoot scenarios show that additional warming due to permafrost carbon feedback rises with higher global warming levels, and the most substantial unclean feedback in overshoot scenarios is anticipated during the cooling phase. The additional permafrost area loss due to permafrost carbon feedback, which accounts for 5 [2 to 11] % of the total loss, is influenced by both the magnitude of additional warming and the sensitivity of permafrost area to global warming. Moreover, the responses of permafrost area. permafrost carbon and associated radiative forcing to a broad range of global warming exhibit near-linear relationships under stabilization scenarios. Permafrost carbon feedback is unlikely to initiate a self-perpetuating global tipping process under both stabilization and overshoot scenarios. These findings have significant implications for long-term climate change and Is this a generalized statement or based on the simulations presented here? mitigation strategies.

**25 1 Introduction**

Permafrost soils in the northern high latitudes contain an estimated 1100-1700 Pg of carbon, primarily in the form of frozen organic matter, which is roughly twice the amount of carbon in the atmosphere (Hugelius et al., 2014; Schuur et al., 2015). As the climate warms, the gradual of abrupt permafrost thaw along with subsequent microbial decomposition, would release carbon dioxide (CO2) and methane (CH4) into the atmosphere, thereby amplifying the warming effect (Koven et al.,

2011; Feng et al., 2020; Smith et al., 2022). The positive feedback mechanism, combined with the fact that warming rates in the Arctic exceed the global average (Fyfe et al., 2013; Liang et al., 2022; Rantanen et al., 2022), underscores the critical role of permafrost as a key tipping element in the climate system (Armstrong McKay et al., 2022). However, current Earth system models inadequately represent or omit the permafrost carbon processes, which has become one of the largest sources of uncertainty in future climate projections (Schädel et al., 2024). Therefore, researching the release of permafrost carbon and its feedback is crucial for accurately assessing climate risks and formulating effective emission reduction strategies.

The Paris Agreement aims to limit global average temperature rise to well below 2 °C above pre-industrial levels, with efforts to keep it below 1.5 °C. Despite these goals, global warming has already exceeded 1 °C and is on track to surpass 3 °C by the end of the 21st century, primarily due to increased anthropogenic CO2 emissions (Haustein et al., 2017; Hausfather and Peters, 2020). If current emission rates persist, the remaining carbon budgets compatible with the 1.5 °C target will be critically tight and likely exhausted within the next few years (Rogelj et al., 2015; Goodwin et al., 2018; Masson-Delmotte et al., 2018; Forster et al., 2023; Smith et al., 2023). It is unlikely that the 1.5 °C target set by the Paris Agreement will be met (Raftery et al., 2017). However, it might still be achievable after a period of temperature overshoot, by compensating for excessive past and near-term emissions with net-negative emissions at a later time – i.e., through on-site CO2 capture at emission sources and carbon dioxide removal from the atmosphere (Gasser et al., 2015; Sanderson et al., 2016; Seneviratne et al., 2018; Drouet et al., 2021; Schwinger et al., 2022).

Several existing studies have assessed the climate response to overshoot pathways. Many components of the physical climate system have been identified as reversible, although typically with some hysteresis behavior (Boucher et al., 2012; Wu et al., 2015; Tokarska and Zickfeld, 2015; Li et al., 2020; Cao et al., 2023). In this context, reversibility refers to a partial recovery of climate conditions in an overshoot scenario toward an Earth system state without overshoot. These studies demonstrate that global mean temperature, sea surface temperature, and permafrost area can recover within decades to centuries in response to net negative emissions. Carbon release from permafrost has been shown to be irreversible on multidecadal to millennial timescales (MacDougall et al., 2013; Schwinger et al., 2022). The presence or absence of hysteresis effect in the permafrost processes is influenced by multiple factors, including the thermal inertia of permafrost soils, potential shifts in vegetation composition, and the extent to which irreversible permafrost carbon losses are offset by gains in vegetation and non-permafrost soil carbon reservoirs (MacDougall, 2013; Schwinger et al., 2022). Furthermore, the soil carbon loss under overshoot scenarios significantly affects the hydrological and thermal properties of soils (Zhu et al., 2019), which in turn modulate the processes involved. The interactions between physical and biophysical processes can potentially stabilize the carbon, water, and energy cycles at distinct post-overshoot equilibria (de Vrese and Brovkin, 2021). Therefore, a temporary warming of the permafrost regions entails important legacy effects and lasting impacts on its physical state and carbon cycle. However, theselstudies have yet to assess permafrost carbon feedback under overshoot scenarios.

Few studies have examined permafrost carbon response and feedback under a long-term climate stabilization scenario, but such scenarios may be more realistic if ambitious emission mitigation strategies are not implemented or carbon dioxide removal methods are not effective. Most carbon dioxide removal methods have not yet been proven at large scales (Anderson et al., 2023), and sustainability concerns further limit land-based carbon dioxide removal options (Deprez et al., 2024). Multi-model ensemble mean projections suggest that global temperature change following the cessation of anthropogenic greenhouse gas emissions will likely be close to zero in the following decades, although individual models show a range of temperature evolution after emissions cease from continued warming for centuries to substantial cooling (MacDougall et al., 2020; MacDougall, 2021; Jayakrishnan et al., 2024). MacDougall (2021) demonstrated that under a cumulative emission scenario of 1000 PgC, permafrost carbon feedback could raise global temperature by approximately 0.06 °C within 50 years after emissions cease. However, systematic research is still lacking on how permafrost carbon will be released and further amplify global warming under temperature stabilization scenarios, as well as how these processes differ from temperature overshoot scenarios reaching the same peak warming. Additionally, most post-Paris Agreement studies have focused on understanding the impacts of 1.5 °C and 2 °C global warming levels, with little attention given to the higher global warming levels (Rogelj et al., 2011; Comyn-Platt et al., 2018; King et al., 2024). Assessment of permafrost carbon feedback under various global warming levels would provide a more comprehensive understanding of the critical role of permafrost in the climate system.

This study aims to fill these gaps using an Earth system model of intermediate complexity to systematically assess the permafrost response and feedback under temperature stabilization or overshoot scenarios achieving various global warming levels. The structure of the remainder of this paper is as follows: Section 2 introduces the model description, design of temperature stabilization and overshoot scenarios at different global warming levels, as well as the methodology for perturbed parameter ensemble simulations. Section 3 provides the results of these simulations. Finally, Section 4 offers conclusions and discussion of our findings.

**2 Methods**

**2.1 Model Description**

This study uses the University of Victoria Earth System Climate Model version 2.10 (UVic ESCM v2.10), an intermediate-complexity Earth system model with a uniform horizontal resolution of 3.6° longitude by 1.8° latitude, to simulate permafrost carbon response and feedback under temperature stabilization and overshoot scenarios. The atmospheric component of UVic ESCM is a single layer moisture-energy balance model. The oceanic component incorporates a fully three-dimensional general circulation model, with a vertical resolution of 19 levels, coupled to a thermodynamic and dynamic sea-ice model. A more detailed description of UVic ESCM v2.10 can be found in Weaver et al. (2001) and Mengis et al. (2020).

The terrestrial component of UVic ESCM v2.10 uses the Top-down Representation of Interactive Foliage and Flora Including Dynamics (TRIFFID) vegetation model to describe the states of five plant functional types (PFT): broadleaf tree, needleleaf tree, C3 grass, C4 grass, and shrub (Cox. 2001; Meissner et al., 2003). A coupled photosynthesis-stomatal conductance model is used to calculate carbon uptake via photosynthesis, which is subsequently allocated to vegetation primary productivity

growth and respiration. The resulting net earbon fluxes drive changes in vegetation characteristics, including areal coverage. leaf area index, and canopy height for each PFT. The UVic ESCM v2.10 utilized in this study does not account for nutrient limitations in the terrestrial carbon cycle, leading to an overestimation of global gross primary productivity and an enhanced capacity of land to take up atmospheric carbon (De Sisto et al., 2023). However, the model reasonably represents the dominant PFTs of C3 grass, shrub and needleleaf tree at northern high latitudes, although it underestimates vegetation carbon density over this area (Mengis et al., 2020).

The UVic ESCM v2.10 represents the terrestrial subsurface with 14 layers, extending to a total depth of 250.3 m to correctly capture the transient response of permafrost on centennial timescales. The top eight layers (10.0 m) are involved rulic cycle, while the deeper layers are modeled as impermeable bedrock (Avis et al., 2011). The carbon cycle is active in the top six layers (3.35 m), where organic carbon from litterfall, simulated by the TRIFFID vegetation model, is allocated to soil layers with temperatures above 1 °C according to an exponentially decreasing function with depth. If all soil layers are below 1 °C, the organic carbon is added to the top soil layer. The soil respiration is calculated for each layer individually as a function of temperature and moisture, but the respiration ceases when the soil layer temperature falls below 0 °C (Meissner et al., 2003; Mengis et al., 2020). In regions where permafrost exists-defined as areas where soil temperature remains below 0 °C for at least two consecutive years—the model applies a revised diffusion-based cryoturbation scheme to redistribute soil carbon within the soil column. Compared to the original diffusion-based cryoturbation scheme proposed by Koven et al. (2009), the revised cryoturbation scheme calculates carbon diffusion using an effective carbon concentration that incorporates the volumetric porosity of the soil layer, rather than the actual carbon concentration, thereby resolving the disequilibrium problem of the permafrost carbon pool during model spin-up (MacDougall and Knutti, 2016). However, as the UVic ESCM v2.10 only simulates permafrost carbon in the top 3.35 m of soil, the current cryoturbation scheme cannot initiate the formation of Yedoma. As a result, soil carbon stored in deep In the UVic ESCM v2.10. Jusual soil carbon and permafrost carbon are depicted as two distinct depth-resolved carbon

In the UVic ESCM v2.10 dusual soil carbon and permafrost carbon are depicted as two distinct depth-resolved carbon pools within the upper six soil layers. Soil carbon that is transported downward and crosses the permafrost table (i.e. the depth of the active layer) is transformed into permafrost carbon. Conversely, permafrost carbon that is moved upward and crosses the permafrost table is converted back into usual soil carbon. Permafrost carbon can only be decomposed into CO2, as the UVic ESCM v2.10 does not include a methane production module (MacDougall and Knutti, 2016). The permafrost carbon pool is characterized by four key parameters: (1) a decay rate constant; (2) the available fraction of the pool, which represents the combined size of both the fast and slow carbon pools subject to decay; (3) a passive pool transformation rate, which governs the rate at which passive permafrost carbon transitions into the available fraction; and (4) a saturation factor used to calibrate the total size of the permafrost carbon pool, which is linked to soil mineral porosity and accounts for the decreasing concentration of soil carbon with depth in permafrost regions (Hugelius et al., 2014). These four key parameters determine the size and vulnerability to decay of the permafrost carbon pool. A more detailed description of the UVic

ESCM's permafrost carbon parameterization scheme and its simulated permafrost carbon characteristics can be found in Dougall and Knutti (2016).

As an Earth system model of intermediate complexity, the UVic ESCM offers relatively low computational costs and MacDougall and Knutti (2016). 130

serves as an ideal instrument for performing simulations that are not yet feasible with state-of-the-art Earth system models (Weaver et al., 2001; MacIsaac et al., 2021). In recent years, the UVic ESCM has played a key role in assessments of carbon-climate feedbacks (Matthews and Caldeira, 2008; Matthews et al., 2009; Tokarska and Zickfeld, 2015; Zickfeld et al., 2016) and long-term climate change projection (Ehlert et al., 2018; Mengis et al., 2020; MacDougall et al., 2021). UVic ESCM v2.10 accurately reproduces historical changes in temperature and carbon fluxes (Mengis et al., 2020), as compared with the observational dataset (Haustein et al., 2017) and Global Carbon Project 2018 (Le Quéré et al., 2018). Moreover, it has been validated for simulating permafrost area, permafrost carbon stocks and the distribution of anthropogenic carbon of the GCP takes cares of this in the previous sentence. emissions across the atmosphere, ocean and land (Mengis et al., 2020).

**2.2 Experimental Design**

To quantify the impact of permafrost carbon response under different climate conditions, we designed a series of idealized climate scenarios, including four stabilized warming level (SWL) and three overshoot (OS) trajectories, spanning a long period from 1850 to 2300. In these scenarios, the global mean temperature increases to a maximum of 1.5 °C, 2.0 °C, 3.0 °C, and 4.0 °C above pre-industrial levels (1850-1900). Following this initial warming phase, the scenarios diverge into two groups following distinct temperature trajectories. In the stabilization scenarios, temperature stabilizes at the respective global warming levels and remains steady until 2300, referred to as SWL-1.5, SWL-2, SWL-3, and SWL-4. In contrast, the overshoot scenarios allow temperatures to temporarily reach these warming peaks before entering a cooling phase symmetrical to the warming phase, gradually reducing the global mean temperature to 1.5 °C above pre-industrial levels, where it remains steady until 2300. These overshoot pathways are referred to as OS-2, OS-3, and OS-4, respectively.

A simple proportional control scheme (Zickfeld et al., 2009) was used to derive the CO2 emissions pathway corresponding to each stabilization and overshoot scenarios, adjusting emissions to align the global mean temperature with the prescribed trajectory. The proportional control equation is given by:

$$E_{i+1} = k_{PE} \left( T_i - T_{i,goal} \right) ,$$

Where  $E_{i+1}$  represents the carbon emission at year i+1,  $T_i$  represents the simulated annual mean global mean temperature at year i, and  $T_{i,aoal}$  represents the prescribed global mean temperature trajectory at year i for each scenario. The time-invariant coefficient  $k_{PE}$  is set to 1515 PgC K-1, ensuring that the control scheme responds neither too quickly nor too slowly to the diagnosed temperature derivation. Base on the proportional control scheme, a set of emission trajectories diagnosed from initial simulations of stabilization and overshoot scenarios using the UVic ESCM v2.10 with the permafrost carbon module deactivated, thereby excluding the influence of permafrost carbon emissions. Specifically, all 160

these initial simulations of stabilization and overshoot scenarios firstly follow a unified emissions trajectory, which is based on historical CO2 emissions (Friedlingstein et al., 2022) until 2021, and thereafter follow the emissions trajectory of the Shared Socioeconomic Pathway 5–8.5 (SSP5-8.5) (O'Neill et al., 2016) until the highest temperatures of each scenario are approached. As the highest temperatures of each scenario are approached, the proportional control gradually, steps in to mediate CO2 emissions and realize the warming trajectory of each scenario. The CO2 emission trajectories diagnosed from the initial simulations were used later to drive formal simulations of stabilization and overshoot scenarios (Fig. 1a). Once these emission trajectories were established, there is no further application of the proportional control in the formal simulations.

To isolate the contribution of permafrost carbon feedback, two parallel sets of formal simulations were conducted for each scenario, both driven by the same CO2 emission trajectories diagnosed from the initial simulations. One set activated the permafrost carbon module and is referred to as PFC simulations, while the other set deactivated the permafrost carbon module and is referred to as NPFC simulations. Since the emission trajectories were diagnosed from the initial simulations in which the proportional control scheme was applied to achieve the desired temperature pathways, applying them in the formal simulations with the permafrost carbon module deactivated can effectively achieve the designed temperature trajectories (Fig. 1b). However, applying the diagnosed emissions in simulations with the permafrost module activated results in any permafrost carbon fluxes being effectively added on top of the diagnosed emissions, thereby causing additional warming. In other words, to achieve the intended climate targets under the same emission pathways, removals equivalent to the permafrost carbon emissions would be required. Therefore, the comparison between the PFC and NPFC simulation sets provides a robust framework to isolate and quantify the additional warming and radiative forcing effects due to permafrost carbon emissions under stabilization and overshoot scenarios. For comparison, two parallel simulations with permafrost carbon module activated or deactivated were also conducted for the high-emissions SSP5-8.5 scenario.

Figure 1. Temperature stabilization and overshoot scenarios designed through a simple proportional control scheme (Zickfeld et al., 2009) on CO2 emission and UVie ESCM v2.10 with permafrost carbon module deactivated (corresponding to the NPFC simulations). (a) Accumulated CO2 emission and (b) global warming relative to the pre-industrial levels (1850-1900), in the

Cumulative temperature change historical and SSP5-8.5 scenario (black), the stabilization (dashed lines) and overshoot (solid lines) scenarios at 1.5 °C (blue), 2.0 °C (green), 3.0 °C (red) and 4.0 °C (purple) global warming levels.

To evaluate the uncertainty in permafrost carbon response under stabilization and overshoot scenarios, we perturbed the four key permafrost carbon parameters following the methodologies of MacDougall and Knutti (2016) and MacDougall (2021). The permafrost carbon decay constant was derived from the mean residence time (MRT) of the slow soil carbon pool at 5 °C in permafrost soils and adjusted to reflect decay at 25 °C using the method proposed by Kirschbaum (2006). The probability distribution function of mean residence time was taken as a normal distribution with a mean of 7.45 years and a standard deviation of 2.67 years from Schädel et al. (2014). The available fraction of permafrost carbon was derived from the size of the fast, slow and passive soil organic carbon pools separately for organic, shallow mineral, and deep mineral soils measured by Schädel et al. (2014). The probability distribution function of available fraction of permafrost carbon was described by weighted gamma distributions, with each distribution respectively describing the probability distribution function of available fraction of permafrost carbon in organic, shallow mineral, and deep mineral soils. The passive carbon pool transformation rate was estimated from the 14C age of the passive carbon pool from midlatitude soils, yielding an estimated value of  $0.25 \times 10^{-10}$  to  $4 \times 10^{-10}$  s-1 (Trumbore, 2000). The probability distribution function of the passive carbon pool transformation rate was taken as uniform in base-two logarithmic space (MacDougall and Knutti, 2016). For the initial quantity of permafrost region soil carbon, its probability distribution function was taken as a normal distribution with a mean of 1035 PgC and a standard deviation of 75 PgC, informed by Hugelius et al. (2014). A series of 5,000-year sensitivity runs were performed under preindustrial steady conditions with varying saturation factors to determine their relationship with the quantity of the permafrost region soil carbon pool. This relationship was then utilized to tune the permafrost region soil carbon pool, ensuring alignment with observational data (Hugelius et al., 2014). Fig. 2 illustrates the probability distribution function for each perturbed parameter. MacDougall and Knutti (2016) and MacDougall (2021) additionally perturbed two physical climate parameters controlling climate sensitivity and Arctic amplification, but they are not perturbed in this study due to their limited influence on global mean temperature in stabilization scenarios.

The Latin hypercube sampling method (McKay et al., 1979) was used to explore the effects of parameter uncertainty on projections of permafrost carbon change. In this study, the probability distribution function of each key permafrost carbon parameter was divided into 25 intervals of equal probability. One value was randomly selected from each interval for a given parameter, and then randomly matched with values of the other three key parameters selected in the same manner to generate parameter sets. This sampling procedure was repeated 10 times, resulting in 250 unique parameter sets (i.e., 250 model variants). For each parameter set, the UVic ESCM v2.10 was first run through a 10,000-year spin-up phase under preindustrial conditions to achieve a quasi-equilibrium state. For these spin-up runs, the atmospheric CO2 concentration was fixed at 284.7 ppm and the solar constant was set to 1360.747 W m-2. Following the spin-up, emission-driven transient simulations were conducted under the stabilization, overshoot and SSP5-8.5 scenarios. The results are presented as the median across all model variants, with uncertainty quantified as the range between the 5th to the 95th percentiles.

eguivalent to time scale of  $\frac{1}{0.25 \times 10^{10}}$  5 = 1268 years.

time 205 years?

Figure 2. Probability distribution functions of the four key permafrost carbon parameters perturbed in the UVic ESCM v2.10 to represent uncertainty in permafrost carbon response. Panel (d) employs a logarithmic scale on the horizontal axis to better illustrate the distribution of the corresponding parameter. This figure is reproduced from MacDougall (2021).

Results

This is linear scale

If you were to convert this to
a time scale then, I think, you will need The UVic ESCM v2.10 reliably simulates historical temperature changes, permafrost area, and the partitioning of anthropogenic carbon emissions among the atmosphere, ocean and land. Under pre-industrial conditions, the simulated Northern Hemisphere permafrost area, defined as regions where the soil layer remains perennially frozen for at least two consecutive years, was 17.01 [17.00 to 17.04] million km2, the simulated total soil carbon stock in the permafrost regions was 1031 [915 to 1149] PgC, of which 484 [383 to 590] PgC was classified as perennially frozen carbon and 547 [533 to 559] PgC was classified as usual/soil carbon. For the period 1960–1990, the model simulated Northern Hemisphere permafrost area at 16.8 [16.7 to 16.9] million km2, which falls within the reconstructed range from 12.0 to 18.2 million km2 (Chadburn et al., 2017) and the observation derived extent from 12.21 to 16.98 million km2 (Zhang et al., 2000).

Additionally, the simulated soil carbon stock in the top 3.35 m of permafrost regions for this same period was 1034 [919 to 1151] PgC, with 483 [382 to 587] PgC classified as perennially frozen carbon, accounting for 47% [42% to 51%] of the total permafrost soil carbon stock, in agreement with Hugelius et al. (2014). During the period 2011-2020, the model estimated a global mean temperature increase of 1.14 [1.13 to 1.15] °C relative to preindustrial levels, which is closely aligned with the observed rise of 1.09 [0.91 to 1.23] °C (Gulev et al., 2021). From 2010 to 2019, the model estimated that anthropogenic carbon emissions were distributed as follows: 5.5 [5.4 to 5.6] PgC yr-1 to the atmosphere, 3.0 [2.98 to 3.03] PgC yr-1 to the ocean, and 2.5 [2.4 to 2.6] PgC yr1 to terrestrial ecosystems. These estimates are broadly consistent with the global anthropogenic CO2 budget assessment by the Global Carbon Project (GCP) with figures of 5.1±0.02 PgC yr1 for the atmosphere, 2.5±0.6 PgC yr1 for the ocean, and 3.4±0.9 PgC yr1 for terrestrial ecosystems (Friedlingstein et al., 2020).

**3.1 Permafrost Response**

by year 21xx The Northern Hemisphere high-latitude permafrost area is strongly correlated with changes in global mean temperature (Fig. 3a, b; Fig. S1a, b). As global warming increases from 1.5 °C to 4 °C, the permafrost area declines from 13.9 to 8.3 million km2 under the SSP5-8.5 scenario. In the stabilization scenarios, when global warming is stabilized at the Paris Agreement targets of 1.5 °C or 2 °C, permafrost degradation is effectively suppressed compared to the SSP5-8.5 scenario. By 2300, permafrost area decreases by 4.6 [4.5 to 4.7] million km2 and 6.6 [6.4 to 6.8] million km2 from the pre-industrial level of 17.0 million km2 under SWL-1.5 and SWL-2 scenarios, respectively, accounting for 39 [38 to 40] % and 56 [54 to 58] % of the reduction observed under SSP5-8.5 scenario. The incremental permafrost degradation under the SWL-3 compared to SWL-2 is significantly larger than that under SWL-4 compared to SWL-3. This is mainly because the remaining permafrost available for degradation becomes progressively limited under higher global warming levels, and the permafrost area under higher stabilization scenarios has not yet reached a steady state in our simulations. Additionally, the permafrost area under the SWL-3 and SWL-4 scenarios exceeds that under the SSP5-8.5 scenario by only 2.0 [1.9 to 2.1] million km2 and 0.9 [0.8 to 1.0] million km2 by 2300, respectively. This is primarily because the permafrost area in the SSP5-8.5 scenario is smaller and, as a transient simulation, is further from equilibrium compared to SWL-3 and SWL-4. However, during the cooling phase of the overshoot scenarios, as the global mean temperature returns to 1.5 °C above preindustrial levels, the permafrost area gradually recovers to that under the SWL-1.5 scenario (Fig. 4a; Fig. S2a). By 2300, it converges to similar levels of 11.1~12.4 million km2 under the OS-2, OS-3 and OS-4 scenarios, with an additional loss of only 0.2~1.2 million km2 compared to the SWL-1.5 scenario. This indicates that permafrost area is nearly reversible and largely follows the global mean temperature trajectory, recovering as temperature reduction, consistent with previous studies (MacDougall, 2013; Lee et al., 2021; Schwinger et al., 2022). It is noteworthy that permafrost area loss under overshoot scenarios typically exhibits a hysteresis effect (Boucher et al., 2012; MacDougall, 2013; Eliseev et al., 2014), peaking about 10~30 years after global mean temperature reaches its maximum in our simulations.

Figure 3. Timeseries of annual mean (a) global warming, (b) permafrost area, (c) permafrost carbon loss and (d) permafrost region soil carbon loss under stabilization (dashed lines) and overshoot (solid lines) scenarios at 1.5°C (blue), 2.0°C (green), 3.0°C (red), and 4.0 °C (purple) global warming levels, as well as the SSP5-8.5 scenario. Square markers indicate the time points when the temperature overshoot reaches its peak or stabilized warming begins, while circle markers indicate when the overshoot returns to 1.5°C. All changes are relative to the pre-industrial period (1850-1900). Results represent the ensemble median of 250 simulations based on the PFC simulations. Dots on the right panels represent values in the year 2300, with uncertainty ranges estimated as the 5th to 95th percentiles.

Figure 4. Similar to Figure 3, but showing timeseries of changes relative to the SWL-1.5 scenario in (a) permafrost area loss, (b) permafrost carbon loss, and (c) permafrost region soil carbon loss under stabilization and overshoot scenarios at 2.0 °C (green), 3.0 °C (red), and 4.0 °C (purple) global warming levels. Square markers indicate the time points when the temperature overshoot reaches its peak or stabilized warming begins, while circle markers indicate when the overshoot returns to 1.5 °C. Results represent the ensemble median of 250 simulations based on the PFC simulations. Dots on the right panels represent values in the year 2300, with uncertainty ranges estimated as the 5th to 95th percentiles.

Under both temperature stabilization and overshoot scenarios, permafrost carbon declines monotonically over time (Fig. 3c; Fig. S1c), driven by the imbalance between weaker permafrost carbon inputs and relatively faster decomposition. The primary source of permafrost carbon input is a very slow physical process of downward diffusion through the permafrost table due to cryoturbation mixing effect (MacDougall and Knutti, 2016). As a result, the rates of permafrost carbon decomposition significantly exceed its inputs (Fig. 5a, c; Fig. S3a, c). Under the SWL-1.5, SWL-2, SWL-3, and SWL-4 scenarios, permafrost carbon losses are projected to be 54 [32 to 79], 72 [42 to 104], 106 [64 to 152], and 127 [74 to 180] PgC by 2300, respectively, while under the OS-2, OS-3, and OS-4 scenarios, the losses are 60 [35 to 87], 78 [50 to 111], and 97 [63 to 135] PgC. This indicates that despite global mean temperature returning to 1.5 °C in overshoot scenarios. considerable permafrost carbon losses still occur, accounting for 82 [80 to 85] %, 73 [71 to 78] %, and 76 [72 to 85] % of the losses in the corresponding stabilization scenarios achieving same global warming levels. Additionally, the extra permafrost carbon losses under the SWL-2, SWL-3, and SWL-4 scenarios relative to the SWL-1.5 scenario continue to increase over time; whereas in the overshoot scenarios, these additional losses decrease during the stabilization phase (Fig. 4b; Fig. S2b) and reduce to 24 [4 to 52] PgC by 2300. This decrease occurs because, after global warming cools to 1.5 °C, the permafrost carbon decomposition rate under the overshoot scenarios closely aligns with that of the SWL-1.5 scenario, while the permafrost carbon inputs under the overshoot scenarios surpass that of the SWL-1.5 scenario.

Our simulations show that permafrost carbon inputs do not follow the same trajectory as soil carbon especially under overshoot scenarios. This is likely due to inaccurate parameterization adopted in the current model. As noted in the model description (Section 2.1), litterfall is allocated to soil layers with temperatures above 1 °C according to an exponentially decreasing function of depth. When all soil layers are below 1 °C, organic carbon from the litterfall is added to the top soil layer. Meanwhile, permafrost carbon and non-permafrost soil carbon are both represented as depth-resolved carbon pools within the top six soil layers. The movement of permafrost carbon due to cryoturbation mixing is parameterized as being proportional to the gradient of total soil carbon with depth. Soil carbon that diffuses downward through the permafrost table is converted to permafrost carbon. During the cooling phase of overshoot scenarios, increased litterfall and a rising permafrost table lead to elevated carbon concentrations in surface soil layers, resulting in enhanced vertical diffusion and a surge in permafrost carbon inputs. Conversely, under the SSP5-8.5 scenario, permafrost carbon inputs exhibit only a minor peak around the 2150s, followed by a sharp decline (Fig. 5a; Fig. S3a). This is due to the continuous reduction in permafrost area and the deepening of the permafrost table, both of which reduce carbon concentrations in the upper soil layers and weaken vertical diffusion, despite the increasing litter flux under a strong CO2 fertilization background. We note that the approach adopted in the model may not accurately describe natural processes of vertical carbon movement, which are influenced by soil porosity heterogeneity, freeze-thaw cycles, and ice expansion upon freezing.

Are you comparing Figures 5a and 11
5b ?

Permafrost region soil carbon shows a strong tendency to recover after temperature overshoot. In contrast, permafrost region soil carbon continues to decrease under temperature stabilization scenarios, but the rate of decrease is gradually slowing down. The permafrost region soil carbon release in the overshoot scenarios is significantly mitigated compared to stabilization scenarios at same global warming levels (Fig. 3d; Fig. S1d). By 2300, permafrost region soil carbon losses under OS-2, OS-3, and OS-4 scenarios are projected to be 41 [15 to 72], 61 [29 to 98], and 81 [44 to 124] PgC, respectively, with reductions of 14 [8 to 21], 41 [25 to 57], and 62 [41 to 82] PgC compared to the SWL-2, SWL-3, and SWL-4 scenarios, respectively. During the stabilization phase in the overshoot scenarios, additional permafrost region soil carbon losses compared to the SWL-1.5 scenario decrease, but the permafrost region soil carbon in the overshoot scenarios does not fully recover to the level under the SWL-1.5 scenario (Fig. 4c; Fig. S2c). Notably, in all stabilization and overshoot scenarios simulated in this study, the permafrost region soil serves as a net carbon source for atmospheric CO2 by 2300. However, during the stabilization phase of OS-3 and OS-4, the permafrost region soil turns into a carbon sink, as soil carbon inputs surpass the reduced decomposition activity due to the depletion of soil carbon stocks and reduced warming levels. Permafrost region soil carbon inputs primarily originate from robust biophysical processes related to vegetation litterfall, with their intensity influenced by warming levels and CO2 fertilization effects, while permafrost region soil carbon decomposition is closely tied to global mean temperature. In the overshoot scenarios, the peak for permafrost region soil carbon decomposition happens marginally earlier than the global mean temperature peak (Fig. 5b, d; Fig. S3b, d).

The permafrost region soil carbon inputs generally track the trajectory of litter flux across the same area, with an approximate delay of 10-20 years (not shown). To attribute the contribution of permafrost region soil carbon inputs, we examined how dominant vegetation types (needleleaf tree, C3 grass and shrub) over the permafrost region adapt to temperature and atmospheric CO2 concentrations in both overshoot and stabilization scenarios (Fig. 6). Needleleaf trees expand slowly and continuously in the permafrost region in both overshoot and stabilization scenarios, whereas that of shrubs closely follows the trajectory of global mean temperature. The combined areal coverage of trees and shrubs is projected to cover about 62% upon 1.5 °C warming relative to pre-industrial levels around 2040s, slightly higher than the 24-52% range projected for 2050 using a statistical approach that links climate conditions to vegetation types under two "berna distinct emission trajectories (Pearson et al., 2013). During the warming and cooling phases of overshoot scenarios, the expansion and reduction of shrubs correspond with the degradation and expansion of C3 grasses, respectively. Among the three dominant PFTs, only shrubs show a nearly reversible response in areal coverage, net primary productivity (NPP) and vegetation carbon with respect to global mean temperature under overshoot scenarios. In contrast, the continuous reduction carbon with respect to global mean temperature under overshoot scenarios. of C3 grasses and the expansion of needleleaf trees suggest a degree of irreversibility in the structure and vegetation carbon preindustred density of northern high latitude terrestrial ecosystems under overshoot scenarios. Our results are in line with an earlier study by Tokarska and Zickfeld (2015), but contrast with Schwinger et al. (2022) who reported only minor differences in vegetation carbon after the overshoots compared to the reference simulation with no overshoot by prescribing vegetation

As we discussed in our emoule, please clarify "permationst toil C" vs. "permationst vegron soil C", and that the "region" area is changing.

Fig 3 uses the term distributions. In our study, the shifts in vegetation composition and changes in living biomass, especially those associated with woody vegetation, are key drivers of permafrost region soil earbon inputs.

The uncertainty in permafrost region soil carbon (clease is nearly the same as that of permafrost carbon (clease) (Fig. 3c, d: Fig. S1c, d). For example, the 5th to 95th percentile range of permafrost region soil carbon release under the OS-2 and OS-4 scenarios is 58 PgC and 81 PgC respectively by 2300, compared to 52 PgC and 72 PgC for permafrost carbon release. This indicates that the uncertainty in permafrost region soil carbon release is largely driven by the uncertainty in permafrost carbon release. Therefore, we evaluate the relative importance of perturbed permafrost carbon parameters on permafrost region soil carbon release under different temperature pathways through calculating their correlations across all ensemble simulations. The influence of model parameters on the uncertainty of permafrost carbon losses by 2300 is relatively consistent across the SSP5-8.5, OS-4, and SWL-4 scenarios, with the strongest correlations observed for the permafrost passive carbon pool transformation rate (R=0.81~0.85), followed by the initial quantity of permafrost region soil carbon (R=0.55~0.61). This finding aligns with Ji et al. (2024), who highlights the critical role of these two parameters in the uncertainty of permafrost region soil carbon loss under temperature overshoot and 1.5 °C warming stabilization scenarios.

Figure 5. Timeseries of changes in (a) permafrost carbon inputs, (b) permafrost region soil carbon inputs, (c) permafrost carbon decomposition and (d) permafrost region soil carbon decomposition, under the stabilization (dashed lines) and overshoot (solid lines) scenarios at 1.5 °C (blue), 2.0 °C (green), 3.0 °C (red) and 4.0 °C (purple) global warming levels, along with the SSP5-8.5

Figure 6. Timeseries of annual mean areal fraction (left column), net primary productivity (middle column) and vegetation carbon (right column) under stabilization (dashed lines) and overshoot (solid lines) scenarios at 1.5 °C (blue), 2.0 °C (green), 3.0 °C (red), and 4.0 °C (purple) global warming levels, as well as the SSP5-8.5 scenario (black). Each row represents one of the three dominant plant functional type (PFT): (a-c) needleleaf tree, (d-f) C3 grass and (g-i) shrub. Square markers indicate the time points when the temperature overshoot reaches its peak or stabilized warming begins, while circle markers indicate when the overshoot returns to 1.5 °C. Results represent the ensemble median of 250 simulations based on the PFC simulations, and the shadings denote the 5th and 95th percentile uncertainty ranges.

**3.2 Radiative Impacts of Permafrost Carbon Release**

The permafrost carbon release with increase global mean radiative forcing and surface temperature. By comparing two parallel sets of simulations with the permafrost carbon module activated (PFC) or deactivated (NPFC), we were able to quantify the additional radiative forcing and warming caused by permafrost carbon release. The time evolution of additional global warming closely resembles the additional radiative forcing (Fig. 7a, b; Fig. S4a, b) due to approximately linear relationship between radiative forcing and temperature change based on the energy balance of the climate system (Forster et al., 1997; Myhre et al., 2014). Both in the stabilization and overshoot scenarios, the magnitude of additional radiative forcing and warming increases with higher global warming levels. By 2300, the additional warming in the overshoot scenarios steadily rises from 0.10 [0.06 to 0.15] °C in OS-2 to 0.14 [0.09 to 0.20] °C in OS-3 and 0.18 [0.11 to 0.25] °C in OS-4. Similarly, in the stabilization scenarios, the additional warming increases from 0.10 [0.06 to 0.14] °C in SWL-1.5 to 0.13 [0.07 to 0.19] °C in SWL-2, 0.21 [0.12 to 0.31] °C in SWL-3, and 0.24 [0.15 to 0.35] °C in SWL-4. This is because higher global warming levels lead to more significant reductions in permafrost carbon and permafrost region soil carbon, and further intensifying global warming. Furthermore, in the stabilization scenarios, the additional warming continues to increase over time due to delayed permafrost degradation and positive permafrost carbon feedback. In overshoot scenarios, the additional warming tends to stabilize once the temperature returns to 1.5 °C above pre-industrial levels. By 2300, the additional warming under SWL-2, SWL-3, and SWL-4 exceeds that of OS-2, OS-3, and OS-4 by 0.03 [0.01 to 0.04] °C, 0.07 [0.02 to 0.11] °C and 0.07 [0.03 to 0.10] °C, respectively, amplifying the additional warming in the overshoot scenarios by 22 % to 56 %.

However, the additional warming during the cooling phase is most substantial in overshoot scenarios, and it is also

However, the additional warming during the cooling phase is most substantial in overshoot scenarios, and it is also significantly greater than that in stabilization scenarios over the same period. This is primarily due to the sustained reduction in atmospheric CO2 concentration during the cooling phase, which amplifies the radiative forcing caused by permafrost carbon release. Specifically, because of the logarithmic relationship between CO2 concentration and radiative forcing (Etminan et al., 2016), the decline of background CO2 concentration to low levels causes the additional increases in CO2 concentration due to permafrost carbon release to produce more significant changes in radiative forcing. Similarly, the global mean warming under the SSP5-8.5 scenario is 2.43 [2.41 to 2.44] °C higher than under the SWL-1.5 scenario by 2100, but the additional warming difference due to permafrost carbon release is minimal at 0.01 [0 to 0.03] °C. Despite global mean warming reaching 8.20 [8.12 to 8.28] °C by 2300 in the SSP5-8.5 scenario, the additional warming is limited to 0.14 [0.08 to 0.20] °C due to the profoundly higher background CO2 concentration, the additional warming is only comparable to that of SWL-2 and OS-3 scenarios.

The additional warming caused by permafrost carbon release can be utilized to assess whether the permafrost carbon feedback could be classified as a global tipping point process. This means it is not only positive but also sufficiently strong to sustain itself. To qualify, an initial rise in global mean temperature would need to trigger permafrost carbon emissions that result in a further increase in global mean temperature surpassing the initial warming. As a result, the positive permafrost carbon feedback would induce sufficient additional thawing to initiate a self-sustaining feedback loop (Nitzbon et al., 2024). We employed the permafrost feedback factor, which is defined as the ratio of the additional warming to the initial warming simulated with the permafrost carbon module deactivated, to determine if the permafrost carbon feedback can be considered as a global tipping process. In all perturbed parameter ensemble simulations for the stabilization, overshoot and SSP5-8.5 scenarios, the maximum permafrost feedback factor is 0.21 °C °C-1. By 2300, the permafrost feedback factor for the OS4 and SSP5-8.5 scenarios are estimated at 0.12 [0.08 to 0.17] °C °C-1 and 0.02 [0.01-0.03] °C °C-1, respectively. The permafrost feedback parameter is the highest under the OS4 scenario, while it is the lowest under the SSP5-8.5 scenario (Fig. 8; Fig. S5). Interestingly, the feedback factors are quite similar across the stabilization scenarios, with values of 0.064 [0.037 to 0.096] °C °C-1, 0.064 [0.036 to 0.095] °C °C-1, 0.069 [0.040 to 0.103] °C °C-1 and 0.061 [0.038 to 0.089] °C °C-1 for the

SWL-1.5, SWL-2, SWL-3 and SWL-4 scenarios by 2300, respectively. Although the feedback factor in the overshoot scenarios is substantially larger than the recent estimate of 0.035 (0.004–0.110) °C °C-1 based on the Sixth Assessment Report of the Intergovernmental Panel on Climate Change (Nitzbon et al., 2024), our findings indicate that the positive permafrost carbon feedback is unlikely to result in enough additional thawing and corresponding carbon emissions to initiate a self-perpetuating tipping process. Since this study only models the gradual thawing of permafrost through the deepening of the active layer, we cannot rule out the possibility of tipping points associated with the abrupt thawing of talik development, thermokarst and thermo-erosion processes.

Figure 7. Additional changes in (a) radiative forcing, (b) global warming and (c, d) permafrost area due to permafrost carbon feedback, calculated as the difference between the PFC and NPFC simulations. Shown are results for the stabilization (dashed lines) and overshoot (solid lines) scenarios at 1.5 °C (blue), 2.0 °C (green), 3.0 °C (red) and 4.0 °C (purple) global warming levels, along with the SSP5-8.5 scenario (black). Square markers indicate the time points when the temperature overshoot reaches its peak or stabilized warming begins, while circle markers indicate when the overshoot returns to 1.5 °C. Results represent the ensemble median of 250 simulations. Dots on the right panels represent values in the year 2300, with uncertainty ranges estimated as the 5th to 95th percentiles. In panel (a), the additional radiative forcing is calculated using the simplified expressions (Etminan et al., 2016) based on simulated CO2 concentrations. In panels (c) and (d), the additional permafrost area loss is smoothed using a 5-year rolling average to eliminate interannual variability.

what's the difference between (c) and (d),

They're the same 16

Y-axis label and description in the figure caption.

Oh I see! (c) is overshoot sceneris, and (d) is

Stabilization. PIS make this clear in the caption.

Figure 8. Timeseries of permafrost feedback factor under stabilization (dashed lines) and overshoot (solid lines) scenarios at 1.5 °C (blue), 2.0 °C (green), 3.0 °C (red), and 4.0 °C (purple) global warming levels, as well as the SSP5-8.5 scenario. Square markers indicate the time points when the temperature overshoot reaches its peak or stabilized warming begins, while circle markers indicate when the overshoot returns to 1.5 °C. Results represent the ensemble median of 250 simulations. Dots on the right panels represent values in the year 2300, with uncertainty ranges estimated as the 5th to 95th percentiles. The permafrost feedback factor is calculated as the ratio of additional global warming caused by the permafrost carbon feedback (i.e., the difference between the PFC and NPFC simulations) to the global mean temperature change in the NPFC simulations.

The permafrost carbon feedback also causes additional permafrost area loss under stabilization, overshoot and SSP5-8.5 scenarios. During the cooling phase of OS-2, OS-3, and OS-4, there is a significant additional permafrost area loss of 0.3 [0.1 to 0.4], 0.4 [0.2 to 0.6], and 0.5 [0.3 to 0.7] million km² respectively (Fig. 7c; Fig. S4c), contributing 6 [2 to 8] %. 8 [4 to 12] % and 9 [5 to 12] % of total permafrost area loss by 2300. Although global warming has been identified as the primary driver of permafrost degradation (McGuire et al., 2016; Lawrence and Slate, 2005), the temporal evolution of additional permafrost area loss does not align with changes in the additional warming (Fig. 7b; Fig. S4b), particularly in the stabilization and SSP5-8.5 scenarios (Fig. 7d; Fig. S4d). To better understand this puzzle, we conducted sensitivity analysis based on the SSP5-8.5 scenario. The results show the sensitivity of permafrost area to global warming (SPAW, units of million km² °C-1) is not constant. The maximal sensitivity occurs below a global warming level of 1.5 °C. The magnitude of SPAW decreases as global warming surpasses 1.5 °C (Fig. 9a), suggesting the response of permafrost area to rising temperature weakens and indicating a diminishing feedback effect. Interestingly, by multiplying transient SPAW by the additional warming, the temporal evolution of the additional permafrost area loss can be well reconstructed across various scenarios (Fig. 9b-i). The root mean square error (RMSE) between original and reconstructed additional permafrost area loss, ranging from 0.04 to 0.09 million km², indicates high reconstruction accuracy.

Permafrost carbon feedback usually induces greater additional warming at higher global warming levels, but the lower SPAW diminishes its impact on permafrost area loss. Under the SSP5-8.5 scenario, the additional permafrost area loss peaks around the 2060s, reaching approximately 0.19 million km², coinciding with a strong sensitivity of 3.4 million km² °C⁻¹. After this peak, despite continued rise in additional warming, the permafrost area loss declines notably due to the gradual weakening of SPAW. Similarly, in the stabilization scenario, SPAW decreases markedly at higher global warming levels, even as additional warming increases, preventing a positive correlation between the additional permafrost area loss and global warming levels (Fig. 7d; Fig. S4d). For example, the additional warming in the SWL-4 scenario reaches 0.24 [0.15 to 0.35] °C by 2300, nearly twice that of the SWL-2 scenario (0.13 [0.07 to 0.19] °C). However, due to its SPAW being only 0.8 million km² °C⁻¹, approximately one quarter of 3.4 million km² °C⁻¹ under SWL-2, the additional permafrost area loss under SWL-4 scenario is only 0.2 [0.1 to 0.3] million km², significantly lower than 0.5 [0.2 to 0.7] million km² under SWL-2. This suggests the maximal SPAW occurring near 1.5 °C and 2 °C global warming levels has significant implications for the Paris Agreement's targets of limiting global warming at the same levels.

Figure 9. (a) Relationship between SPAW and global warming derived from the SSP5-8.5 scenario based on the PFC simulations.
Black curve represents SPAW calculated as the slope of a local regression (Cleveland et al., 1979, 1988, 1991) between changes in permafrost area and global mean temperature under the SSP5-8.5 scenario. Colored circle markers represent transient SPAW at 1.5 °C (blue), 2.0 °C (green), 3.0 °C (red), and 4.0 °C (purple) global warming levels. (b-i) Time series of original (solid line) and reconstructed (dashed line) additional permafrost area loss (ΔPF) due to permafrost carbon feedback for each scenario. The constructed results are derived by multiplying the transient SPAW with the additional warming (Fig. 7b; Fig. S4b) under each scenario. Smaller RMSE values indicate higher reconstruction accuracy.

**3.3 Linearity of Permafrost Response and Feedback**

After exploring the response and feedback of permafrost under temperature stabilization and overshoot scenarios at various global warming levels, it is natural to question the (non-)linearity of these response and feedback as functions of global warming levels. Our results show that the responses of permafrost area, permafrost carbon feedback and associated radiative forcing to a broad range of global warming are nearly linear (Fig. 10). The permafrost area change exhibits a strongly nonlinear relationship with global warming below 1.5 °C level, then a quasilinear relation between them in the global warming ranges from 1.5 °C to 3 °C. Above 3 °C global warming, the sensitivity of permafrost area to global warming decreases nonlinearly, and it is evident in both stabilization and SSP5-8.5 scenarios (Fig. 10a). In contrast, permafrost carbon loss and associated radiative forcing exhibit a nearly linear response to increasing global warming levels, especially above 1 °C, for both stabilization and SSP5-8.5 scenarios (Fig. 10b, c).

Meanwhile, the sensitivities of permafrost area, permafrost carbon loss, and associated radiative forcing to global warming under stabilization scenarios are all stronger than those under the SSP5-8.5 scenario. Specifically, based on the simulated permafrost area in the year 2300 under stabilization scenarios with global warming levels between 1.5 °C and 3 °C, the sensitivity of permafrost area to global warming is -3.19 [-3.01 to -3.36] million km2 °C-1. In comparison, a linear fit of permafrost area change against global warming levels over the same temperature range in the SSP5-8.5 scenario yields a sensitivity of -2.85 [-2.77 to -2.89] million km2 °C-1. Similarly, the permafrost carbon feedback per degree of global warming derived from a linear fit based on the total permafrost carbon loss in the year 2300 under stabilization scenarios, is -27.6 [-16.5 to -38.2] PgC °C-1. In contrast, the corresponding value under the SSP5-8.5 scenario, estimated from a linear fit over the 1.5 °C to 4.0 °C warming range, is -19.3 [-15.7 to -24.1] PgC °C-1. Applying the same approach, the associated radiative forcing per degree of global warming is estimated to be 0.08 [0.05 to 0.12] W m-2 °C-1 for the stabilization scenarios and 0.04 [0.03 to 0.05] W m-2 oC-1 for the SSP5-8.5. These differences between the stabilization and SSP5-8.5 scenarios are mainly attributable to the differing response time scales represented by the two scenarios: SSP5-8.5 reflects a typical transient response, while the stabilization scenarios maintain stabilized temperatures over extended periods and thus approximate a quasi-equilibrium response of the climate-carbon system. Furthermore, the smaller sensitivity of permafrost radiative forcing per degree of global warming under the SSP5-8.5 can be partially attributed to its higher background atmospheric CO2 concentration compared to the stabilization scenarios. The same amount of CO2 emissions would produce smaller additional radiative forcing under a higher background atmospheric CO2 concentration, due to the logarithmic relationship between CO2 concentration and radiative forcing (Etminan et al., 2016).

To a certain extent, our findings align with those of Nitzbon et al. (2024), who suggested that the accumulated response of Arctic permafrost to climate warming is approximately quasilinear. Nitzbon et al. (2024) reported a quasilinear decrease

**Is this also from Canadell et al. 2021**

in the equilibrium permafrost extent to global warming, with a rate of approximately 3.5 million km² °C¹. This quasilinear relation holds for global warming ranges from 0 °C to 4 °C, derived from the empirical relationship between the local permafrost fraction and the annual mean global temperature. However, our results indicate the quasilinear relationship only holds for global warming levels between 1.5 °C and 3 °C. Furthermore, the permafrost carbon feedback and the associated radiative forcing per degree of warming, as derived from our simulations of both stabilization and SSP5-8.5 scenarios, are within the ranges of -18 [-3.1 to -41] PgC °C¹ and 0.09 [0.02 to 0.20] W m⁻² °C⁻¹, respectively, reported by Canadell et al. (2021). Our estimates also align with the estimated range of equilibrium sensitivity of permafrost carbon decline to global warming, which is -21 [-4 to -48] PgC °C⁻¹. This may represent an upper limit for permafrost carbon feedback per degree of global warming, considering that the estimated reduction in permafrost carbon does not equate directly to carbon emissions released into the atmosphere, as noted by Nitzbon et al. (2024). 1 Think, 1 understand what is meant have

Under overshoot scenarios, permafrost area responds nearly reversibly and presents an almost closed loop (Fig. 10a). In contrast, permafrost carbon loss exhibits an open loop with respect to global warming levels. In other words, permafrost carbon loss does not reverse as temperatures decline, indicating irreversible permafrost carbon radiative forcing. Among the three metrics investigated here, only permafrost area exhibits strong reversibility under the overshoot scenarios. This also explains why, in Fig. 9a, the permafrost area sensitivity derived from the SSP5-8.5 scenario, when multiplied by additional warming, can reasonably reconstruct permafrost area loss in the stabilization and overshoot cases.

Figure 10. Relationship between global warming levels and three permafrost metrics: (a) permafrost area loss, (b) permafrost carbon loss, and (c) permafrost radiative forcing in the stabilization (colored dashed lines) and overshoot (colored solid lines) scenarios at 1.5 °C (blue), 2.0 °C (green), 3.0 °C (red) and 4.0 °C (purple) global warming levels, along with the SSP5-8.5 scenario (black). Square and circle markers indicate values in the year 2300 for the stabilization and overshoot scenarios, respectively. All results are based on the PFC simulations. Grey solid lines show linear fits of permafrost metrics to global warming levels in stabilization scenarios by 2300, while black dashed lines show corresponding fits for the SSP5-8.5 scenario. Note that in panel (a), both the stabilization scenarios and the corresponding SSP5-8.5 points included in the linear fit are limited to global warming levels between 1.5 °C and 3.0 °C, whereas in panels (b) and (c), the fits include points with global warming levels ranging from 1.5 °C to 4.0 °C. For stabilization scenarios, only the results from the year 2300 are used for fitting, while for the SSP5-8.5 scenario, all results within the specified global warming level ranges are used for fitting. Shaded regions represent the 5th to 95th percentile ranges across 250 ensemble simulations.

observation based as I don't think me for extent fine series of set extent

**4 Conclusions and Discussion**

This study utilizes the intermediate-complexity Earth system model UVic ESCM v2.10 and perturbed parameter ensemble modelling approach to study the response and feedback of permafrost under temperature stabilization and overshoot scenarios across different global warming levels. The UVic ESCM v2.10 has been validated against observational and reconstructed datasets, demonstrating its ability to reproduce historical permafrost area and permafrost carbon stocks. In addition to presenting the changes in permafrost area, permafrost carbon and permafrost region soil carbon under various warming trajectories, this study also quantifies the additional radiative forcing, global warming and permafrost area loss induced by permafrost carbon release and suggests that permafrost carbon feedback is unlikely to initiate a self-perpetuating global tipping process under both stabilization and overshoot scenarios. In addition, this study reveals quasilinear responses of permafrost area, permafrost carbon and associated radiative forcing to a broad range of global warning, providing insights into the implications of permafrost carbon feedback for long-term climate change and mitigation strategies.

Reductions in permafrost area and carbon exhibit a strong correlation with global warming under both stabilization and overshoot scenarios. In stabilization scenarios, lower global warming levels effectively mitigate permafrost degradation compared to the SSP5-8.5 scenario, whereas higher global warming levels lead to substantial permafrost degradation due to cumulative warming effects. In overshoot scenarios, permafrost area largely recovers as global warming returns to 1.5 °C levels, though this recovery is delayed by hysteresis effects, with degradation persisting for decades after temperature peaks. Permafrost carbon declines under both stabilization and overshoot scenarios, driven by the dynamic balance between soil carbon inputs and decomposition. The overshoot scenarios partially mitigate permafrost carbon losses compared to the stabilization scenarios at the same global warming levels. Significant carbon losses persist during the cooling and stabilization phases of overshoot scenarios, highlighting the essentially irreversible nature of this process. Permafrost region soil carbon exhibits a certain degree of recovery under the overshoot scenarios. In fact, soil in these regions even transitions into a carbon sink during the stabilization phase of the overshoot scenarios with high global warming levels, supported by reduced decomposition rates and sustained inputs from vegetation litterfall. However, the higher the overshoot levels, the less recovery there is. These findings underscore the critical role of temporary temperature overshooting levels in affecting long-term permafrost carbon release and recovery potential.

The responses of permafrost area, permafrost carbon, and associated radiative forcing to a broad range of global warming are nearly linear. The permafrost area and global mean temperature exhibit a quasilinear relation for the global warming ranges from 1.5 °C to 3 °C in both the stabilization and SSP5-8.5 scenarios. The permafrost carbon loss and associated radiative forcing exhibit a quasilinear relation to global warming ranges from 1 °C to 4 °C under the stabilization and SSP5-8.5 scenarios. The sensitivity of permafrost area to global warming derived from the stabilization and SSP5-8.5 scenarios is much higher than 1.6 million km² °C-1 derived through an equilibrium permafrost model (Liu et al., 2021), but our result is close to that derived from an observation-constrained equilibrium projection, approximately 3.5 million km² °C-1 (Nitzbon et al., 2024). Nitzbon et al. (2024) noted that permafrost area decreases quasi-linearly with increasing global mean temperature. However, we found that the relationship holds most strongly for the global warming ranges from 1.5 °C to 3 °C. According to the SSP5-8.5 simulations, the sensitivity of permafrost area to global warming reaches its peak below 1.5 °C global warming. It then decreases to a relatively stable level between 1.5 °C and 3 °C warming levels, and continues to decline beyond the 3 °C warming level. This is in line with Comyn-Platt et al. (2018), who found the feedback processes due to permafrost thaw respond more quickly at temperatures below 1.5 °C. The maximal sensitivity occurring below 1.5 °C global warming level suggests the fastest permafrost degradation is anticipated to take place within Paris Agreement's warming levels. Our findings have significant implications for the development of mitigation and adaptation strategies addressing permafrost-thaw impacts consistent with keeping global warming at the Paris Agreement's levels.

Our study highlights the substantial permafrost carbon feedback during the cooling phase of overshoot scenarios. Permafrost carbon release evidently increases global radiative forcing and amplifies global warming. The permafrost carbon feedback can be more profound in temperature stabilization and overshoot scenarios than in high-emissions scenarios. In stabilization scenarios, additional radiative forcing and warming persistently increase over time due to delayed degradation and positive permafrost carbon feedback. In overshoot scenarios, additional warming almost stabilizes once global warming drops to 1.5 °C levels. During the cooling phase of overshoot scenarios, lower background CO2 concentrations amplify the warming effect of permafrost carbon release. In contrast, under the high-emissions SSP5-8.5 scenario, the additional warming is limited due to higher background CO2 levels reducing the additional radiative forcing from permafrost carbon release. The additional permafrost area loss due to permafrost carbon feedback occupies 5 [2 to 11] % of the total loss by 2300 in the stabilization and overshoot scenarios. These additional permafrost area loss can be well explained by the sensitivity of permafrost area to global warming and the magnitude of additional warming. The complex interactions between global warming and permafrost degradation emphasize the importance of accounting for these nonlinear effects in climate projections, particularly at 1.5~2 °C global warming levels.

Our results show incomplete recovery of permafrost area under the overshoot scenarios, which is influenced by multiple factors: First, the additional permafrost carbon release leads to greater additional warming under the overshoot scenarios than the SWL-1.5 scenario, causing additional permafrost degradation. By 2300, the northern high-latitude permafrost regions are 0.01 ~ 0.13 °C warmer compared to the SWL-1.5 scenario. Second, the thermal inertia of deep soil layers limits the rate of permafrost recovery. Even after global mean temperatures return to the 1.5 °C target, residual heat accumulated in deeper soil layers during temperature overshoot period continues to inhibit permafrost refreezing, preventing full restoration to its pre-overshoot state. Third, greater soil carbon loss under overshoot scenarios substantially alters the hological and thermal properties of soil, affecting the processes that govern carbon cycling (Zhu et al., 2019; Avis, 2012; Lawrence and Slate, 2008), which in turn affects the recovery of permafrost area. Moreover, irreversible shifts in vegetation composition of high-latitude terrestrial ecosystems also contribute to the incomplete recovery of permafrost area under overshoot scenarios. For instance, among the two dominant vegetation types, needleleaf trees continue to expand while C3 grasses decline, even after global temperatures return to the 1.5 °C warming level. These irreversible changes may stabilize the carbon, water, and energy cycles over the permafrost region at different equilibria after overshoot, through the interactions between physical and biophysical processes (de Vrese and Broykin, 2021), thereby constraining the ability of permafrost to fully recover under overshoot scenarios.

Although permafrost carbon loss is essentially irreversible, overshoot scenarios exhibit a certain degree of recovery the overshoot scenarios.

relative to the SWL-1.5 stabilization scenario (Fig. 4b; Fig. S2b). It is therefore curious to know whether permafrost carbon under overshoot scenarios will eventually converge with that under SWL-1.5. Our results show that permafrost carbon inputs are consistently higher under overshoot scenarios than under SWL-1.5, while permafrost carbon decomposition differ only slightly between the two (Fig. 5a, c; Fig. S3a, c). This tends to suggest that the smaller permafrost carbon stocks under overshoot scenarios by 2300 would eventually catch up to the levels under SWL-1.5. To assess this potential convergence, we extended our simulations of both SWL-1.5 and overshoot scenarios to the year 2400 (data not shown). Then we estimated 615 the convergence time by calculating the ratio between the difference in permafrost carbon stocks and the difference in net permafrost carbon inputs (i.e., annual permafrost carbon inputs minus decomposition) for the overshoot scenarios relative to the SWL-1.5 scenario. Based on simulation results for the year 2300, the median estimated convergence times for OS-2. OS-3 and OS-4 are 1076, 1008 and 1433 years, respectively. When using results from the year 2400, the corresponding estimates 620 increase to 1377, 1199 and 1568 years. This means that convergence would take even longer if estimated from later simulation results, mainly due to gradually weakened permafrost carbon inputs. The relatively larger permafrost carbon inputs under overshoot scenarios result mainly from increased litterfall during the overshoot phase. The extra litterfall during the overshoot phase gradually moves through the active layer and is transported to the permafrost zone. Over time, however, the effect of this extra litterfall gradually diminishes, leading to a reduction in permafrost carbon inputs. Consequently, it may take extremely long timescales for the overshoot scenarios to fully converge with SWL-1.5 in terms of permafrost 625 carbon stocks. In addition, due to incomplete recovery of permafrost area and persistent changes in surface climate and soil properties, the overshoot scenarios might ultimately fail to converge to SWL-1.5 scenario in terms of permafrost carbon stocks.

Different permafrost carbon release and associated additional warming under overshoot scenarios confirm the pathdependent fate of permafrost region carbon (Kleinen and Broykin, 2018) and the path-dependent reductions in CO2 emission budgets (MacDougall et al., 2015; Gasser et al., 2018). As the permafrost carbon was accumulated very slowly during the last millions of years, its release would be tacked onto the anthropogenic CO2 emissions, and the resulting additional warming poses a challenge to achieving global climate goals by substantially reducing the remaining carbon budget compatible with the Paris Agreement (MacDougall et al., 2015; Natali et al., 2021). In the overshoot scenarios simulated in this study, permafrost carbon release by 2300 ranges from 60 [35 to 87] PgC to 97 [63 to 135] PgC. The associated additional warming caused by the release ranges from 0.10 [0.06 to 0.15] °C to 0.18 [0.11 to 0.25] °C. This permafrost carbon feedback contributes a substantial addition on top of 1.5 °C warming target under overshoot scenarios, and the magnitude of this additional warming rises with the amplitude of overshoot. To accomplish the 1.5 °C target under the OS-2. OS-3, and OS-4 scenarios, anthropogenic carbon emissions would be reduced by amounts equivalent to the permafrost carbon release. The proportion of carbon removal required to offset permafrost emissions is estimated at 4.9 [2.9 to 7.1] %,

6.5 [4.1 to 9.2] %, and 8.3 [5.4 to 11.6] % by 2300, respectively. Our findings are consistent with previous research utilizing the Monte Carlo ensemble method to evaluate the response of permafrost carbon and its influence on CO2 emission budgets under overshoot scenarios targeting a 1.5 °C warming limit (Gasser et al., 2018). Specifically, for overshoot amplitudes of 0.5 °C (peak warming of 2 °C) and 1 °C (peak warming of 2.5 °C), the reductions in anthropogenic CO2 emissions due to permafrost are estimated to be 130 (with a range of 30–300) Pg CO2 and 210 (with a range of 50–430) Pg CO2, respectively, to meet the long-term 1.5 °C target (Gasser et al., 2018). These results are comparable to our estimates of 60 [35 to 87] PgC under OS-2 and 78 [50 to 111] PgC under OS-3. The differences between the two studies can be partly attributed to different warming trajectories to achieve the same 1.5 °C target. Our study further confirms that if negative CO2 emissions were to be used to reverse the anthropogenic climate change, the delayed permafrost carbon release would reduce its effectiveness (MacDougall, 2013; Tokarska and Zickfeld, 2015).

This study does not simulate the changes of deep Yedoma carbon under the temperature stabilization and overshoot scenarios. Yedoma deposits represent a significant deep carbon reservoir and are widespread across Siberia, Alaska, and the Yukon region of Canada, having primarily formed during the late Pleistocene, especially in the late glacial period. These deep, perennially frozen sediments are particularly ice-rich, and the freeze-locked organic matter in such deposits can be remobilized on short time-scales, representing one of the most vulnerable permafrost carbon pools under future warming scenarios (Schuur et al., 2015; Strauss et al., 2017). According to Zimov et al. (2006), these perennially frozen Yedoma sediments cover more than 1 million km2, with an average depth of approximately 25 m. Recent estimates place the organic carbon stock in Yedoma deposits at 213 ± 24 PgC, constituting a significant portion of the total permafrost carbon stocks (Strauss et al., 2017). However, the UVic ESCM v2.10 utilized in this study simulates permafrost carbon only within the top 3.35 m of soil, limiting our ability to directly assess the impacts of temperature overshoot on deep Yedoma carbon. Considering their ice-rich nature and potential susceptibility to rapid-thaw processes, we analyzed the average and maximum active layer thickness (ALT) in Yedoma regions (Strauss et al., 2021, 2022) under the simulated scenarios to approximate potential impacts. We find that the average ALT in Yedoma regions remains below 1 m in all stabilization and overshoot scenarios, while the maximum ALT rarely exceeds 3.35 m in overshoot scenarios but does exceed this depth in some stabilization scenarios. However, in all scenarios, the maximum ALT does not exceed 6 m, which is relatively shallow compared to the average depth (-25 m) of Yedoma deposits (Figure S6). Consequently, the impact on Yedoma is comto be minimal in all scenarios, and the effect of overshoot scenarios on the deep Yedoma carbon is relatively minor compared to stabilization scenarios as well.

This study, like previous ones, uncovers considerable uncertainty in projections of permafrost carbon under global warming. The uncertainty represented by perturbed model parameters for each scenario can be interpreted as model uncertainty. We note that model uncertainty in permafrost carbon release gradually increases with the peak warming level and the duration of overshoot for each scenario (Fig. 3c; Fig. S1c). However, the uncertainty ranges in permafrost carbon release substantially overlap for overshoot and stabilization scenarios with adjacent warming levels, such as OS-2, SWL-1.5 and SWL-2. This is especially evident in low-level warming scenarios, where the uncertainty in projected permafrost carbon release is mainly driven by model uncertainty due to parameter perturbations, rather than scenario-related uncertainty. Given the significant roles of the permafrost passive carbon pool transformation rate and the initial quantity of permafrost region soil carbon in determining the uncertainty of permafrost region soil carbon release, it is expected that these two parameters contribute significantly to the overlapping uncertainty ranges of permafrost carbon and permafrost region soil carbon losses across different warming levels. Due to the interaction with soil carbon inputs, the overlapping uncertainty in permafrost region soil carbon release tends to differ from that of permafrost carbon release. For example, the uncertainty ranges in permafrost carbon release under OS-4 and SSP5-8.5 scenarios show considerable overlap, but the same does not apply to permafrost region soil carbon release, which results from significant differences in soil carbon inputs under distinct CO2 fertilization backgrounds. The large overlapping uncertainty in projecting permafrost carbon release under low-level warming scenarios, as shown in this study and in previous research (MacDougall, 2015; MacDougall and Knutti, 2016; once you Gasser et al., 2018), constitutes a significant challenge in accurately estimating the remaining carbon budgets consistent with use different terminology.

This study aims to serve as a meaningful supplement to the limited existing research on permafrost response and feedback under temperature stabilization and overshoot scenarios. For example, CMIP6 includes only one overshoot scenario (SSP5-3.4-OS), which is insufficient for comprehensively analyzing the potential contribution of permafrost feedback under different global warming levels (Melnikova et al., 2021). The upcoming CMIP7 is planned to encompass a broader range of overshoot scenarios, from very low to medium and high overshoot scenarios. This expansion aims to delve into the potential for climate restoration, the feasibility of achieving Paris Agreement targets, and the risks of irreversibility and hysteresis in the slower components of the Earth system from beyond 2125 to 2300 (WRCP, 2024). Our study investigated the permafrost response and feedback under overshoot scenarios with varying levels of warming, which can be used for comparative analysis of the upcoming CMIP7 multi-model results. Additionally, research on permafrost changes under different climate stabilization scenarios is also limited. Some of existing studies are based on equilibrium permafrost models (e.g. Liu et al., 2021), which do not consider transient effects and cannot predict the timing of permafrost loss based on imposed warming scenarios (Smith et al., 2022). Our study utilizes a process-based model and provides insights into the limitate medianear relationship/between permafrost degradation and global warming.

This study demonstrates a method to quantify the permafrost carbon feedback under specified global warming levels or warming trajectories with climate-carbon cycle fully coupled Earth system model driven by pre-designed CO2 emission pathways. As permafrost carbon feedback is highly sensitive to permafrost states and processes, controlling temperature or warming trajectory is beneficial to isolate individual contributions from climate condition and permafrost carbon processes. Due to tight link between terrestrial carbon feedback and physical climate feedback, Goodwin (2019) found around half the uncertainty in derived terrestrial carbon feedback originates from uncertainty in the physical climate feedback. Therefore, controlling global warming level or warming trajectory might be helpful to accurately assess a specific aspect of terrestrial carbon feedback by isolating the uncertainty from the physical climate feedback. We encourage the exploration of potential application for this method in quantifying other aspects of carbon-climate feedbacks, especially for model intercomparison.

Study. Of course, he findings of this study are based on a single Earth system model, and it motivates further studies aimed 710 at comprehensively understanding of permafrost response and feedback and their impacts on Arctic communities, since your paper doosn't address this. infrastructure, and climate policies.

Data availability

The source code of the UVic ESCM version 2.10 can be downloaded from https://terra.seos.uvic.ca/model/2.10/. All simulation data utilized in this study are publicly accessible and can be obtained from Cui. (2024).

**715 Author contributions**

DJ designed the research and set up the simulation. MC conducted the simulation and interpreted the data. MC and DJ wrote the initial manuscript. DJ, MC and YC revised the manuscript.

**Competing interests**

The authors declare that they have no conflict of interest.

**720 Acknowledgements**

The authors thank the Super Computing Center of Beijing Normal University and the National Key Scientific and Technological Infrastructure project "Earth System Numerical Simulation Facility" (EarthLab) for providing computing resources.

**Financial support**

This research has been supported by the National Natural Science Foundation of China (No. 41875126).

**References**

Anderson, K., Buck, H. J., Fuhr, L., Geden, O., Peters, G. P., and Tamme, E.: Controversies of carbon dioxide removal, Nat Rev Earth Environ, 4, 808-814, https://doi.org/10.1038/s43017-023-00493-y, 2023.

Armstrong McKay, D. I., Staal, A., Abrams, J. F., Winkelmann, R., Sakschewski, B., Loriani, S., Fetzer, I., Cornell, S. E., Rockström, J., and Lenton, T. M.: Exceeding 1.5°C global warming could trigger multiple climate tipping points, Science, 377, eabn7950, https://doi.org/10.1126/science.abn7950, 2022.

---

## Author Response (AR2)

**Dear Vivek,**

We are sincerely grateful for the time you devoted to annotating our manuscript. In response, we have revised every section where you requested clarification or additional context; your comments have been invaluable in raising the manuscript's quality.

First, every language issue you raised—grammar, terminology, word choice, and unclear phrasing—has been corrected in the revised manuscript. Below, we elaborate on several key points; your handwritten comments appear in black, and our responses are shown in blue.

1) Abstract: Is this a generalized statement or based on the simulations presented here?

Thank you for your comment. The statement is based on the simulations presented in this study. We have revised the sentence as follows: "Based on the simulations presented, permafrost carbon feedback is unlikely to initiate a self-perpetuating global tipping process under both stabilization and overshoot scenarios."

2) Line 200: Is it possible to report there in time scale years?

Yes. We have converted the passive carbon pool transformation rate from s-1 to its corresponding mean residence time in years and clarified its temperature dependence:

"The passive carbon pool transformation rate was estimated from the  $^{14}$ C age of the passive carbon pool in midlatitude soils (Trumbore, 2000). Its MRT at 5 °C is 300 to 5000 years, yielding a passive carbon pool transformation rate of  $0.25 \times 10 - 10$  to  $4 \times 10^{-10}$  s-1 after adjustment to 25 °C. The MRT of the passive carbon pool was assumed to follow a uniform distribution (MacDougall and Knutti, 2016)."

3) In Figure 3, panel c) = permafrost C loss is this frozen C? panel d) = permafrost region C loss is this all (frozen + unfrozen) C? If yes, please change the wording in figure caption to make this clear.

Panel (c) shows the loss of permafrost carbon only; usual soil carbon can also be frozen, but its loss is excluded here. Panel (d) shows the total soil carbon loss from the permafrost region, encompassing both permafrost carbon and usual soil carbon. To clarify this distinction, we have revised the manuscript's terminology, replacing "permafrost region soil carbon" with "total permafrost region soil carbon" and have updated the figure caption accordingly.

**4) Line 297: Are you comparing Figures 5a and 5b?**

Yes, we are comparing Figures 5a and 5b in this context. To clarify this point, we have revised the sentence to: "Our simulations show that permafrost carbon inputs (Fig. 5a) do not follow the same trajectory as total permafrost region soil carbon inputs (Fig. 5b), especially under overshoot scenarios. Permafrost carbon inputs continue rising for some time after peak warming, whereas total permafrost region soil carbon inputs closely track the temperature trajectory (Fig. 3a)."

Yes, this refers to Figure 3d. The figure reference has been added in the revised manuscript.

6) Line 335: Is this over "permafrost region"? How does this compare to pre-industrial?

Yes. The percentage refers to the combined areal coverage of trees and shrubs within the permafrost region, and the 1.5 °C warming is measured relative to the pre-industrial level (1850-1900). The permafrost region with permafrost carbon changes only minimally in all simulations. We have revised the sentence for clarity: "Upon 1.5 °C warming (projected for the 2040s relative to 1850–1900), trees and shrubs are expected to cover approximately 62 % of the area where permafrost carbon is non-zero......".

Meanwhile, we have clarified the distinction between "permafrost area" and "permafrost region" in the first paragraph of Section 3 ("Results"), since "permafrost region" is used when discussing permafrost carbon and related fluxes: "the permafrost area is defined as the area where soil temperature remains below 0 °C for at least two consecutive years, whereas for carbon-related variables the permafrost region is defined as the area where permafrost carbon exceeds zero in UVic ESCM. The area with non-zero permafrost carbon is 17.2 million km² and changes only minimally throughout the simulations."

7) As we discussed in our emails, please clarify "permafrost soil C" vs "permafrost region soil C", and that the "region" area is changing.

Thank you for the comment. To clarify the distinction, we have revised the terminology throughout the manuscript by replacing "permafrost region soil carbon" with "total permafrost region soil carbon". Regarding the changing permafrost "region" area, we have clarified this point at the end of the first paragraph in the Section 3 ("Results").

8) Figure 6: Are there no or very few broadleaf trees in the permafrost region and that's why they are not shown here?

Yes. Broadleaf trees are absent and C4 grasses have negligible coverage in the permafrost region, so they are omitted from Figure 6. We have clarified this in the caption: "Notably, broadleaf trees and C4 grasses are not shown because their presence in the permafrost region is negligible."

9) Line 393: But you just said additional warming is higher in stabilization scenarios.

Thank you for pointing this out. The original sentence sought to state that, by 2300, the additional warming in stabilization scenarios is 22–56 % greater than in their overshoot counterparts, but its wording was ambiguous. We have now revised it for clarity: "By 2300, the additional warming under SWL-2, SWL-3, and SWL-4 exceeds that of OS-2, OS-3, and OS-4 by 0.03 [0.01 to 0.04] °C, 0.07 [0.02 to 0.11] °C and 0.07 [0.03 to 0.10] °C, respectively, which is 22 % to 56 % higher than in the corresponding overshoot scenarios."

10) Figure 7: What's the difference between (c) and (d)? They're the same y-axis label and description in the figure caption. Oh I see! (c) is overshoot scenarios, and (d) is stabilization. Pls make this clear in the caption.

Panels (c) and (d) indeed both showed the additional permafrost area loss due to permafrost carbon feedback, but under different scenarios. To avoid confusion and ensure consistency with other figures, we have now combined them into a single panel (c) that displays all scenarios together.

11) Line 519: Is this also from Canadell et al. 2021

No, this is from Nitzbon et al. (2024), which has now been added to the revised manuscript.

12) Line 577: But in Fig 10a and b changes are relatively small until about 1.0 °C

We agree. Our intention was to highlight that the strongest permafrost degradation occurs just below 1.5 °C warming level. We have revised the sentence to clarify this point as follows: "The maximal sensitivity occurring just below 1.5 °C global warming level suggests the fastest permafrost degradation is anticipated to take place within Paris Agreement's warming levels."

13) Line 680: hopefully this will become more clear once you use different terminology.

Thank you for the suggestion. we have replaced all instances of "carbon release" with "carbon loss" throughout the revised manuscript.

---

## Editor Decision (ED2)

**Permafrost response and feedback under temperature stabilization and overshoot scenarios with different global warming levels**

Min Cui1, Duoying Ji1 and Yangxin Chen1

+Faculty State Key Laboratory of Earth Surface Processes and Disaster Risk Reduction, Faculty of Geographical Science, Beijing Normal University, Beijing, 100875, China

Correspondence to: Duoying Ji (duoyingji@bnu.edu.cn)

**Abstract.**

Permafrost regions in the northern high latitudes face faces significant degradation risks under global warming and threater the achievement of global climate goals. This study explores the response and feedback of permafrost and the associated carbon loss under temperature stabilization scenarios, where the global mean temperature stabilizes at various global warming levels, and overshoot scenarios, where the global mean temperature temporarily exceeds the 1.5 °C warming target-before returning. Under the 1.5 °C and 2 °C stabilization scenarios, permafrost area is projected to decrease by 4.6 [4.5 to 4.7] million km2 and 6.6 [6.4 to 6.8] million km2, respectively, from a pre-industrial level of 17.0 million km2. Corresponding permafrost carbon losses are estimated at 54 [32 to 79] PgC and 72 [42 to 104] PgC, relative to a pre-industrial carbon stock of 484 [383 to 590] PgC. In overshoot scenarios, permafrost area shows effective recovery, with additional losses of only 0.6 [0.3 to 1.1] million km2 compared to the 1.5 °C stabilization scenario. In contrast, permafrost carbon loss remains largely irreversible, with additional loss of 24 [4 to 52] PgC compared to the 1.5 °C stabilization scenario. Both stabilization and overshoot scenarios show that additional warming due to permafrost carbon feedback rises with higher global warming levels, and the most substantial feedbackadditional warming in overshoot scenarios is anticipated becomes most pronounced during the cooling phase. The additional permafrost area loss due to permafrost carbon feedback, which accounts for 5 [2 to 11] % of the total loss, is influenced by both the magnitude of additional warming and the sensitivity of permafrost area to global warming. Moreover, the responses of permafrost area, permafrost carbon, and associated radiative forcing to a broad range of global warming exhibit near-linear relationships under stabilization scenarios. PermafrostBased on the simulations presented, permafrost carbon feedback is unlikely to initiate a self-perpetuating global tipping process under both stabilization and overshoot scenarios. These findings have significant implications for long-term climate change and mitigation strategies.

**1 Introduction**

Permafrost soils in the northern high latitudes contain an estimated 1100-1700 Pg of carbon, primarily in the form of frozen organic matter, which is roughly twice the amount of carbon in the atmosphere (Hugelius et al., 2014; Schuur et al., 2015). As the climate warms, both the gradual orand abrupt permafrost thaw processes, along with subsequent microbial decomposition, would release carbon dioxide (CO2) and methane (CH4) into the atmosphere, thereby amplifying the warming effect (Koven et al., 2011; Feng et al., 2020; Smith et al., 2022). The positive feedback mechanism, combined with the fact that warming rates in the Arctic exceed the global average (Fyfe et al., 2013; Liang et al., 2022; Rantanen et al., 2022), underscores the critical role of permafrost as a key tipping element in the climate system (Armstrong McKay et al., 2022). However, current Earth system models inadequately represent or omit the permafrost carbon processes, which has become one of the largest sources of uncertainty in future climate projections (Schädel et al., 2024). Therefore, researching the release of permafrost carbon loss and its feedback on the climate system is crucial for accurately assessing climate risks and formulating effective emission reduction strategies.

The Paris Agreement aims to limit global average temperature rise to well below 2 °C above pre-industrial levels, with efforts to keep it below 1.5 °C. Despite these goals, global warming has already exceeded 1 °C and is on track to surpass 3 °C by the end of the 21st century, primarily due to increased anthropogenic CO2 emissions (Haustein et al., 2017; Hausfather and Peters, 2020). If current emission rates persist, the remaining carbon budgets compatible with the 1.5 °C target will be critically tight and likely exhausted within the next few years (Rogelj et al., 2015; Goodwin et al., 2018; Masson-Delmotte et al., 2018; Forster et al., 2023; Smith et al., 2023). It is unlikely that the 1.5 °C target set by the Paris Agreement will be met (Raftery et al., 2017). However, it might still be achievable after a period of temperature overshoot, by compensating for excessive past and near-term emissions with net-negative emissions at a later time – i.e., through on-site CO2 capture at emission sources and carbon dioxide removal from the atmosphere (Gasser et al., 2015; Sanderson et al., 2016; Seneviratne et al., 2018; Drouet et al., 2021; Schwinger et al., 2022).

Several existing studies have assessed the climate response to overshoot pathways. Many components of the physical climate system have been identified as reversible, although typically with some hysteresis behavior (Boucher et al., 2012; Wu et al., 2015; Tokarska and Zickfeld, 2015; Li et al., 2020; Cao et al., 2023). In this context, reversibility refers to a partial recovery of climate conditions in an overshoot scenario toward an Earth system state without overshoot. These studies demonstrate that global mean temperature, sea surface temperature, and permafrost area can recover within decades to centuries in response to net negative emissions. Carbon releaseloss from permafrost has been shown to be irreversible on multidecadal to millennial timescales (MacDougall et al., 2013; Schwinger et al., 2022). The presence or absence of hysteresis effect in the permafrost processes is influenced by multiple factors, including the thermal inertia of permafrost soils, potential shifts in vegetation composition, and the extent to which irreversible permafrost carbon losses are offset by gains in vegetation and non-permafrostusual soil carbon reservoirs (MacDougall, 2013; Schwinger et al., 2022). Furthermore, the soil carbon loss under overshoot scenarios significantly affects the hydrological and thermal properties of soils (Zhu et al., 2019), which in turn modulate the processes involved. The interactions between physical and biophysical processes can potentially stabilize the carbon, water, and energy cycles at distinct post-overshoot equilibria (de Vrese and Brovkin, 2021). Therefore, a temporary warming of the permafrost regions entails important legacy effects and lasting impacts on its physical state and carbon cycle. However, these existing studies have yet to assess permafrost carbon feedback under overshoot scenarios.

The terrestrial component of UVic ESCM v2.10 uses the Top-down Representation of Interactive Foliage and Flora Including Dynamics (TRIFFID) vegetation model to describe the states of five plant functional types (PFT): broadleaf tree, needleleaf tree, C3 grass, C4 grass, and shrub (Cox, 2001; Meissner et al., 2003). A coupled photosynthesis-stomatal conductance model is used to calculate carbon uptake via photosynthesis, which is subsequently allocated to vegetation growth and respiration. The resulting net carbon fluxes driveprimary productivity drives changes in vegetation characteristics, including areal coverage, leaf area index, and canopy height for each PFT. The UVic ESCM v2.10 utilized in this study does not account for nutrient limitations in the terrestrial carbon cycle, leading to an overestimation of global gross primary productivity and an enhanced capacity of land to take up atmospheric carbon (De Sisto et al., 2023). However, the model reasonably represents the dominant PFTs of C3 grass, shrub, and needleleaf tree at northern high latitudes, although it underestimates vegetation carbon density over this area (Mengis et al., 2020).

The UVic ESCM v2.10 represents the terrestrial subsurface with 14 layers, extending to a total depth of 250.3 m to correctly capture the transient response of permafrost on centennial timescales. The top eight layers (10.0 m) are involved in the hydraulie cyclehydrologically active, while the deeper layers are modeled as impermeable bedrock (Avis et al., 2011). The carbon cycle is active in the top six layers (3.35 m), where organic carbon from litterfall, simulated by the TRIFFID vegetation model, is allocated to soil layers with temperatures above 1 °C according to an exponentially decreasing function with depth. If all soil layers are below 1 °C, the organic carbon is added to the top soil layer. The soil respiration is calculated for each layer individually as a function of temperature and moisture, but the respiration ceases when the soil layer temperature falls below 0 °C (Meissner et al., 2003; Mengis et al., 2020). In regions where permafrost exists defined as areas where soil temperature remains below 0 °C for at least two consecutive years. In permafrost regions the model applies a revised diffusion-based cryoturbation scheme to redistribute soil carbon within the soil column. Compared to the original diffusionbased cryoturbation scheme proposed by Koven et al. (2009), the revised cryoturbation scheme calculates carbon diffusion using an effective carbon concentration that incorporates the volumetric porosity of the soil layer, rather than the actual carbon concentration, thereby resolving the disequilibrium problem of the permafrost carbon pool during model spin-up (MacDougall and Knutti, 2016). However, as the UVic ESCM v2.10 only simulates permafrost carbon in the top 3.35 m of soil, the current cryoturbation scheme cannot initiate the formation of Yedoma. As a result, soil carbon stored in deep deposits of Yedoma regions is omitted in our simulations.

In the UVic ESCM v2.10, the usual soil carbon and the permafrost carbon are depicted as two distinct depth-resolved carbon pools within the upper six soil layers. Soil carbon that is transported diffused downward and crosses the permafrost table (i.e. the depth of the active layer) is transformed into permafrost carbon. Conversely, permafrost carbon that is moveddiffused upward and crosses the permafrost table is converted back into usual soil carbon. However, as the active layer deepens, thawed permafrost carbon is not transferred to the usual soil carbon pool; it remains in the permafrost carbon pool and decomposes at a rate distinct from that of usual soil carbon. Permafrost carbon can only be decomposed into CO2, as the UVic ESCM v2.10 does not include a methane production module (MacDougall and Knutti, 2016). The permafrost carbon pool is characterized by four key parameters: (1) a decay rate constant; (2) the available fraction of the pool for decomposition,

I'm confused again. This indicales 3 pools.

permatrost that has thousand permatrost C that has

Figure 1. Temperature stabilization and overshoot scenarios designed through a simple proportional control scheme (Zickfeld et al., 2009) on CO2 emission and UVic ESCM v2.10 with permafrost carbon module deactivated (corresponding to the NPFC simulations).

(a) Accumulated Cumulative CO2 emission and (b) global warmingtemperature change relative to the pre-industrial levels (1850-1900), in the historical and SSP5.8.5 scenario (black), the stabilization (dashed lines) and overshoot (solid lines) scenarios at for 1.5 °C (blueorange), 2.0 °C (green linna), 3.0 °C (red)green), and 4.0 °C (purple) global warming levels.

To evaluate the uncertainty in permafrost carbon response under stabilization and overshoot scenarios, we perturbed the four key permafrost carbon parameters following the methodologies of MacDougall and Knutti (2016) and MacDougall (2021). The permafrost carbon decay constant was derived from the mean residence time (MRT) of the slow soil carbon pool at 5 °C in permafrost soils and adjusted to reflect decay at 25 °C using the method proposed by Kirschbaum (2006). The probability distribution function of mean residence time was taken as a normal distribution with a mean of 7.45 years and a standard deviation of 2.67 years from Schädel et al. (2014). The available fraction of permafrost carbon for decomposition was derived from the size of the fast, slow, and passive soil organic carbon pools separately for organic, shallow mineral, and deep mineral soils measured by Schädel et al. (2014). The probability distribution function of available fraction of permafrost carbon for decomposition was described by weighted gamma distributions, with each distribution respectively describing the probability distribution function of available fraction of permafrost carbon for decomposition in organic, shallow mineral, and deep mineral soils. The passive carbon pool transformation rate was estimated from the 14C age of the passive carbon pool from midlatitude soils (Trumbore, 2000). Its mean residence time at 5 °C is 300 to 5000 years, yielding an estimated value

Not defined yet passive carbon pool transformation rate of 0.25×10-10 to 4×10-10 s-1 (Trumbore, 2000). after adjustment to 25 °C. The probability distribution function mean residence time of the passive carbon pool transformation rate was taken as assumed to follow a uniform in base two logarithmic spacedistribution (MacDougall and Knutti, 2016). For the initial quantity of total permafrost region soil carbon, its probability distribution function was taken as a normal distribution with a mean of 1035 PgC and a standard deviation of 75 PgC, informed by Hugelius et al. (2014). A series of 5,000-year sensitivity runs were performed under preindustrial steady conditions with varying saturation factors to determine their relationship with the quantity of the total permafrost region soil carbon bool. This relationship was then utilized to tune the total permafrost region soil carbon pool, ensuring alignment with observational data (Hugelius et al., 2014). Fig. 2 illustrates the probability distribution function for each perturbed parameter. MacDougall and Knutti (2016) and MacDougall (2021) additionally perturbed two physical climate parameters controlling climate sensitivity and Arctic amplification, but they are not perturbed in this study due to their limited influence on global mean temperature in stabilization scenarios.

The Latin hypercube sampling method (McKay et al., 1979) was used to explore the effects of parameter uncertainty on projections of permafrost carbon change. In this study, the probability distribution function of each key permafrost carbon parameter was divided into 25 intervals of equal probability. One value was randomly selected from each interval for a given parameter, and then randomly matched with values of the other three key parameters selected in the same manner to generate parameter sets. This sampling procedure was repeated 10 times, resulting in 250 unique parameter sets (i.e., 250 model variants). For each parameter set, the UVic ESCM v2.10 was first run through a 10,000-year spin-up phase under pre-industrial conditions to achieve a quasi-equilibrium state. For these spin-up runs, the atmospheric CO2 concentration was fixed at 284.7 ppm and the solar constant was set to 1360.747 W m-2. Following the spin-up, emission-driven transient simulations were conducted under the stabilization, overshoot, and SSP5-8.5 scenarios. The results are presented as the median across all model variants, with uncertainty quantified as the range between the 5th to the 95th percentiles.

Figure 2. Probability distribution functions of the four key permafrost carbon parameters perturbed in the UVic ESCM v2.10 to represent uncertainty in permafrost carbon response. Panel (d) employs a logarithmic scale on the horizontal axis to better illustrate the distribution of the corresponding parameter. This figure is reproduced This figure is adapted from MacDougall (2021).

**3 Results**

The UVic ESCM v2.10 reliably simulates historical temperature changes, permafrost area, and the partitioning of anthropogenic permafrost carbon-emissions among the atmosphere, ocean and land, and total permafrost region soil carbon stocks. Under pre-industrial conditions, the simulated Northern Hemisphere permafrost area, defined as regions where the soil layer remains perennially frozen for at least two consecutive years, was 17.01 [17.00 to 17.04] million km², the simulated total soil carbon stock in the permafrost regions was 1031 [915 to 1149] PgC, of which 484 [383 to 590] PgC was classified as perennially frozen permafrost carbon and 547 [533 to 559] PgC was classified as usual soil carbon. For the period 1960–1990, the model simulated Northern Hemisphere permafrost area at 16.8 [16.7 to 16.9] million km², which falls within the reconstructed range from 12.0 to 18.2 million km² (Chadburn et al., 2017) and the observation derived extent from 12.21 to

"total perma forst region soil carbon" on page 8.

16.98 million km² (Zhang et al., 2000). Additionally, the simulated total soil carbon stock in the top 3.35 m of permafrost regions for this same period was 1034 [919 to 1151] PgC, with 483 [382 to 587] PgC classified as perennially frozen permafrost carbon, accounting for 47% [42% to 51%] of the total permafrost region soil carbon stock, in agreement with Hugelius et al. (2014). Note that the permafrost area is defined as the area where soil temperature remains below 0 °C for at least two consecutive years, whereas for carbon-related variables the permafrost region is defined as the area where permafrost carbon exceeds zero in UVic ESCM. The area with non-zero permafrost carbon is 17.2 million km² and changes only minimally throughout the simulations.

The UVic ESCM v2.10 also realistically simulates historical temperature changes and the partitioning of anthropogenic carbon emissions among the atmosphere, ocean and land. During the period 2011–2020, the model estimated a global mean temperature increase of 1.14 [1.13 to 1.15] °C relative to preindustrial levels, which is closely aligned with the observed rise of 1.09 [0.91 to 1.23] °C (Gulev et al., 2021). From 2010 to 2019, the model estimated that anthropogenic carbon emissions of 11 PgC yr-1 were distributed as follows: 5.5 [5.4 to 5.6] PgC yr-1 to the atmosphere, 3.0 [2.98 to 3.03] PgC yr-1 to the ocean, and 2.5 [2.4 to 2.6] PgC yr-1 to terrestrial ecosystems. These estimates are broadly consistent with the global anthropogenic CO2 budget assessment by the Global Carbon Project (GCP) with figures of 5.1±0.02 PgC yr-1 for the atmosphere, 2.5±0.6 PgC yr-1 for the ocean, and 3.4±0.9 PgC yr-1 for terrestrial ecosystems (Friedlingstein et al., 2020).

**3.1 Permafrost Response**

The Northern Hemisphere high-latitude permafrost area is strongly correlated with changes in global mean temperature (Fig. 3a, b; Fig. S1a, b). As global warming increases from 1.5 °C to 4 °C, the permafrost area declines from 13.9 to 8.3 million km2 under the SSP5-8.5 scenario, by year 2300. In the stabilization scenarios, when global warming is stabilized at the Paris Agreement targets of 1.5 °C or 2 °C, permafrost degradation is effectively suppressed compared to the SSP5-8.5 scenario. By 2300, permafrost area decreases by 4.6 [4.5 to 4.7] million km2 and 6.6 [6.4 to 6.8] million km2 from the pre-industrial level of 17.0 million km2 under SWL-1.5 and SWL-2 scenarios, respectively, accounting for 39 [38 to 40] % and 56 [54 to 58] % of the reduction observed under SSP5-8.5 scenario. The incremental additional permafrost degradation under the SWL-3 compared to SWL-2 is significantly larger than that under SWL-4 compared to SWL-3. This is mainly because the remaining permafrost available for degradation becomes progressively limited under higher global warming levels, and the permafrost area under higher stabilization scenarios has not yet reached a steady state in our simulations. Additionally, the permafrost area under the SWL-3 and SWL-4 scenarios exceeds that under the SSP5-8.5 scenario by only 2.0 [1.9 to 2.1] million km2 and 0.9 [0.8 to 1.0] million km2 by 2300, respectively. This is primarily because the permafrost area in the SSP5-8.5 scenario is smaller and, as a transient simulation, is further from equilibrium compared to SWL-3 and SWL-4. However, during the cooling phase of the overshoot scenarios, as the global mean temperature returns to 1.5 °C above pre-industrial levels, the permafrost area gradually recovers to that under the SWL-1.5 scenario (Fig. 4a; Fig. S2a). By 2300, it converges to similar levels of 11.1~12.4 million km2 under the OS-2, OS-3 and OS-4 scenarios, with an additional loss of only 0.2~1.2 million km2 compared to the SWL-1.5 scenario. This indicates that permafrost area is nearly reversible and largely follows the global mean to the top soil layer. Meanwhile, permafrost carbon and non-permafrost susual soil carbon are both represented as depth-resolved carbon pools within the top six soil layers. The movement of permafrost carbon due to cryoturbation mixing is parameterized as being proportional to the gradient of total soil carbon with depth. Soil carbon that diffuses downward through the permafrost table is converted to permafrost carbon. During the cooling phase of overshoot scenarios, increased litterfall and a rising permafrost table lead to elevated carbon concentrations in surface soil layers, resulting in enhanced vertical diffusion and a surge in permafrost carbon inputs. Conversely, under the SSP5-8.5 scenario, permafrost carbon inputs exhibit only a minor peak around the 2150s, followed by a sharp decline (Fig. 5a; Fig. S3a). This is due to the continuous reduction in permafrost area and the deepening of the permafrost table, both of which reduce carbon concentrations in the upper soil layers and weaken vertical diffusion, despite the increasing litter flux under a strong CO2 fertilization background. We note that the approach adopted in the model may not accurately describe natural processes of vertical carbon movement, which are influenced by soil porosity heterogeneity, freeze-thaw cycles, and ice expansion upon freezing.

Permafrost Total permafrost region soil carbon shows a strong tendency to recover after temperature overshoot-(Fig. 3d). In contrast, total permafrost region soil carbon continues to decrease under temperature stabilization scenarios, but the rate of decrease is gradually slowing down. The total permafrost region soil carbon releaseloss in the overshoot scenarios is significantly mitigated compared to stabilization scenarios at same global warming levels (Fig. 3d; Fig. S1d). By 2300, total permafrost region soil carbon losses under OS-2, OS-3, and OS-4 scenarios are projected to be 41 [15 to 72], 61 [29 to 98], and 81 [44 to 124] PgC, respectively, with reductions of 14 [8 to 21], 41 [25 to 57], and 62 [41 to 82] PgC compared to the SWL-2, SWL-3, and SWL-4 scenarios, respectively. During the stabilization phase in the overshoot scenarios, additional total permafrost region soil carbon losses compared to the SWL-1.5 scenario continue to decrease, but the total permafrost region soil carbon in the overshoot scenarios does not fully recover to the level under the SWL-1.5 scenario by 2300 (Fig. 4c; Fig. S2c). Notably, in all stabilization and overshoot scenarios simulated in this study, the permafrost region soil serves as a net cumulative carbon source for atmospheric CO2 by 2300. However, during the stabilization phase of OS-3 and OS-4, the permafrost region soil turns into a carbon sink, as soil carbon inputs surpass the reduced decomposition activity due to the depletion of soil carbon stocks and reduced warming levels. Permafrost Total permafrost region soil carbon inputs primarily originate from-robust biophysical processes related to vegetation litterfall, with their intensity influenced by warming levels and CO2 fertilization effects, while total permafrost region soil carbon decomposition is closely tied to global mean temperature. In the overshoot scenarios, the peak for total permafrost region soil carbon inputs occurs slightly later than the global mean temperature peak, whereas the peak for total permafrost region soil carbon decomposition happens marginally earlier than the global mean temperature peak (Fig. 5b, d; Fig. S3b, d).

The total permafrost region soil carbon inputs generally track the trajectory of litter flux across the same area, with an approximate delay of 10-20 years (not shown). To attribute the contribution of total permafrost region soil carbon inputs, we examined how dominant vegetation types (needleleaf tree, C3 grass and shrub) over the permafrost region adapt to temperature and atmospheric CO2 concentrations in both overshoot and stabilization scenarios (Fig. 6). Needleleaf trees expand slowly and continuously in the permafrost region in both overshoot and stabilization scenarios, whereas that of shrubs closely follows the ishet does that's refers to?

**simulations? Why the use of "expected"?**

trajectory of global mean temperature. The combined areal coverage of trees and shrubs is projected to cover about 62% upon Upon 1.5 °C warming (projected for the 2040s relative to pre-industrial levels around 2040s 1850–1900), trees and shrubs are expected to cover approximately 62% of the area where permafrost carbon is non-zero, slightly higher than the 24~52% range projected for 2050 using a statistical approach that links climate conditions to vegetation types under two distinct emission trajectories (Pearson et al., 2013). During the warming and cooling phases of overshoot scenarios, the expansion and reduction of shrubs correspond with the degradation and expansion of C3 grasses, respectively. Among the three dominant PFTs, only shrubs show a nearly reversible response in areal coverage, net primary productivity (NPP) and vegetation carbon with respect to global mean temperature under overshoot scenarios. In contrast, the continuous reduction of C3 grasses and the expansion of needleleaf trees suggest a degree of irreversibility in the structure and vegetation carbon density of northern high latitude terrestrial ecosystems under overshoot scenarios. Our results are in line with an earlier study by Tokarska and Zickfeld (2015), but contrast with Schwinger et al. (2022) who reported only minor differences in vegetation carbon after the overshoots compared to the reference simulation with no overshoot by prescribing vegetation distributions. In our study, the shifts in vegetation composition and changes in living biomass, especially those associated with woody vegetation, are key drivers of total permafrost region soil carbon inputs.

The uncertainty in total permafrost region soil carbon releaseloss is nearly the same as that of permafrost carbon releaseloss (Fig. 3c, d; Fig. S1c, d). For example, the 5th to 95th percentile range of total permafrost region soil carbon releaseloss under the OS-2 and OS-4 scenarios is 58 PgC and 81 PgC respectively by 2300, compared to 52 PgC and 72 PgC for permafrost carbon releaseloss. This indicates that the uncertainty in total permafrost region soil carbon releaseloss is largely driven by the uncertainty in permafrost carbon releaseloss. Therefore, we evaluate the relative importance of perturbed permafrost carbon parameters on total permafrost region soil carbon releaseloss under different temperature pathways through calculating their correlations across all ensemble simulations. The influence of model parameters on the uncertainty of permafrost carbon losses by 2300 is relatively consistent across the SSP5-8.5, OS-4, and SWL-4 scenarios, with the strongest correlations observed for the permafrost passive carbon pool transformation rate (R=0.81~0.85), followed by the initial quantity of total permafrost region soil carbon (R=0.55~0.61). This finding aligns with Ji et al. (2024), who highlights the critical role of these two parameters in the uncertainty of total permafrost region soil carbon loss under temperature overshoot and 1.5 °C warming stabilization scenarios.

to result in enough additional thawing and corresponding carbon emissions to initiate a self-perpetuating tipping process. Since this study only models the gradual thawing of permafrost through the deepening of the active layer, we cannot rule out the possibility of tipping points associated with the abrupt thawing of talik development, thermokarst and thermo-erosion processes.

Figure 7. Additional stress in (a) radiative forcing, (b) global warming and (c, d) termafrost area due to permafrost carbon feedback, calculated as the difference between the PFC and NPFC simulations. ShownResults are results for the shown under stabilization (dashed lines) and overshoot (solid lines) scenarios at for 1.5 °C (blue orange) 2.0 °C (green sienna), 3.0 °C (red) green), and 4.0 °C (purple) global warming levels, along with the SSP5-8.5 scenario (black). Square markers indicate the time points when the temperature overshoot reaches its peak or stabilized warming begins, while circle markers indicate when the overshoot returns to 1.5 °C. Results represent the ensemble median of 250 simulations. Dots on the adjacent right panels represent values in the year 2300, with uncertainty ranges estimated as the 5th to 95th percentiles. In panel (a), the additional radiative forcing is calculated

we don't have panel d'any more

loss and associated radiative forcing exhibit a nearly linear response to increasing global warming levels, especially above l °C, for both stabilization and SSP5-8.5 scenarios (Fig. 10b, c).

Meanwhile, the sensitivities of permafrost area, permafrost carbon loss, and associated radiative forcing to global warming under stabilization scenarios are all stronger than those under the SSP5-8.5 scenario. Specifically, based on the simulated permafrost area in the year 2300 under stabilization scenarios with global warming levels between 1.5 °C and 3 °C, the sensitivity of permafrost area to global warming is -3.19 [-3.01 to -3.36] million km2 °C-1. In comparison, a linear fit of permafrost area change against global warming levels over the same temperature range in the SSP5-8.5 scenario yields a sensitivity of -2.85 [-2.77 to -2.89] million km2 °C-1. Similarly, the permafrost carbon feedback per degree of global warming derived from a linear fit based on the total permafrost carbon loss in the year 2300 under stabilization scenarios, is -27.6 [-16.5 to -38.2] PgC °C-1. In contrast, the corresponding value under the SSP5-8.5 scenario, estimated from a linear fit over the 1.5 °C to 4.0 °C warming range, is -19.3 [-15.7 to -24.1] PgC °C-1. Applying the same approach, the associated radiative forcing per degree of global warming is estimated to be 0.08 [0.05 to 0.12] W m-2 °C-1 for the stabilization scenarios and 0.04 [0.03 to 0.05] W m-2 °C-1 for the SSP5-8.5. These differences between the stabilization and SSP5-8.5 scenarios are mainly attributable to the differing response time scales represented by the two scenarios: SSP5-8.5 reflects a typical transient response, while the stabilization scenarios maintain stabilized temperatures over extended periods and thus approximate a quasi-equilibrium response of the climate-carbon system. Furthermore, the smaller sensitivity of permafrost radiative forcing per degree of global warming under the SSP5-8.5 can be partially attributed to its higher background atmospheric CO2 concentration compared to the stabilization scenarios. The same amount of CO2 emissions would produce smaller additional radiative forcing under a higher background atmospheric CO2 concentration, due to the logarithmic relationship between CO2 concentration and radiative forcing (Etminan et al., 2016).

To a certain extent, our findings align with those of Nitzbon et al. (2024), who suggested that the accumulated response of Arctic permafrost to climate warming is approximately quasilinear. Nitzbon et al. (2024) reported a quasilinear decrease in the equilibrium permafrost extent to global warming, with a rate of approximately 3.5 million km² °C⁻¹. This quasilinear relation holds for global warming ranges from 0 °C to 4 °C, derived from the empirical relationship between the local permafrost fraction and the annual mean global temperature. However, our results indicate the quasilinear relationship only holds for global warming levels between 1.5 °C and 3 °C. Furthermore, the permafrost carbon feedback and the associated radiative forcing per degree of warming, as derived from our simulations of both stabilization and SSP5-8.5 scenarios, are within the ranges of -18 [-3.1 to -41] PgC °C⁻¹ and 0.09 [0.02 to 0.20] W m⁻² °C⁻¹, respectively, reported by Canadell et al. (2021). Our estimates also align with the estimated range of equilibrium sensitivity of permafrost carbon decline to global warming, which is -21 [-4 to -48] PgC °C⁻¹ (Nitzbon et al., 2024). This may represent an upper limit for bound on the permafrost carbon feedback per degree of global warming—considering that the estimated reduction in permafrost: the estimate reflects the maximum carbon does not equate directly tothat could be lost from frozen soils, yet only a fraction of that carbon emissions is ultimately released into the atmosphere, as noted by Nitzbon et al. (2024).

C lost from frozen & rils # C entering the atmosphere why?

**4 Conclusions and Discussion**

This study utilizes the intermediate-complexity Earth system model UVic ESCM v2.10 and perturbed parameter ensemble modelling approach to study the response and feedback of permafrost under temperature stabilization and overshoot scenarios across different global warming levels. The UVic ESCM v2.10 has been validated against observational and reconstructed datasets, demonstrating its ability to reproduce historical observation-based permafrost area and permafrost carbon stocks: for the present day. In addition to presenting the changes in permafrost area, permafrost carbon, and total permafrost region soil carbon under various warming trajectories, this study also quantifies the additional radiative forcing, global warming, and permafrost area loss induced by permafrost carbon releaseloss and suggests that permafrost carbon feedback is unlikely to initiate a self-perpetuating global tipping process under both stabilization and overshoot scenarios. In addition, this study reveals quasilinear responses of permafrost area, permafrost carbon, and associated radiative forcing toover a broad range of global warmingtemperature change, providing insights into the implications of permafrost carbon feedback for long-term climate change and mitigation strategies.

Reductions in permafrost area and carbon exhibit a strong correlation with global warming underfor both stabilization and overshoot scenarios. In stabilization scenarios, lower global warming levels effectively mitigate permafrost degradation compared to the SSP5-8.5 scenario, whereas higher global warming levels lead to substantial permafrost degradation due to cumulative warming effects. In overshoot scenarios, permafrost area largely recovers as global warming returns to 1.5 °C levels, though this recovery is delayed by hysteresis effects, with degradation persisting for decades after temperature peaks. Permafrost carbon declines under both stabilization and overshoot scenarios, driven by the dynamic balance between soil carbon inputs and decomposition. The overshoot scenarios partially mitigate permafrost carbon losses compared to the stabilization scenarios at the same global warming levels. Significant carbon losses persist during the cooling and stabilization phases of overshoot scenarios, highlighting the essentially irreversible nature of this process. PermafrostTotal permafrost region soil carbon exhibits a certain degree of recovery under the overshoot scenarios. In fact, soil in these regions even transitions into a carbon sink during the stabilization phase of the overshoot scenarios with high global warming levels, supported by reduced decomposition rates and sustained inputs from vegetation litterfall. However, the higher the overshoot levels, the less recovery there is. These findings underscore the critical role of temporary temperature overshooting levels in affecting long-term permafrost carbon releaseloss and recovery potential.

The responses of permafrost area, permafrost carbon, and associated radiative forcing to a broad range of global warming are nearly linear. The permafrost area and global mean temperature exhibit a quasilinear relation for the global warming ranges from 1.5 °C to 3 °C in both the stabilization and SSP5-8.5 scenarios. The permafrost carbon loss and associated radiative forcing exhibit a quasilinear relation to global warming ranges from 1 °C to 4 °C under the stabilization and SSP5-8.5 scenarios. The sensitivity of permafrost area to global warming derived from the stabilization and SSP5-8.5 scenarios is much substantially higher than 1.6 million km2 °C-1 derived through an equilibrium permafrost model (Liuthe estimates obtained using the Stefan solution (Peng et al., 2021-2023), but our result is close to that derived from an observation-

I think, it's ok to mention the lower number of 1.6 MICm²/·c

Should Mis J'm looking aa.

(see Figure 9a)

constrained equilibrium projection, approximately 3.5 million km² °C-¹ (Nitzbon et al., 2024). Nitzbon et al. (2024) noted that permafrost area decreases quasi-linearly with increasing global mean temperature. However, we found that the relationship holds most strongly for the global warming ranges from 1.5 °C to 3 °C. According to the SSP5-8.5 simulations, the sensitivity of permafrost area to global warming reaches its peak below 1.5 °C global warming. It then decreases to a relatively stable level between 1.5 °C and 3 °C varming levels, and continues to decline beyond the 3 °C varming level. This is in line with Comyn-Platt et al. (2018), who found the feedback processes due to permafrost thaw respond more quickly at temperatures below 1.5 °C. The maximal sensitivity occurring just below 1.5 °C global warming level suggests the fastest permafrost degradation is anticipated to take place within Paris Agreement's warming levels. Our findings have significant implications for the development of mitigation and adaptation strategies addressing permafrost-thaw impacts consistent with keeping global warming at the Paris Agreement's levels.

Our study highlights the substantial permafrost carbon feedback during the cooling phase of overshoot scenarios. Permafrost carbon releaseloss evidently increases global radiative forcing and amplifies global warming. The permafrost carbon feedback can be more profound in temperature stabilization and overshoot scenarios than in high-emissions scenarios. In stabilization scenarios, additional radiative forcing and warming persistently increase over time due to delayed degradation and from permafrost carbon loss continue to rise after temperature stabilizes, driven by delayed permafrost thaw and the attendant positive permafrost carbon feedback. In overshoot scenarios, additional warming almost stabilizes once global warming drops to 1.5 °C levels. During the cooling phase of overshoot scenarios, lower background CO2 concentrations amplify the warming effect of permafrost carbon releaseloss. In contrast, under the high-emissions SSP5-8.5 scenario, the additional warming is limited due to higher background CO2 levels reducing the additional radiative forcing from permafrost carbon releaseloss. The additional permafrost area loss due to permafrost carbon feedback occupies around 5 [2 to 11] % of the total loss by 2300 in the stabilization and overshoot scenarios. These This additional permafrost area losses can be well explained by the sensitivity of permafrost area to global warming and the magnitude of additional warming. The complex interactions between global warming and permafrost degradation emphasize the importance of accounting for these nonlinear effects in climate projections, particularly at 1.5~2 °C global warming levels.

Our results show incomplete recovery of permafrost area under the overshoot scenarios, which is influenced by multiple factors: First, the additional permafrost carbon releaseloss leads to greater additional warming under the overshoot scenarios than the SWL-1.5 scenario, causing additional permafrost degradation. By 2300, the northern high-latitude permafrost regions are 0.01~0.13 °C warmer compared to the SWL-1.5 scenario. Second, the thermal inertia of deep soil layers limits the rate of permafrost recovery. Even after global mean temperatures return to the 1.5 °C target, residual heat accumulated in deeper soil layers during temperature overshoot period continues to inhibit permafrost refreezing, preventing full restoration to its preovershoot state. Third, greater soil carbon loss under overshoot scenarios substantially than in the SWL-1.5 scenario alters the hydrological and thermal properties of soil, affecting the processes that govern carbon cycling (Zhu et al., 2019; Avis, 2012; Lawrence and Slate, 2008), which in turn affects the recovery of permafrost area. Moreover, irreversible shifts in vegetation composition of high-latitude terrestrial ecosystems also contribute to the incomplete recovery of permafrost area under overshoot scenarios. For instance, among the two dominant vegetation types, needleleaf trees continue to expand while C3 grasses decline, even after global temperatures return to the 1.5 °C warming level. These irreversible changes may stabilize the carbon, water, and energy cycles over the permafrost region at different equilibria after overshoot, through the interactions between physical and biophysical processes (de Vrese and Brovkin, 2021), thereby constraining the ability of permafrost to fully recover under the overshoot scenarios.

Although permafrost carbon loss is essentially irreversible, overshoot scenarios exhibit a certain degree of recovery relative to the SWL-1.5 stabilization scenario (Fig. 4b; Fig. S2b). It is therefore eurious cientifically interesting to know whether permafrost carbon under overshoot scenarios will eventually converge with that under SWL-1.5. Our results show that permafrost carbon inputs are consistently higher under overshoot scenarios than under SWL-1.5, while permafrost carbon decomposition differ only slightly between the two (Fig. 5a, c; Fig. S3a, c). This tends to suggest that the smaller permafrost carbon stocks under overshoot scenarios by 2300 would eventually catch up to the levels under SWL-1.5. To assess this potential convergence, we extended our simulations of both SWL-1.5 and overshoot scenarios to the year 2400 (data not shown). Then we estimated the convergence time by calculating the ratio between the difference in permafrost carbon stocks and the difference in net permafrost carbon inputs (i.e., annual permafrost carbon inputs minus decomposition) for the overshoot scenarios relative to the SWL-1.5 scenario. Based on simulation results for the year 2300, the median estimated convergence times for OS-2, OS-3 and OS-4 are 1076, 1008 and 1433 years, respectively. When using results from the year 2400, the corresponding estimates increase to 1377, 1199 and 1568 years. This means that convergence would take even longer if estimated from later simulation results, mainly due to gradually weakened permafrost carbon inputs. The relatively larger permafrost carbon inputs under overshoot scenarios result mainly from increased litterfall during the overshoot phase. The extra litterfall during the overshoot phase gradually moves through the active layer and is transported to the permafrost zone. Over time, however, the effect of this extra litterfall gradually diminishes, leading to a reduction in permafrost carbon inputs. Consequently, it may take extremely long timescales for the overshoot scenarios to fully converge with SWL-1.5 in terms of permafrost carbon stocks. In addition, due to incomplete recovery of permafrost area and persistent changes in surface climate and soil properties, the overshoot scenarios might ultimately fail to converge to SWL-1.5 scenario in terms of permafrost carbon stocks.

Different permafrost carbon releaseloss and associated additional warming under overshoot scenarios confirm the path-dependent fate of permafrost region soil carbon (Kleinen and Brovkin, 2018) and the path-dependent reductions in CO2 emission budgets (MacDougall et al., 2015; Gasser et al., 2018). As Since the permafrost carbon washas accumulated very slowly during the last millions few tens of thousands of years, its releaseloss would be tacked onto the anthropogenic CO2 emissions, and the resulting additional warming poses a challenge to achieving global climate goals by substantially reducing the remaining carbon budget compatible with the Paris Agreement (MacDougall et al., 2015; Natali et al., 2021). In the overshoot scenarios simulated in this study, permafrost carbon releaseloss by 2300 ranges from 60 [35 to 87] PgC to 97 [63 to 135] PgC. The associated resulting additional warming caused by from the release permafrost carbon loss ranges from 0.10 [0.06 to 0.15] °C to 0.18 [0.11 to 0.25] °C. This permafrost carbon feedback contributes a substantial addition on top of 1.5 °C

one 1. of what? burden are it of mospheric corrients?

required?

warming target under overshoot scenarios, and the magnitude of this additional warming rises with the amplitude of overshoot. To accomplish the 1.5 °C target under the OS-2, OS-3, and OS-4 scenarios, anthropogenic carbon emissions would be reduced by amounts equivalent to the permafrost carbon releaseloss. The proportion of carbon removal required to offset permafrost emissions is estimated at 4.9 [2.9 to 7.1] %, 6.5 [4.1 to 9.2] %, and 8.3 [5.4 to 11.6] % by 2300, respectively. Our findings are consistent with previous research utilizing the Monte Carlo ensemble method to evaluate the response of permafrost carbon and its influence on CO2 emission budgets under overshoot scenarios targeting a 1.5 °C warming limit (Gasser et al., 2018). Specifically, for overshoot amplitudes of 0.5 °C (peak warming of 2 °C) and 1 °C (peak warming of 2.5 °C), the reductions in anthropogenic CO2 emissions due to permafrost are estimated to be 130 (with a range of 30–300) Pg CO2 and 210 (with a range of 50–430) Pg CO2, respectively, to meet the long-term 1.5 °C target (Gasser et al., 2018). These results are comparable to our estimates of 60 [35 to 87] PgC under OS-2 and 78 [50 to 111] PgC under OS-3. The differences between the two studies can be partly attributed to different warming trajectories to achieve the same 1.5 °C target. Our study further confirms that ifdeploying negative CO2 emissions were to be used to reverse the anthropogenic elimate change, the delayedwarming would be undermined by substantial post-peak warming arising from permafrost carbon release would reduce its loss, thereby diminishing their overall effectiveness (MacDougall, 2013; Tokarska and Zickfeld, 2015).

This study does not simulate the changes of deep Yedoma carbon under the temperature stabilization and overshoot scenarios. Yedoma deposits represent a significant deep carbon reservoir and are widespread across Siberia, Alaska, and the Yukon region of Canada, having primarily formed during the late Pleistocene, especially in the late glacial period. These deep, perennially frozen sediments are particularly ice-rich, and the freeze-locked organic matter in such deposits can be re-mobilized on short time-scales, representing one of the most vulnerable permafrost carbon pools under future warming scenarios (Schuur et al., 2015; Strauss et al., 2017). According to Zimov et al. (2006), these perennially frozen Yedoma sediments cover more than I million km2, with an average depth of approximately 25 m. Recent estimates place the organic carbon stock in Yedoma deposits at 213 ± 24 PgC, constituting a significant portion of the total permafrost carbon stocks (Strauss et al., 2017). However, the UVic ESCM v2.10 utilized in this study simulates permafrost carbon only within the top 3.35 m of soil, limiting our ability to directly assess the impacts of temperature overshoot on deep Yedoma carbon. Considering their ice-rich nature and potential susceptibility to rapid-thaw processes, we analyzed the average and maximum active layer thickness (ALT) in Yedoma regions (Strauss et al., 2021, 2022) under the simulated scenarios to approximate potential impacts. We find that the average ALT in Yedoma regions remains below 1 m in all stabilization and overshoot scenarios, while the maximum ALT rarely exceeds 3.35 m in overshoot scenarios but does exceed this depth in some stabilization scenarios. However, in all scenarios, the maximum ALT does not exceed 6 m, which is relatively shallow compared to the average depth (~25 m) of Yedoma deposits (Figure S6). Consequently, the impact on Yedoma is considered expected to be minimal in all scenarios, and the effect of overshoot scenarios on the deep Yedoma carbon is relatively minor compared to stabilization scenarios as well.

This study, like previous ones, uncovers considerable uncertainty in projections of permafrost carbon under global warming. The uncertainty represented by perturbed model parameters for each scenario can be interpreted as model uncertainty. We note that model uncertainty in permafrost carbon releaseloss gradually increases with the peak warming level

---

## Author Response (AR3)

**Dear Vivek,**

We sincerely appreciate the time you devoted to annotating our manuscript. In response, we have revised every section where you requested clarification or additional context; your comments have been invaluable in enhancing the manuscript's quality.

First, every language issue you raised has been corrected in the revised manuscript. Below, we elaborate on several key points; your handwritten comments appear in black, and our responses are shown in blue.

1) Line 122-125: I'm confused again. This indicates 3 pools: usual C, permafrost C that has thawed, permafrost C that has not thawed?

Thank you for your comment. In the UVic ESCM, permafrost carbon is stored in a depth-resolved pool consisting of six discrete layers, each acting as a sub-pool. This structure allows the permafrost carbon pool to contain both thawed and frozen carbon simultaneously. To emphasize this point, we have added the following clarification: "The coexistence of thawed and frozen permafrost carbon is attributed to the depth-resolved soil carbon scheme."

2) Line 214: "total permafrost region soil carbon" is not defined yet. & Line 245: This should come earlier than the first use of "total permafrost region soil carbon" on page 8.

Thank you for pointing this out. We agree that the definition of "total permafrost region soil carbon" should be provided prior to its first use. To address this, we have moved the explanatory sentence – "It is worth noting that the permafrost region for carbon-related variables is defined as the area where permafrost carbon exceeds zero. This differs from the permafrost area, which is defined as the area where soil temperature remains below 0 °C for at least two consecutive years." – to the end of the third paragraph in Section 2.1 (Model Description).

3) Line 355: What does "that" refers to?

Thank you for your comment. We have revised the sentence by replacing "that" with "the expansion" to avoid ambiguity. The revised sentence now reads: "Needleleaf trees expand slowly and continuously in the permafrost region in both overshoot and stabilization scenarios, whereas the expansion of shrubs closely follows the trajectory of global mean temperature."

4) Line 358: Sorry is 62% the result from your model simulations? Why the use of "expected"?

Yes, the 62% value is derived from the output of our model simulations. We initially used "expected" to indicate that this result is slightly above the 24~52% range projected for 2050 by Pearson et al. (2013), and therefore falls within a plausible range. To avoid ambiguity, we have replaced "expected" with "projected" in the revised manuscript.

5) Figure 7: We don't have panel d anymore

Thank you for pointing this out. We have corrected the figure caption to remove the reference to panel d.

**6) Line 550: C lost from frozen soils $\neq$ C entering the atmosphere. Why?**

Thank you for your question. We agree that the sentence was potentially misleading and have revised it accordingly. As noted by Nitzbon et al. (2024), their estimates of total carbon loss from frozen soils likely represent an upper bound on the actual atmospheric release. This is because Nitzbon et al. (2024) inferred carbon loss based on the presence or absence of permafrost, whereas in reality, permafrost carbon may persist in the soil for some time even after permafrost disappears. To avoid confusion, we have removed the latter half of the original sentence: ", yet only a fraction of that carbon is ultimately released into the atmosphere, as noted by Nitzbon et al. (2024)."

**7) Line 603: I think, it's ok to mention the lower number of 1.6 M km2/°C**

Thank you for your comment. The value was derived from Liu et al. (2021), where the permafrost sensitivity to climate warming was calculated based on the regional mean warming over permafrost regions, rather than the global mean temperature increase. To ensure conceptual consistency across comparisons, we therefore chose to exclude this figure in the current revision. By the way, we initially followed the interpretation of Liu et al. (2021) as presented in the IPCC AR7 Chapter 9, Page 1283, Left column, Lines 10-14. However, we found that the IPCC AR7 incorrectly interpreted the results of Liu et al. (2021).

**8) Line 609: Should this be 2 °C**

Thank you for your comment. We agree that this part of the manuscript was not clearly expressed. In fact, two different methods were used to assess the relationship between permafrost area and global mean temperature based on the SSP5-8.5 scenario. We have now clarified this point in Section 3.3 (Linearity of Permafrost Response and Feedback), third paragraph, which reads: "Nitzbon et al. (2024) reported a quasilinear decrease in the equilibrium permafrost extent to global warming, with a rate of approximately 3.5 million km2 °C-1. This quasilinear relation holds for global warming ranges from 0 °C to 4 °C, derived from the empirical relationship between the local permafrost fraction and the annual mean global temperature. However, when applying a local regression method across the full SSP5-8.5 trajectory (Fig. 9a), we find that the quasilinear relationship only holds between 1.5 °C and 2 °C. In contrast, when applying a simple linear regression, a quasilinear behavior still holds over the 1.5 °C to 3 °C warming range (Fig. 10a)."

In the sentence at Line 609, we retained the 1.5 to 3 °C range, but clarified the context to avoid confusion. The revised sentence now read: "However, we found that this relationship holds most strongly within 1.5 °C to 3 °C global warming range when using a linear regression approach (Fig. 10a)."

**9) Line 632: was this mentioned earlier? Or did I miss this.**

Thank you for your careful reading. You are correct that this information was not mentioned earlier in the manuscript. To improve the clarity and logical flow of the results, we have now moved the relevant sentence to an earlier position in Section 3.1 (Lines 272 - 276), where we first report the incomplete recovery of permafrost area under the overshoot scenarios. The revised text reads: "This indicates that permafrost area is nearly reversible and largely follows the global mean temperature trajectory, recovering as temperature reduction, consistent with previous studies (MacDougall, 2013; Lee et al., 2021; Schwinger et al., 2022). The incomplete recovery partly results from slightly higher regional temperatures in permafrost regions. By 2300, the northern high-latitude permafrost regions are  $0.01 \sim 0.13$  °C warmer compared to the SWL-1.5 scenario."

**10) Line 676: Sorry, there are % of what? Atmospheric CO2 burden or emissions?**

Thank you for your question. The percentages refer to the cumulative anthropogenic CO2 emissions, not atmospheric CO2 burden. To improve clarity, we have revised the text to read: "To accomplish the 1.5 °C target under the OS-2, OS-3, and OS-4 scenarios, anthropogenic carbon emissions would need to be reduced by amounts equivalent to the permafrost carbon loss. The proportion of carbon removal required to offset permafrost emissions corresponds to 27 [16 to 39] % under OS-2, 10 [6 to 15] % under OS-3, and 7 [4 to 9] % under OS-4, relative to the cumulative carbon removal from the peak warming to the 1.5 °C target sustained until 2300."

We would also like to note that the percentage values reported in the current version differ from those in the previous version. This revision results from a change in the reference baseline: instead of permafrost emissions to cumulative anthropogenic CO2 emissions from 1850 to 2300, we now refer only to the cumulative emissions that would need to be removed from the peak warming to reach the 1.5 °C target. This adjustment better reflects the additional carbon removal effort required due to permafrost carbon release in the context of achieving the climate goal.

Your sincerely,

Min Cui, Duoying Ji, and Yangxin Chen